# Comparative analysis of Copernicus, TanDEM-X and UAV-SfM DEMs to estimate lavaka (gully) volumes and mobilization rates in the Lake Alaotra region (Madagascar)

Liesa Brosens[1,2], Benjamin Campforts[3], Gerard Govers[2], Emilien Aldana-Jague[4], Vao Fenotiana Razanamahandry[2], Tantely Razafimbelo[5], Tovonarivo Rafolisy[5], and Liesbet Jacobs[2,6]

[1]Research Foundation Flanders (FWO), Egmontstraat 5, 1000 Brussels, Belgium
[2]Department of Earth and Environmental Sciences, KU Leuven, Leuven, Belgium
[3]Institute for Arctic and Alpine Research, University of Colorado at Boulder, Boulder, CO
[4]Earth and Life Institute, Georges Lemaître Centre for Earth and Climate Research, Université Catholique de Louvain, Louvain-la-Neuve, 1348, Belgium
[5]Laboratoire des Radio Isotopes, Université d'Antananarivo, Antananarivo, Madagascar
[6]Ecosystem & Landscape Dynamics, Institute for Biodiversity and Ecosystem Dynamics, University of Amsterdam, Amsterdam, Netherlands

**Correspondence:** Liesa Brosens (liesa.brosens@kuleuven.be)

**Abstract.** Over the past decades, developments in remote sensing have resulted in an ever growing availability of topographic information on a global scale. A recent development is TanDEM-X, an interferometric SAR mission of the Deutsche Zentrum für Luft- und Raumfahrt, providing near-global coverage and 12 m resolution DEMs. Moreover, ongoing developments in uncrewed aerial vehicle (UAV) technology has enabled acquisitions of topographic information at a sub-meter resolution.

Although UAV products are generally preferred for volume assessments of geomorphic features, their acquisition remains time-consuming and is spatially constrained. However, some applications in geomorphology, such as the estimation of regional or national erosion quantities of specific landforms, require data over large areas. TanDEM-X data can be applied at such scales, but this raises the question of how much accuracy is lost because of the lower spatial resolution. Here, we evaluated the performance of the 12 m TanDEM-X DEM to i) estimate gully volumes, ii) establish an area-volume relationship, and iii)

determine mobilization rates, through comparison with a higher resolution (0.2 m) UAV-SfM DEM and a lower resolution (30 m) Copernicus DEM. We did this for six study areas in the Lake Alaotra region (central Madagascar) where *lavaka* (gullies) are omnipresent and surface area changes over the period 1949-2010s are available for 699 lavaka. Copernicus derived lavaka volume estimates were systematically too low, indicating that the Copernicus DEM is too coarse to accurately estimate volumes of geomorphic features at the lavaka-scale (100 - $10^5$ m$^2$). Lavaka volumes obtained from TanDEM-X were similar to UAV-

SfM volumes for the largest features, whereas the volumes of smaller features were generally underestimated. To deal with this bias we introduce a breakpoint analysis to eliminate volume reconstructions that suffer from processing errors as evidenced by significant fractions of negative volumes. This elimination allowed the establishment of an area-volume relationship for the TanDEM-X data with fitted coefficients within the 95% confidence interval of the UAV-SfM relationship. Our calibrated area-volume relationship enabled us to obtain large-scale lavaka mobilization rates ranging between $18 \pm 3$ and $311 \pm 82$ t ha$^{-1}$ yr$^{-1}$

for the six different study areas, with an average of $108 \pm 26$ t ha$^{-1}$ yr$^{-1}$ for the full dataset. These results indicate that current

lavaka mobilization rates are two orders of magnitude higher than long-term erosion rates. With this study we demonstrate that the global TanDEM-X 12 m DEM can be used to accurately estimate volumes of gully-shaped features at the lavaka-scale (100 - $10^5$ m$^2$), where the proposed breakpoint-method can be applied without requiring the availability of a higher resolution DEM. Furthermore, we use this information to make a first assessment of regional lavaka erosion rates in the central highlands of Madagascar.

## 1  Introduction

Over the past decades advanced technology has become increasingly available for the assessment of surface topography: SfM (structure-from-motion) algorithms applied to UAV (uncrewed aerial vehicle) imagery now allow centimeter-scale resolution, thereby revolutionizing the way we study earth-surface processes (Passalacqua et al., 2015; Tarolli, 2014; Clapuyt et al., 2016). Obtaining sub-meter resolution DEMs from UAV-SfM still requires substantial fieldwork and is spatially limited due to the nature of the technology (Bangen et al., 2014). On the other hand, TanDEM-X is a spaceborne product with global coverage at 12 m resolution and, while being less detailed and accurate than these sub-meter resolution DEMs, is a major step forward in comparison to 30 m resolution DEMs with a global coverage (Mudd, 2020).

This raises the question to which extent TanDEM-X imagery can be used to map three-dimensional morphological features requiring a higher degree of topographical detail over relatively large areas (> 10 km$^2$). One example is the use of remotely sensed data to map the process of gully formation, which is known to significantly contribute to surface erosion (e.g. Poesen et al., 2003; Vanmaercke et al., 2021; Frankl et al., 2021). The mapping and monitoring of gully erosion was conventionally based on time-consuming and spatially limited field surveys (Castillo et al., 2012; Evans and Lindsay, 2010; Guzzetti et al., 2012). More recently, however, (sub-)meter resolution DEMs have enabled the development of (semi-)automated gully-delineation and volume determination methods (Niculiță et al., 2020; Evans and Lindsay, 2010; Perroy et al., 2010; Eustace et al., 2009; Liu et al., 2016). TanDEM-X has, for example, already been successfully used for automatic gully detection (Vallejo Orti et al., 2019).

Not only the extent to which TanDEM-X data can be used to estimate gully volumes, but also its capability in establishing accurate area-volume relationships is important to evaluate. This latter question is relevant since sub-meter resolution surface imagery from a multitude of sources and moments in time is now globally and freely available through, for example, Google Earth (Fisher et al., 2012). This imagery can be used to identify geomorphic features and estimate their surface area with great detail. Area-volume or length-volume relationships then enable to obtain estimates of volume-changes over time when historical imagery from which areas or lengths can be derived is available. Work on gully and landslide erosion has shown that the establishment of these relationships enables us to estimate sediment mobilization rates (i.e. the average annual volume of hillslope material displaced per unit area) over large spatial and temporal scales (e.g. Malamud et al., 2004; Larsen et al., 2010; Hovius et al., 1997; Guzzetti et al., 2012; Broeckx et al., 2019; Frankl et al., 2013; Kompani-Zare et al., 2011). Furthermore, work on gully headcut retreat rates and associated sediment mobilization has indicated the importance of long

measurement periods: large year-to-year variations result in very large (>100%) uncertainties over short (<5 years) measuring periods (Vanmaercke et al., 2016).

Here, we evaluate the performance of TanDEM-X to estimate gully volumes and to establish area-volume relationships by comparing estimates obtained from a 0.2 m resolution UAV-SfM DEM, the 12 m resolution TanDEM-X DEM the 30 m resolution Copernicus DEM. The resolution of a DEM should be viewed in relation to the size of the landform, where sampling theory states that landforms should have dimensions of at least twice the DEM resolution (Theobald, 1989; Frankl et al., 2013). This would mean that a gully should have a theoretical minimum size of 0.16 m$^2$, 576 m$^2$ and 3600 m$^2$ for the UAV-SfM,

TanDEM-X and Copernicus DEM, respectively, to be represented in the DEM. We used the *lavaka* of the central highlands of Madagascar as a case study. Lavaka are amphitheater shaped gullies that are on average 60 m long, 30 m wide and 15 m deep, with small outlets (Cox et al., 2010; Wells and Andriamihaja, 1993). They are omnipresent in the central highlands, leaving the landscape filled with 'holes' ('lavaka' in Malagasy). Unlike other gullies they typically lack surface feeder channels and tend to form on mid-slopes, broadening uphill trough headward erosion (Wells et al., 1991; Wells and Andriamihaja, 1993). While

Madagascar is often claimed to experience amongst the highest global erosion rates due to the presence of lavaka (Milliman and Farnsworth, 2011; Randrianarijaona, 1983), the amount of sediment that is directly produced by lavaka is currently unknown.

The objectives of our study are therefore to evaluate the performance of TanDEM-X to i) estimate lavaka volumes, ii) establish accurate area-volume relationships and iii) obtain a first estimate of lavaka mobilization rates. We derived lavaka volumes and mobilization rates for an existing dataset containing 699 digitized lavaka in six study areas in the Lake Alaotra

region at three moments in time: 1949, 1969 and the 2010s. In a first step, lavaka volumes were calculated for the 2010s lavaka polygons from the DEM as the difference between a reconstructed pre-erosion surface and the current topography. Next, a lavaka area-volume relationship was established between the current lavaka areas (2010s) and calculated volumes. Finally, this relationship was applied to the historical dataset with lavaka areas in 1949, 1969 and the 2010s. This enabled to calculate lavaka volumes at each of these timesteps and the consequent derivation of volumetric growth rates and lavaka mobilization

rates for each of the six study areas. This procedure was followed for a 0.2 m resolution UAV-SfM DEM, the 12 m resolution TanDEM-X DEM and the 30 m resolution Copernicus DEM.

## 2 Material and methods

### 2.1 Study area and lavaka dataset

Six study areas (SA) of ca. 10 km$^2$ were selected in the northeastern part of the central highlands of Madagascar in the area

surrounding Lake Alaotra (Fig. 1(a)). The lake is located in the seismically active NE-SW oriented Alaotra-Ankay graben structure and is surrounded by convex-shaped, deeply weathered (> 10 m) regolith-covered hillslopes that developed on the Precambrian crystalline basement (Kusky et al., 2010; Riquier and Segalen, 1949; Bourgeat, 1972; Mietton et al., 2018). The climate is characterized by a distinct dry and wet season with the regular occurrence of tropical cyclones. The mature rounded hillslopes are covered with open grasslands and contain one of the highest lavaka densities of the country, with up

to 14 lavaka km$^{-2}$ (Cox et al., 2010; Voarintsoa et al., 2012). The lowland areas bordering the lake consist of swamps in the

SW and vast rice fields elsewhere, producing the majority of rice for the country (Lammers et al., 2015). For the six selected study areas digitized lavaka polygons were generated from orthorectified and georeferenced historical aerial images from 1949 and 1969 (2.4 m resolution) and from recent (2011-2018 referred to as 2010s) satellite imagery (WorldView-2, 0.5 m resolution) (Fig. 1(a), Table B1). All shapefiles have WGS84-UTM 39S (EPSG: 32739) coordinates. The dataset (available at https://doi.org/10.6084/m9.figshare.c.5236322.v1) contains the changes in surface area of 699 lavaka over the period 1949-2010s for SA1-5 and 1969-2010s for SA6. Each study area contains 50 to 173 lavaka, resulting in lavaka densities between 4 and 17 lavaka km$^{-2}$ (Table B1).

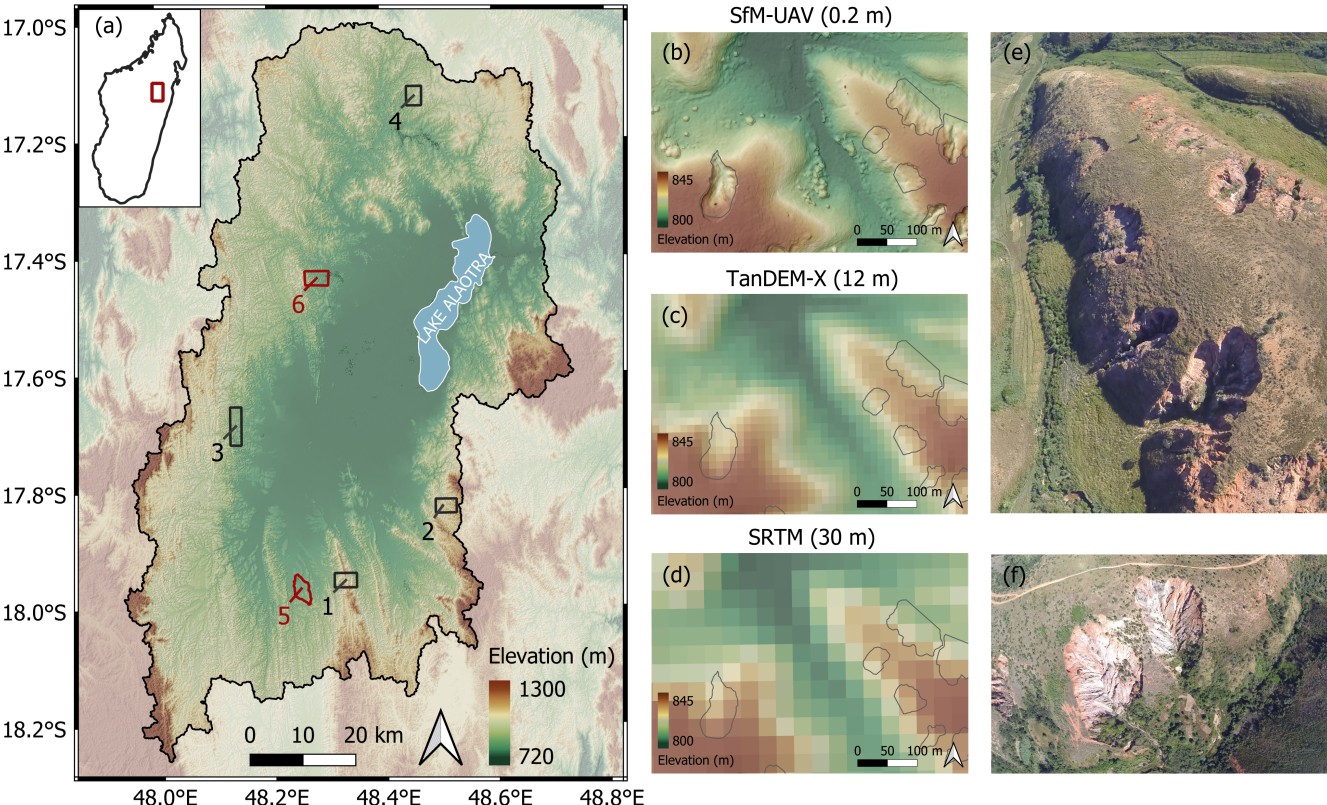

**Figure 1. Study areas, examples of each digital elevation model (DEM) and lavaka examples.** (a) Six study areas of ca. 10 km$^2$ in the Lake Alaotra catchment shown on the TanDEM-X DEM with hillshade. The UAV-SfM DEM is available for study area 5 and 6 (red) (data collected June 2018). (b)-(d) Examples of the Copernicus (AIRBUS, 2020a), TanDEM-X (Krieger et al., 2007) and UAV-SfM DEM (data collected June 2018) in SA6 located at 48°15'18.6"E 17°58'51.7"S with hillshade. Grey outlines indicate the digitized lavaka. (e) UAV fish-eye picture from 200 m height of the eastern ridge shown in (b)-(d). (f) UAV fish-eye picture (200 m height) from two typical amphitheater-shaped lavaka (pictures taken June 2018).

## 2.2 Digital elevation models (DEMs)

Lavaka volumes were determined from three digital elevation models with a range of horizontal resolutions. For two study areas a 0.2 m resolution UAV-SfM DEM was obtained from a field campaign in 2018. For all study areas the 12 m and 30 m resolution TanDEM-X and Copernicus DEMs are available. All DEMs were transformed to WGS84-UTM39S (EPSG: 32739) coordinates using a nearest neighbour resampling method.

### 2.2.1 UAV-SfM DEM (0.2 m)

For study area 5 and 6 (Fig. 1(a) in red) a UAV-field survey was carried out in June 2018 to obtain a 0.2 m resolution DEM. In order to cover a large area during a limited amount of time with a high spatial resolution the post-processing kinematic (PPK) georeferencing approach as developed by Zhang et al. (2019) was used. The UAV-images were directly georeferenced by using a RTK (real-time kinematic) receiver on the UAV which was connected to a RTK base station.

We used a custom made quadcopter UAV with DJI N3 flight controller and fish-eye action camera (Go Pro Hero 3, 12 megapixels, $4000 \times 3000$ pixels, with 2.92mm F/2.8 123° HFOV lens). A compact Tallysman TW2710 multi-Global Navigation Satelite System(GNSS)-RTK-receiver antenna (Reach RTK kit, Emlid Ltd, 23 cm height) was mounted on an aluminium plate centered above the camera. The RTK-receiver was connected to a single-board computer in order to synchronize the GPS time with the geotagging of the images. The RTK base station (Emlid Ltd) provided the positioning correction input. It was mounted on a tripod and located at a fixed position at the center of each study area during the flights. Flights were carried out at 200 m height at a speed of 8 m s$^{-1}$. Pictures were taken every 3.8 seconds resulting in an average ground sampling distance of 0.17 m (Fig. 1(e)-(f)).

The raw RTK-GPS data from the receiver were corrected with the data from the base station and post-processed using the RTKLib package (Takasu and Yasuda, 2009). A fix-and-hold method with 20° satellite elevation mask was used to correct the positions and geotag the images. The geotagged images were processed with Pix4D software using the default settings with the vertical and horizontal accuracy set at 0.5 m. For the generation of the DEM no surface smoothing or filtering was applied. The resulting DEM for both study areas has a resolution of ca. 0.2 m (Fig. 1(b)).

This method was reported to result in a robust and accurate alternative for georeferencing based on ground control points (GCP) with a MAE of 0.02 m and RMSE of 0.03 m for the vertical accuracy and a precision of 0.04 m (Zhang et al., 2019). Comparable studies over relatively flat areas with an UAV-RTK setup report similar vertical accuracies with RMSE values between 0.03 and 0.07 m (Taddia et al., 2020; Stott et al., 2020). UAV-SfM surveys with GCP's over more complex terrain report higher RMSE values between 0.10 and 0.45 m (Clapuyt et al., 2016; Cook, 2017). Given the reported high accuracies of optical acquisitions that are georeferenced with RTK-GPS data, this DEM surface can be considered as the reference of the 'true' elevation (Grohmann, 2018). In this study we therefore consider the UAV-SfM DEM as the ground-truth reference.

### 2.2.2 TanDEM-X DEM (12 m)

The TanDEM-X (TerraSAR-X add-on for digital elevation measurements) mission was launched in 2010 by the public-private
partnership between the German Aerospace Center (DLR) and EADS Astrium GmbH (Krieger et al., 2007). Its configuration
consists of two synthetic aperture radar (SAR) satellites flying in close formation, thereby forming a large X-band single-pass
interferometer. The resulting global DEM has a horizontal resolution of 0.4 arcsecond (ca. 12 m) and aims at <2 m relative
and <10 m absolute vertical accuracy (Krieger et al., 2007, 2013; Wessel, 2016). The final TanDEM-X DEM was published
in 2016 and consists of data collected between December 2010 and early 2015, where all land surfaces were imaged at least
twice and up to 7 or 9 times in difficult terrain (Rizzoli et al., 2017, Fig. 1(c)). Our study areas were imaged 5 to 9 times with
an average of $7 \pm 1$, indicating a good coverage. A good performance of the TanDEM-X DEM has been reported, with a final
global absolute vertical accuracy of 3.49 m and relative vertical accuracy of 0.99 m and 1.37 m on flat (<20°) and steep (>20°)
terrain, respectively (Rizzoli et al., 2017). These results are in line with Wessel et al. (2018) and Purinton and Bookhagen
(2017) who reported absolute vertical accuracies of $0.20 \pm 1.5$ m and $-1.41 \pm 1.97$ m, respectively.

### 2.2.3 Copernicus DEM (30 m)

The 1 arcsecond (ca. 30 m) resolution global Copernicus DEM (GLO-30) was released in 2021 by the European Space Agency
(ESA) and AIRBUS. The DEM is based on the WorldDEM$^{TM}$ which, on its turn, is based on edited and smoothed radar satellite
data acquired during the TanDEM-X Mission (AIRBUS, 2020a). The reported global absolute vertical accuracy is 2.17 m with a
RMSE of 1.68 m. The relative vertical accuracy is smaller than 2 m for <20° slopes and less than 4 m on >20° slopes (AIRBUS,
2020b). Given its recent release, only limited additional validation has been carried out (Guth and Geoffroy, 2021). A lower
absolute vertical accuracy of GLO-30 has been reported for mountainous areas in Europe with RMSE values between 7 and 14
m (Marešová et al., 2021). These estimates should, however, be viewed as maximum estimates as these high relief terrains are
one of the most challenging settings for DEM acquisitions. Upon comparison of different global 1 arcsecond DEMs, Purinton
and Bookhagen (2021) concluded that the Copernicus DEM provides the highest quality landscape representation and should
be the preferred DEM for topographic analysis in areas that lack higher resolution DEMs. They furthermore report a high
inter-pixel consistency for both the TanDEM-X and Copernicus DEM, indicating low relative vertical errors for these DEMs.

### 2.3 Lavaka volume quantification

Individual lavaka volumes were determined as the difference between the current surface and a reconstructed pre-erosion
surface. This was done by developing an automated workflow in PyQGIS written in QGIS version 3.16.10 (code and example
dataset available at https://doi.org/10.5281/zenodo.5768418). The automated PyQGIS workflow consists of four steps which
are explained in detail below.

#### INPUT DATA

Three input files are required to run the automated volume-procedure: i) a shapefile containing the digitized lavaka
(gully) outlines, ii) a DEM raster file and iii) a shapefile containing the digitized outlines of the region surrounding the

lavaka that is not affected by erosion. A manual delineation-procedure was followed to obtain this surface unaffected by erosion, where a horseshoe-shaped polygon was drawn around each individual lavaka on the hillslope parts that were unaffected by erosion (Fig. 2(a)).

**STEP 1: Interpolate the pre-erosion surface**

First, the DEM raster layer is clipped with the horseshoe-shaped polygon in order to extract the pixels not affected by gully erosion. All pixels that fall within this polygon are extracted in order to have a minimum width of one pixel. Next, one point per clipped DEM pixel is generated and used as input for the interpolation. Finally, these points are used to interpolate the pre-erosion surface. Five interpolation methods were tested, of which the method with the lowest error was applied to the lavaka dataset (see section 2.4.1 and section 3.1). Examples of the interpolated pre-erosion surface are shown in Fig. 2 for TIN (b) and regularized spline (c) interpolation.

**STEP 2: Calculate elevation difference**

The current DEM is subtracted from the interpolated pre-erosion surface. The result is a difference raster with positive values indicating a current surface that is lower than the reconstructed pre-erosion surface. Negative values indicate that the current topography is higher than the reconstructed topography.

**STEP 3: Elevation difference clipped to lavaka extent**

The lavaka extent, which is given by the digitized lavaka polygons from the 0.5 m resolution WorldView-2 imagery from 2011-2018, is clipped from the elevation difference raster. In this way a raster with the elevation difference over the lavaka area is obtained (Fig. 2(b)-(c)). If the lavaka is smaller than one pixel (0.04 $m^2$, 144 $m^2$ and 900 $m^2$ for the UAV-SfM, TanDEM-X and Copernicus DEM, respectively) the resulting raster is empty and no volume can be calculated.

**STEP 4: Export results**

The unique values report of the lavaka elevation difference raster is exported. It contains the unique elevation values, their count and dimensions of the raster pixels. These results are used to calculate the volumes of each lavaka.

A manual delineation of the region surrounding the lavaka that is unaffected by erosion was preferred over an automated pre-erosion interpolation (e.g. Evans and Lindsay, 2010) or interpolation based on the lavaka outlines for two reasons. First, digitized lavaka outlines from aerial imagery are often located on DEM pixels that already have lower values. This is especially the case for the coarser resolution DEMs due to surface smoothing (e.g. Fig. 1(c)-(d)). Second, the very dense presence of lavaka often results in a highly dissected topography and a near absence of topography not affected by lavaka erosion, requiring a precise identification of the areas not affected by erosion. For some lavaka no horseshoe-shaped polygons could be delineated. In other cases, lavaka were grouped in one enveloping polygon when they were located next to each other (Fig. C1).

From the exported values report the lavaka volume was calculated as the product of the elevation difference with the pixel area. Both positive and negative elevation differences occur, where a positive difference indicates that the current surface is lower than the reconstructed pre-erosion surface. Negative values indicate that the current topography is higher than the reconstructed topography. Given the presence of both positive and negative elevation differences, three types of volumes (*V*

[m$^3$]) could be calculated for each lavaka: i) the positive volume ($Vpos$) calculated by summing only the positive elevation differences, ii) the negative volume ($Vneg$) calculated from the negative elevation differences, and iii) the total volume ($Vtot$) as the difference between the positive and negative volume. If not further specified the term 'volume' refers to the positive volume.

The lavaka volumes as obtained from the different DEMs were compared in two ways: i) pairwise comparison using only the lavaka for which the volume was determined for each DEM enabling a one-to-one comparison, and ii) using all the data in order to establish the most robust relationships calibrated on a higher number of observations. The number of lavaka for which the volume was determined depends both on the availability of the UAV-SfM DEM (only in SA 5 and 6) and the resolution of the DEM, where no volume could be calculated for the smallest features using the 12 and 30 m resolution DEMs.

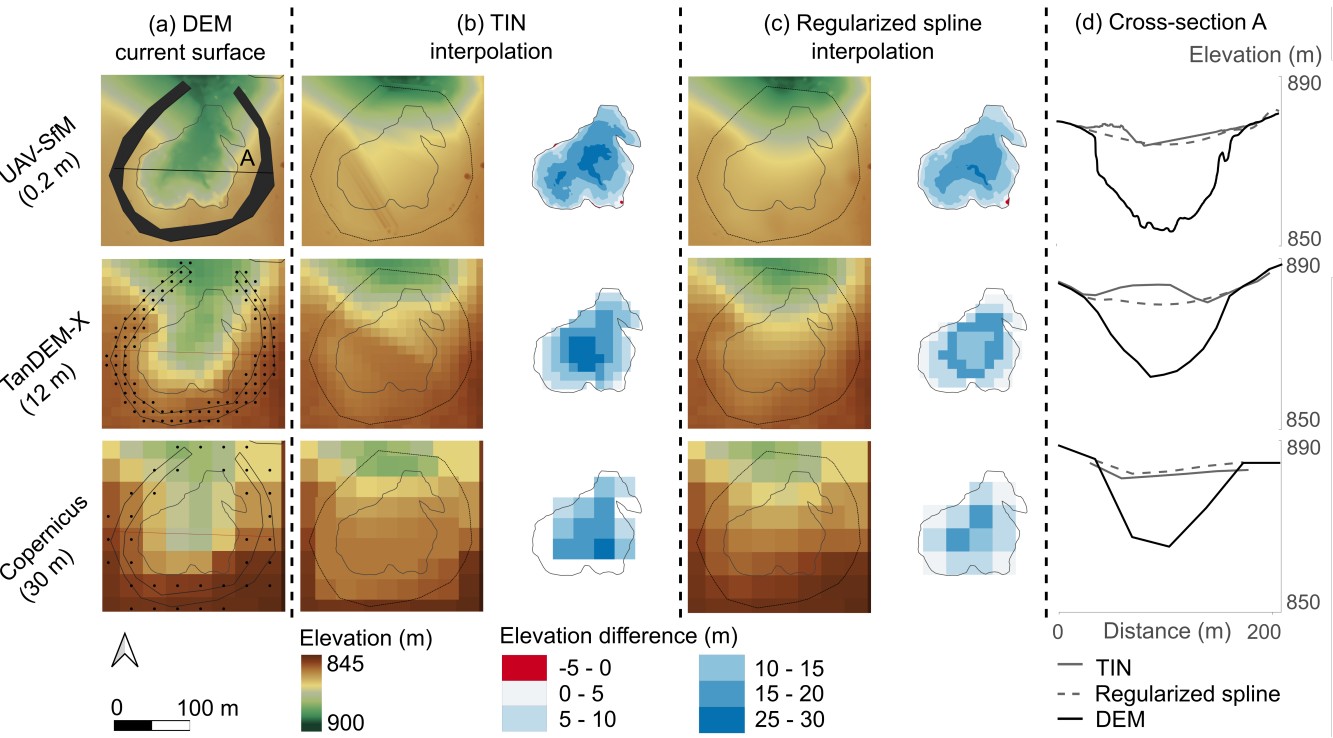

**Figure 2. Lavaka volume determination workflow.** Lavaka volumes were calculated for each individual lavaka following an automated workflow for the three studied DEMs: UAV-SfM (0.20 m, top, data collected June 2018), TanDEM-X (12 m, middle, Krieger et al. (2007)) and Copernicus (30 m, bottom, AIRBUS (2020a)). (a) The digitized lavaka outline (grey), manually determined horseshoe-shaped polygon on the unaffected hillslope surrounding the lavaka and current DEM are the three required inputs for the automated volume-procedure. The DEM pixels that are not affected by erosion are clipped from the DEM with the horseshoe-shaped polygon and one point per pixel is generated. The pre-erosion surface is then reconstructed by interpolating between these points (STEP 1). Two interpolation methods are shown as an example here: TIN (b) and regularized spline (c) interpolation. The grey polygon indicates the outer edge of the interpolated area. The elevation difference between the interpolated pre-erosion surface and current DEM surface is then calculated, which is clipped to the lavaka extent (STEP 2-3). (d) Cross sections of transect A for the DEM, TIN and regularized spline interpolation

### 2.4 Volume uncertainty assessment

Estimated lavaka volumes, determined as the difference between an interpolated pre-erosion surface and the current DEM surface, will entail a number of uncertainties and errors. Given our application, we address two types of uncertainty or error in our volume estimates: i) the interpolation error, and ii) the relative height error of the DEM.

#### 2.4.1 Interpolation error

The pre-erosion surface of the lavaka is reconstructed by interpolating between the DEM pixels (one point per pixel) that are not affected by gully erosion. While it is impossible to assess the real interpolation error for a lavaka - the pre-erosion topography is simply unknown-, this error can be estimated by interpolating the surface at locations where no lavaka are present and the pre-erosion surface is thus known. This method, which is similar to Bergonse and Reis (2015), does not only allow to estimate the uncertainties on derived lavaka volumes, but also allows to objectively select the best interpolation method for a given topographic setting.

Five different lavaka polygons with sizes that span the range of our lavaka dataset were selected: 100 m$^2$, 1000 m$^2$, 5000 m$^2$, 10 000 m$^2$ and 20 000 m$^2$. Each polygon was duplicated ten times, resulting in 50 lavaka polygons. These were placed on unaffected convex-shaped hillslopes on which lavaka typically occur, together with the corresponding horseshoe-shaped polygons. Our automated lavaka volume quantification method (section 2.3) was then applied to this dataset using five different interpolation methods, where the difference between the interpolated surface and the DEM was calculated. This elevation difference gives the interpolation error, as a perfect interpolation would result in a surface identical to the original DEM.

We tested five commonly used interpolation methods for continuous data with their default parameter settings. The first two algorithms are based on a linear method where Delaunay triangles are constructed from the nearest neighbour points, resulting in non-smooth surfaces: i) Linear interpolation (GDAL, 2021), and ii) Triangulated Irregular Network (TIN) interpolation (QGIS, 2020). We also tested three spline interpolation methods, that support the creation of curved surfaces in areas without data: iii) bilinear spline (GRASS, 2021), iv) bicubic spline (GRASS, 2021) and v) regularized spline with tension (GRASS, 2003). The bilinear and bicubic spline interpolations are 2D piece-wise non-zero polynomial functions calculated within a limited 2D area, where the Tykhonov regularization parameter affects the smoothing of the surface (smoothing = 0.01). Linear spline is based on 4 inputs to derive the coefficients, whereas bicubic spline uses 16 inputs, typically resulting in more precise outcomes (Brovelli et al., 2004; GRASS, 2021). In the regularized spline with tension algorithm (tension = 40, no smoothing) the tension parameter tunes the character of the resulting surface from thin plate to membrane and the smoothing parameter controls the deviation between points and the resulting surface (Mitášová and Mitáš, 1993; GRASS, 2003).

For all three DEMs and five interpolation methods the difference between the interpolated surface and DEM surface was calculated for the 50 lavaka polygons (example in Fig. C2 and Fig. C3). Based on the obtained height differences between the interpolated and original DEM surface several error metrics were calculated (mean, median, root mean squared error (RMSE), mean absolute error (MAE) and standard deviation (std)), which were then used to i) identify the best interpolation method and ii) estimate the interpolation error. To estimate the interpolation error, it was verified if the mean interpolation error of a lavaka

depends on its area. If a significant relationship was absent, the mean $\pm$ std interpolation error was used for all lavaka for that DEM. In the other case, where a relationship between lavaka area and mean lavaka interpolation error is present, we estimated the interpolation error based on the fitted linear relationship between both variables. Uncertainties on the fitted coefficients were taken into account by drawing $10^4$ random Monte Carlo coefficient values from a normal distribution with known fitted mean and std, where we used Gaussian copula to account for the correlation between both coefficients.

### 2.4.2    Relative height error

Typically, the performance of a DEM is assessed by considering its absolute vertical accuracy, i.e. the difference in DEM elevation and a high resolution reference dataset (often LIght Detection And Ranging (LiDAR) or Global Navigation Satelite System (GNSS) datapoints) (AIRBUS, 2020a; Wessel, 2016). In our application this absolute height error is, however, not the most important DEM error metric. Our volumes are the relative difference between the interpolated and real DEM, which is

not influenced by the absolute vertical deviation of the DEM as long as this absolute deviation is the same for all considered pixels. Rather we are interested in relative pixel-to-pixel errors, as these are more likely to affect the estimated volumes.

We assume this relative height error to be negligible for our 0.2 m resolution UAV-SfM DEM, given that the reported accuracy and precision values are in the order of a few centimeters (Zhang et al., 2019; Taddia et al., 2020; Stott et al., 2020). For the TanDEM-X and Copernicus DEM we use the Height Error Masks (HEM) that are provided as auxiliary files. The

height error mask gives the theoretical random height error for each pixel in the form of the standard deviation which results from the interferometric phase and the combination of different coverages. This error is considered to be a random error and does not include any contributions of systematic errors (Wessel, 2016; Wessel et al., 2018; AIRBUS, 2020a).

For each lavaka we calculated the mean $\pm$ std HEM-value, which we then use to estimate the relative DEM uncertainty. A positive correlation between mean HEM and lavaka area is observed, where larger lavaka have higher mean relative height

uncertainties (Fig. C4(a)). By calculating the mean HEM for a lavaka we use the lavaka as the observational unit, as we also did for the interpolation error. By doing so we implicitly assume that all lavaka pixels are perfectly autocorrelated. This is further discussed in Text A.

### 2.4.3    Total uncertainty: Monte Carlo simulations

The total uncertainty for each lavaka volume is estimated by running $10^4$ Monte Carlo simulations in which both the interpola-

tion and relative height error are considered. For the relative height error we draw random values from the normal distribution with mean = 0 and std = mean HEM of the lavaka. For the interpolation error we follow two different approaches depending on the presence or absence of a significant relationship between the mean interpolation error and lavaka area as detailed above (section 2.4.1). The result of these Monte Carlo simulations are $10^4$ volume estimates for each lavaka, from which the mean and its uncertainty (standard deviation) are calculated.

## 2.5 Establishing area-volume relationships

Establishing a relationship between lavaka area and volume enables to estimate lavaka volumes when only surface area information is available. Area-volume (typically landslides) and length-volume (typically gullies) relationships obey a power-law relationship $V = aA^b$, where the predicted volume $V$ for a given area $A$ depends on the scaling exponent $a$ and intercept $b$ (Larsen et al., 2010; Frankl et al., 2013). A linear relationship is typically fitted on the log-transformed data in order to obtain equally distributed residual errors, resulting in a more robust fit: $log(V) = a + blog(A)$ (e.g. Guzzetti et al., 2009; Crawford, 1991). As lavaka typically have a specific inverse-teardrop shape and both lengthen and widen when they grow (Wells et al., 1991), we use lavaka area instead of length as a size measure. We have therefore established the relationship between lavaka area and volume by fitting a linear least-squares regression through the log-transformed data (base 10 log). The volumetric uncertainties are propagated into the area-volume relationship by fitting the linear relationship for all the $10^4$ volume estimates from the Monte Carlo simulation and calculating the mean and std of the fitted $a$ and $b$ coefficients. These uncertainties are plotted as the 95% confidence intervals and represent the expected variation in the *mean* estimated volume given a specific area. They do not represent the range in which the next *individual* volume estimate would fall given a specific interval (i.e. the prediction interval). When back-transforming the coefficients of the fitted linear relationship to a power-function a systematic statistical bias enters. This is accounted for by adding a bias-correction factor which depends on the variance $\sigma^2$ (Ferguson, 1986; Crawford, 1991): $V = exp(a + 2.65\sigma^2)A^b$. This correction assumes that the residual errors of the fitted linear relationship are normally distributed with a mean of zero and variance $\sigma^2$. The normal distribution of the residual errors was tested using a Shapiro Wilk test (Shapiro and Wilk, 1965).

## 2.6 Lavaka volume growth and mobilization rates

From the established back-transformed area-volume relationship lavaka volumes could be calculated from the surface areas of lavaka in 1949, 1969 and the 2010s, enabling the derivation of volumetric growth and mobilization rates. The volumetric growth rate ($VGR$ [m$^3$ yr$^{-1}$]) for each lavaka was calculated as the change in volume ($dV$ [m$^3$]) over a given time period ($dt$ [yr]):

$$VGR = \frac{dV}{dt} = \frac{V_i - V_j}{t_i - t_j} \tag{1}$$

where $i$ indicates the most recent and $j$ the oldest observation moment.

Lavaka mobilization rates ($LMR$ [t ha$^{-1}$ yr$^{-1}$]) give the amount of sediment that has been mobilized over a given period and area. $LMR$ were calculated for each study area and are here expressed in tonne per hectare per year. To obtain $LMR$, lavaka volumes were converted to mass using a dry bulk density ($\rho$) of 1.2 t m$^{-3}$ (Montgomery, 2007). The sum of the lavaka masses of each study area was then divided by the length of the observation period and the surface area of the study area ($A$ [ha]) :

$$LMR = \frac{\sum_{k=1}^{N}(V_i\rho - V_j\rho)}{(t_i - t_j)A} \tag{2}$$

with $N$ the number of lavaka in each study area.

Uncertainties on the calculated $VGR$ and $LMR$ were estimated by taking into account the uncertainties on the fitted $a$ and $b$ coefficients of the applied area-volume relationship. This was done by running $10^4$ Monte Carlo simulations with different $a$ and $b$ values. These values were randomly drawn from the normal distribution of their mean and standard deviation. Gaussian copulas were used to take into account the dependence of both coefficients (Frees and Valdez, 1998). From all $10^4$ runs the mean and standard deviation were calculated. All the calculated volumes, volumetric growth rates and mobilization rates are available at: https://doi.org/10.5281/zenodo.5768418.

## 3   Results

### 3.1   Interpolation methods and uncertainty

In order to select the best interpolation method and to quantify the interpolation error, 50 fictive lavaka polygons were placed on intact hillslopes and were interpolated by using five different interpolation methods. The resulting height differences then give the interpolation error (Fig. C2 and Fig. C3). When considering the results for the full dataset (all individual pixels, Table 1, Fig. C5) three main observations can be made. First, regularized spline interpolation has the smallest spread (std, min and max values) and lowest MAE and RMSE for all DEMs. The mean and median error are also lowest when using regularized spline interpolation for the UAV-SfM (-1.75 m and -1.47 m) and Copernicus DEM (-0.89 m and -0.65 m). However, for the TanDEM-X DEM, regularized spline interpolation results in slightly higher mean and median errors when compared to TIN interpolation(-1.76 and -1.38 m vs. -1.62 and -1.17 m). Second, the negative mean and median interpolation errors indicate that the interpolated surface is generally lower than the real surface. The interpolated pre-erosion surface is thus on average underestimated by -0.89 to -1.76 m, which will also result in a corresponding underestimation of the volume if this error is not accounted for. Third, all error metrics are highest for the highest resolution DEM and decrease with decreasing resolution (e.g. RMSE decreases from 3.05 m to 2.97 m and 2.35 m for the UAV-SfM, TanDEM-X and Copernicus DEM, respectively). Based on these results we conclude that the regularized spline method yields overall the best results in our landscape setting. We therefore apply this interpolation method to estimate the lavaka volumes.

**Table 1. Interpolation error metrics**. Minimum (min), maximum( max), mean, median, mean absolute error (MAE), root mean squared error (RMSE) and standard deviation (std) of the elevation differences between the interpolated and DEM surface for the 50 fictive lavaka polygons considering all pixels. The interpolation method yielding the lowest error is indicated in bold for each DEM and error metric.

| | UAV-SfM (0.20 m) | | | | | TanDEM-X (12 m) | | | | | Copernicus (30 m) | | | | |
|---|---|---|---|---|---|---|---|---|---|---|---|---|---|---|---|
| | Linear | TIN | Spline bilinear | Spline bicubic | Spline regularized | Linear | TIN | Spline bilinear | Spline bicubic | Spline regularized | Linear | TIN | Spline bilinear | Spline bicubic | Spline regularized |
| Min | -25.08 | -25.10 | -39.91 | -34.91 | **-21.74** | -18.33 | -18.16 | -16.47 | -16.21 | **-11.87** | -17.72 | -18.45 | -17.96 | -17.39 | **-9.07** |
| Max | 15.86 | 14.12 | 18.11 | 34.34 | **12.88** | **6.37** | 7.18 | 8.77 | 8.52 | 6.89 | 8.32 | 10.25 | 8.58 | 7.99 | **6.56** |
| Mean | -2.29 | -1.83 | -2.63 | -2.66 | **-1.75** | -1.93 | **-1.62** | -2.63 | -2.58 | -1.76 | -1.81 | -1.59 | -3.12 | -2.97 | **-0.89** |
| Median | -1.92 | -1.32 | -2.19 | -2.21 | **-1.47** | -1.40 | **-1.17** | -2.16 | -2.11 | -1.38 | -1.13 | -1.08 | -2.47 | -2.32 | **-0.65** |
| MAE | 2.94 | 2.53 | 3.28 | 3.24 | **2.21** | 2.41 | 2.17 | 3.15 | 3.09 | **2.13** | 2.50 | 3.29 | 3.69 | 3.48 | **1.70** |
| RMSE | 4.21 | 3.90 | 4.66 | 4.41 | **3.05** | 3.48 | 3.18 | 4.22 | 4.14 | **2.97** | 3.72 | 4.51 | 5.05 | 4.81 | **2.35** |
| Std | 3.53 | 3.45 | 3.84 | 3.51 | **2.50** | 2.90 | 2.73 | 3.30 | 3.23 | **2.39** | 3.26 | 4.22 | 3.97 | 3.79 | **2.17** |

Next, it was verified if a significant relationship exists between the mean elevation difference of the 50 fictive lavaka polygons and their area (Fig C6). Pearson correlation coefficients ($\rho$) indicate that a significant negative relationship is present for the UAV-SfM ($\rho$ = -0.53, p = 8.14e-5) and TanDEM-X DEM ($\rho$ = -0.48, p = 1.53e-3), which is absent for the Copernicus DEM ($\rho$ = -0.10, p = 0.59). Therefore the mean ($\pm$ std) elevation difference of -0.89 $\pm$ 2.17 m is used to incorporate the interpolation error for the Copernicus DEM in the Monte Carlo simulations (Table 1). For the UAV-SfM and TanDEM-X DEM the fitted linear relationship between the mean elevation difference and lavaka area and the corresponding uncertainties on the fitted coefficients are used to estimate the interpolation error in the Monte Carlo simulations (Fig. C6).

UAV-SfM: *Interpolation Error* $= -0.22 \pm 0.26 - 1.10e-4 \pm 2.55e-5A$ (3)

TanDEM-X: *Interpolation Error* $= -0.34 \pm 0.32 - 9.95e-5 \pm 2.92e-5A$ (4)

## 3.2 Lavaka volumes

A direct pairwise comparison of the lavaka volumes obtained from the 0.2 m resolution UAV-SfM, 12 m resolution TanDEM-X and 30 m Copernicus DEM indicates that Copernicus generally results in a large volume underestimation (Fig. 3(a)). TanDEM-X, on the other hand, results in fairly similar volume-estimates as obtained by the UAV-SfM DEM, especially for the larger lavaka (> ca. $10^4$ $m^3$, Fig. 3(a)). While the absolute volume difference between UAV-SfM, TanDEM-X and Copernicus increases with increasing lavaka volume (Fig. 3(b)), a different picture emerges when looking at the relative volume differences (Fig. 3(c)). The relative difference is largest for the smallest lavaka and decreases when lavaka become bigger. Both absolute and relative volume differences are largest for Copernicus, with strong volume underestimations for the smallest lavaka and relative underestimations generally remaining above 50% for the larger features. For TanDEM-X large relative differences also occur for the smallest lavaka, however, they decrease to less than 20% for the largest features (Fig. 3(c)).

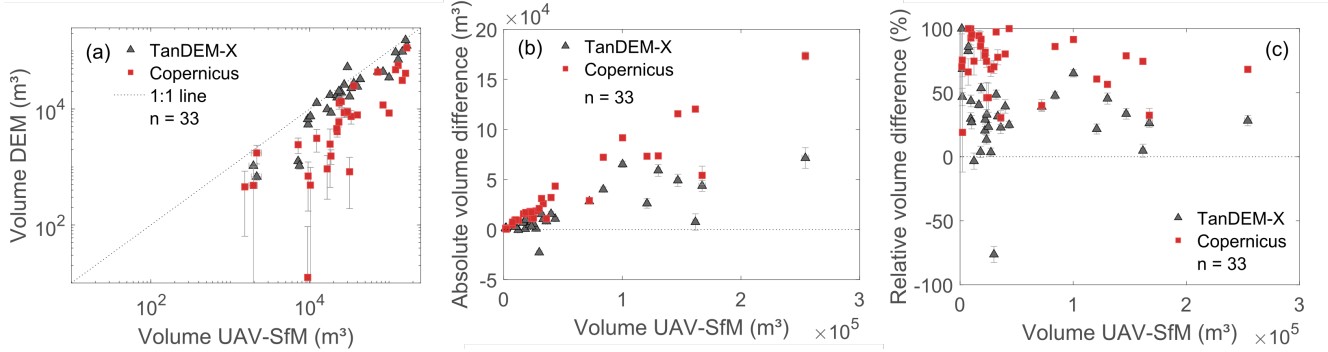

**Figure 3. Pairwise volume comparison** (a) Direct volume comparison between the lavaka volume obtained from the 0.2 m resolution UAV-SfM DEM and 12 and 30 m resolution TanDEM-X and Copernicus DEMs. The black line indicates the 1:1 line. Values are plotted on log-log scale. (b) Absolute and (c) relative volume difference with the UAV-SfM DEM. The dotted horizontal line indicates the zero-difference level. Grey error bars are the standard deviations of the mean calculated volumes representing the total uncertainty (interpolation and relative DEM uncertainty).

## 3.3 Area-volume relationships

Area-volume relationships were established using the log-transformed data. Both the relationships resulting from the pairwise and full dataset comparison clearly deviate for the different DEMs, where the smallest lavaka volumes are generally underestimated by TanDEM-X and Copernicus (Fig. 4). This results in a lower intercept and corresponding higher scaling exponent for the TanDEM-X and Copernicus relationships, which are significantly different (outside of the 95% confidence interval) from the UAV-SfM relationship. This difference with the UAV-SfM relationship increases with decreasing DEM resolution and is largest for the Copernicus DEM (Fig. 4). When considering the full dataset, large volume underestimations for the smallest lavaka are clearly apparent for volumes determined from the TanDEM-X and especially the Copernicus DEM (Fig. 4(b)). These are caused by negative volumes that were calculated over (a fraction) of the lavaka area. For larger lavaka areas and volumes, this discrepancy with the UAV-SfM DEM disappears, suggesting that TanDEM-X and Copernicus are capable of accurately assessing lavaka volumes for features that exceed a given size (based on visual inspection ca. $10^3$ m$^2$ and $10^4$ m$^2$ for the TanDEM-X and Copernicus DEM, respectively). However, because of the large volume underestimations for the smaller lavaka, the TanDEM-X and Copernicus area-volume relationships strongly deviate from the UAV-SfM relationship when incorporating all the data, where the fitted coefficients are not within the confidence interval of the fitted UAV-SfM coefficients.

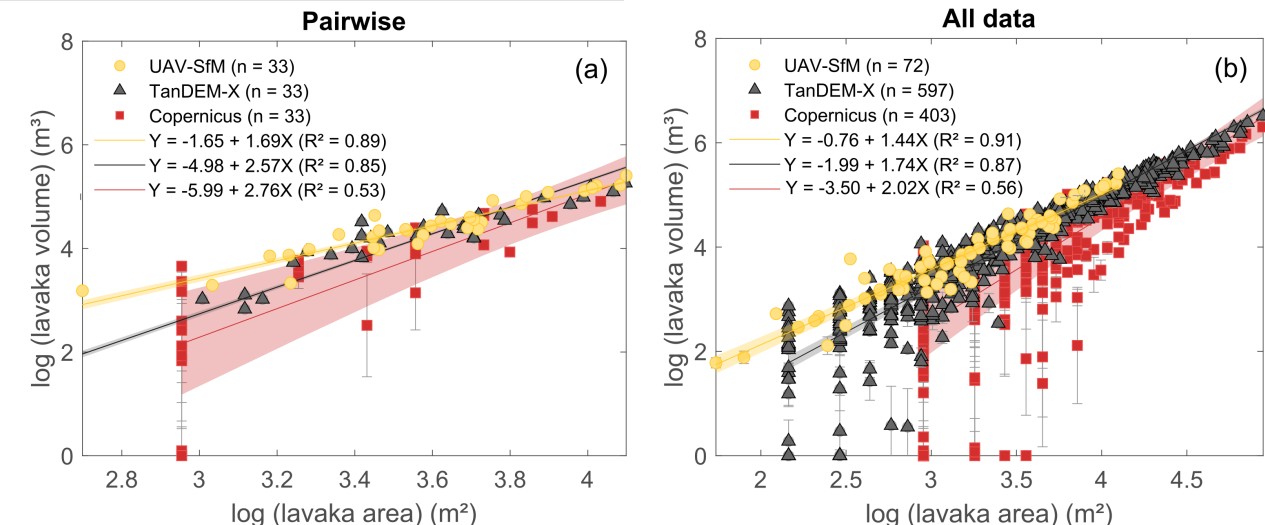

**Figure 4. Area-Volume relationships.** Fitted linear area-volume relationships between the log-transformed lavaka areas and volumes for (a) the pairwise dataset and (b) the full datasets containing all lavaka volumes for all three DEMs. Grey error bars are the standard deviations of the mean calculated volumes representing the total uncertainty (interpolation and relative DEM uncertainty). Shaded bands indicate the 95% confidence intervals of the fitted relationships where the volumetric uncertainties are propagated through Monte Carlo simulations.

While it is clear that the TanDEM-X, and especially the Copernicus, DEM are too coarse to accurately predict lavaka volumes for the smallest features, this issue seems to disappear for the larger features. Therefore, we tried to identify the point below which the analysis based on TanDEM-X and Copernicus suffers from errors in volume reconstruction as evidenced by negative

volume pixels. This breakpoint was identified as the point where the RMSE from the 1:1 $Vpos$-$Vtot$ line becomes smaller than 1%. This breakpoint is for the TanDEM-X DEM located at a positive volume of ca. $2500 \pm 1500$ m$^3$ and corresponding surface area of ca. $800 \pm 250$ m$^2$ or $6 \pm 2$ pixels. For the Copernicus DEM this point is located at a positive volume of ca. 120 $000 \pm 45\,000$ m$^3$, corresponding to a lavaka surface area of $13\,000 \pm 3500$ m$^2$ or $14 \pm 5$ pixels (Fig. 5(a)).

In a next step, we established a new area-volume relationship for the TanDEM-X and Copernicus data containing only lavaka

volumes larger than their identified breakpoints. Back-transforming the fitted linear log-transformed area-volume relationships results in the following power-law lavaka area-volume relationships for the UAV-SfM, TanDEM-X and Copernicus DEM, where the standard deviation of the back-transformed bias-corrected $a$ and $b$ coefficients are indicated:

UAV-SfM: $V = 0.55 \pm 0.08 A^{1.44 \pm 0.04}$                                                      (5)

TanDEM-X: $V = 0.43 \pm 0.03 A^{1.48 \pm 0.02}$                                                  (6)

Copernicus: $V = 1.58 \pm 0.19 A^{1.13 \pm 0.03}$                                               (7)

For TanDEM-X this results in a close match with the fitted relationship based on the UAV-SfM data, with fitted regression coefficients for TanDEM-X that are within uncertainty of the UAV-SfM coefficients (Fig. 5(b), Eq. (5) and Eq. (6)). While fitting the relationship only through the data points above the breakpoint results in an area-volume relationship within uncertainty of

the ground truth UAV-SfM relationship for the TanDEM-X DEM this is not the case for the Copernicus DEM. Even when keeping only the data above the identified breakpoint, the fitted area-volume relationship for Copernicus still largely deviates from the UAV-SfM relationship, with a lower scaling coefficient and higher intercept (Fig. 5(b), Eq. (5) and Eq. (7)). Given the large discrepancy between the UAV-SfM and Copernicus relationship the latter will not be used for further calculations of the volumetric growth and mobilization rates.

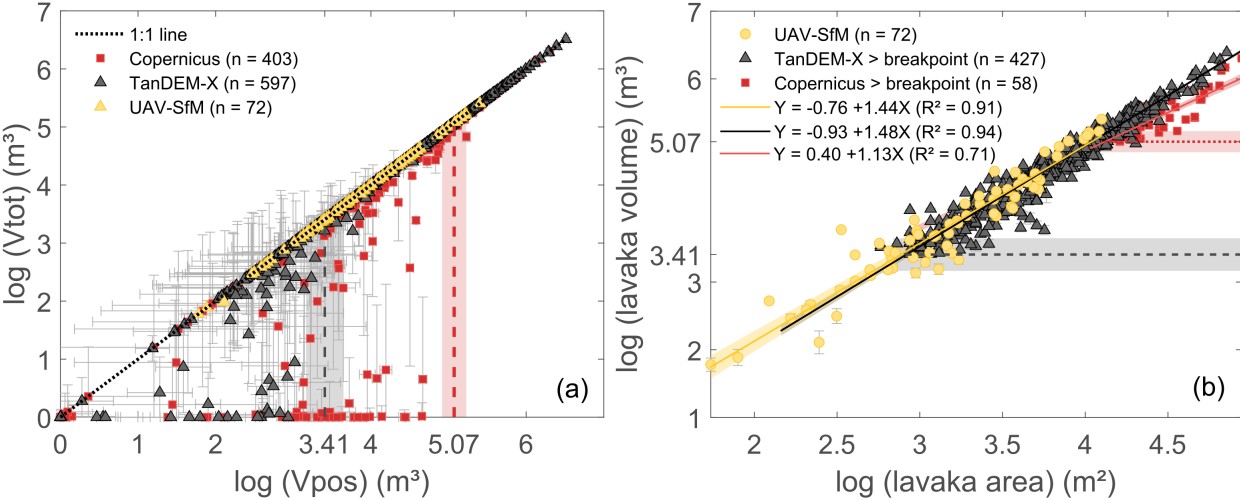

**Figure 5. Breakpoint analysis and final area-volume relationships.** (a) The breakpoint is identified as the point where the RMSE from the 1:1 line of the log-transformed positive ($Vpos$) and total ($Vtot$) volumes is smaller than 1%. The identified breakpoint is located at $\log(Vpos)$ = $3.41 \pm 0.24$ m$^3$ for TanDEM-X and at $5.07 \pm 0.16$ m$^3$ for Copernicus. (b) Linear area-volume relationships fitted through the log-transformed lavaka area and volume data for the full UAV-SfM dataset and for the TanDEM-X and Copernicus volumes that exceed the identified breakpoints ($\log$(Vpos)>$3.41 \pm 0.24$ m$^3$ for TanDEM-X and > $5.07 \pm 0.16$ m$^3$ for Copernicus). Shaded areas indicate the 95% confidence intervals of the fitted relationships and the standard deviation of the breakpoints. Grey error bars are the standard deviations of the mean calculated volumes representing the total uncertainty (interpolation and relative DEM uncertainty)

### 3.4 Lavaka volumetric growth and mobilization rates: 1949-2010s

By applying the established area-volume relationships to the historical (1949 for SA1-5 and 1969 for SA6) and current (2010s) lavaka areas (Table B1), volumetric growth rates ($VGR$ in m$^3$ yr$^{-1}$) could be estimated (Eq. (1)). When using the UAV-SfM relationship (Eq. (5)) a mean and median growth rate of $1149 \pm 275$ m$^3$ yr$^{-1}$ and $320 \pm 56$ m$^3$ yr$^{-1}$ are obtained, respectively. When applying the TanDEM-X relationship (Eq. (6)) these values are ca. 15% higher: $1341 \pm 137$ and $354 \pm 26$ m$^3$ yr$^{-1}$ for the mean and median, respectively. This deviation of 15% is, however, still within uncertainty of the estimates from the UAV-SfM DEM which have an uncertainty range of 24% resulting from the larger uncertainties on the fitted coefficients. This indicates that small variations in the established coefficients can lead to relatively large differences in estimated volumetric growth and mobilization rates.

Volumetric growth rates can be converted to lavaka mobilization rates ($LMR$ in t ha$^{-1}$ yr$^{-1}$) when the bulk density and size of the study areas are taken into account (Eq. (2)). Lavaka mobilization rates as derived from the UAV-SfM relationships range between $18 \pm 3$ and $311 \pm 82$ t ha$^{-1}$ yr$^{-1}$ in our six study areas (Table 2). $LMR$ are again estimated to be ca. 13 to 21% higher when applying the TanDEM-X relationship (Table 2), resulting in $LMR$ between $20 \pm 2$ and $377 \pm 42$ t ha$^{-1}$ yr$^{-1}$, which is within uncertainty of the UAV-SfM estimates.

**Table 2. Lavaka mobilization rates 1949-2010s**. Lavaka mobilization rates (in t ha$^{-1}$ yr$^{-1}$ (tonne per hectare per year)) obtained by applying the area-volume relationships from the UAV-SfM (Eq. (5)) and TanDEM-X (Eq. (6)) DEM to the lavaka areas for the longest time period available: 1949-2010s for SA1-5 and 1969-2010s for SA6. Reported values give the median and standard deviation from the $10^4$ Monte Carlo simulations where the uncertainties on the fitted $a$ and $b$ coefficients of the area-volume relationships are accounted for.

| | Mobilization rate UAV-SfM (t ha$^{-1}$ yr$^{-1}$) | Mobilization rate TanDEM-X (t ha$^{-1}$ yr$^{-1}$) | Difference UAV-SfM - TanDEM-X (%) |
|---|---|---|---|
| SA1 | $311 \pm 82$ | $377 \pm 42$ | -21 |
| SA2 | $111 \pm 27$ | $131 \pm 13$ | -18 |
| SA3 | $55 \pm 12$ | $64 \pm 6$ | -16 |
| SA4 | $148 \pm 34$ | $173 \pm 16$ | -17 |
| SA5 | $27 \pm 6$ | $31 \pm 3$ | -14 |
| SA6 | $18 \pm 3$ | $20 \pm 2$ | -13 |
| All SA's | $108 \pm 26$ | $128 \pm 13$ | -19 |

Lavaka mobilization rates seem to be positively correlated with the mean surface area of lavaka in the study area (Fig. 6(b)). This can be explained by the positive correlation between lavaka area and volumetric growth rate ($r$ = 0.27, p = 1e-10, Fig. 6(a)): larger lavaka mobilize more material. $LMR$ also increase with increasing lavaka density (Fig. 6(c)), which is logical but is also partially explained by the positive correlation between lavaka density and mean lavaka surface area (Fig. 6(d)). The main variations in $LMR$ between our six study areas thus seem to depend mainly on the lavaka density and area distribution. While lavaka presence has been linked to seismic activity (Cox et al., 2010) and is typically associated with slopes ranging between 25 to 30° in the Lake Alaotra region (Voarintsoa et al., 2012), differences in surface growth rates could only be poorly linked to other environmental factors: the combined effects of the percentage bare surface area, distance to stream and distance to drainage divide could only explain 18% of the variation in lavaka growth rates (Brosens et al., 2022).

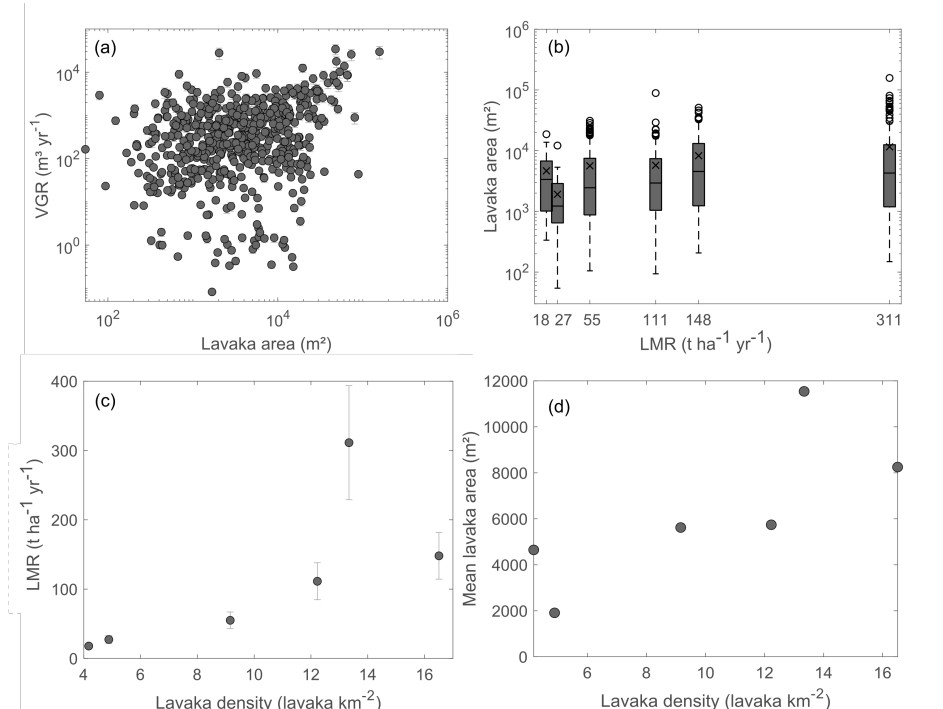

**Figure 6. Variations in volumetric growth rates and lavaka mobilization rates.** (a) Lavaka volumetric growth rates ($VGR$) are positively related with lavaka area (spearman correlation coefficient $r = 0.27$, p = 1e-10). (b) Lavaka mobilization rates ($LMR$) are higher for study areas with larger lavaka. Mean lavaka areas are indicated by the cross in the boxplot. Higher lavaka mobilization rates are linked to higher lavaka densities (c), which are also positively correlated with lavaka area (d). n indicates the number of observations and the error bars indicate the standard deviation of the mean $LMR$ as obtained from the Monte Carlo simulations taking into account the uncertainties on the fitted $a$ and $b$ coefficients.

In order to further evaluate the possible impact of fitting the TanDEM-X relationship on the larger features only (> 800 $\pm$ 250 m$^2$), we quantified the share of the total mobilized sediment that is provided by lavaka smaller than 800 $\pm$ 250 m$^2$. From the relative cumulative sediment mobilization curves it is apparent that larger lavaka contribute most of the mobilized sediment (Fig. C7). Lavaka that are smaller than the identified threshold contribute 0.2% of the total mobilized sediment in the study areas with the largest lavaka and up to 2.6% in the regions with smaller lavaka (Fig. C7). This indicates that the share of smaller lavaka to the total amount of sediment that is mobilized is generally low in our study areas, therefore reducing the risk of erroneous estimates in the case where these smaller lavaka could not be used to establish the TanDEM-X based A-V relationships.

## 4 Discussion

### 4.1 Interpolation methods and DEM uncertainties

The best interpolation results were obtained when using a regularized spline with tension algorithm (Table 1 and Fig. C5). Bergonse and Reis (2015) concluded that spline interpolation methods result in smaller errors compared to linear methods as they are better adjusted to a gully geomorphic context by allowing curved surfaces in no data areas, which is not the case for linear interpolation methods. This general conclusion is, however, not necessarily confirmed by our results: while the lowest errors were obtained for the regularized spline with tension algorithm, both other spline algorithms (bilinear and bicubic) performed worse than the linear and TIN interpolation algorithms (Table 1). This might be related to the fact that spline methods are parameterized, where we used the default settings without optimization as an in-depth comparison of different interpolation methods was out of scope for this study. This indicates that parameterized interpolation methods should be applied with care and do not necessary lead to the best results.

The UAV-SfM DEM was used as the ground-truth reference in this study. However, like other DEMs, it is constructed from an airborne perspective, where vertical morphologies such as overhanging walls, undercutting or piping features are hidden from the observation point (Frankl et al., 2015). The impact on the estimated volumes should, however, be minimal, as earlier reported volumetric differences are only ca. 2.5% (Frankl et al., 2015). Volumetric gully measurements from photogrammetric techniques are furthermore reported to suffer from sun- and sight-shadowing, which is especially the case for narrower gullies and might result in inaccuracies in the DEM (Giménez et al., 2009).

The main limitation of the UAV-SfM DEM is the presence of vegetation, making it a digital surface model (DSM) rather than a digital elevation model (DEM). The same is true for the TanDEM-X and Copernicus DEMs, where the relative impact of vegetation on the final elevation will be smaller due to their coarser resolution. The vegetation was not filtered out of any data-product because most of the land surface in the studied regions is covered with low grassland vegetation. Some trees or bushes are present in the landscape near the hillslope bottoms or inside of stabilizing lavaka (Fig. 1(e)-(f)). While the presence of vegetation at the hillslope bottoms might result in a slight overestimation of the interpolated surface, this effect has a minimal impact on the estimated lavaka volumes because at this location lavaka are typically at their narrowest (Wells et al., 1991) (Fig. 1).

A possible caveat when using a fish-eye camera for UAV image acquisition is vertical 'doming'. However, our flights were carried out with a slightly tilted camera and in a course-aligned way, resulting in oblique images with overlapping areas under a different angle, which is reported to reduce error propagation and doming (James and Robson, 2014). Possible vertical doming could only be verified visually in our case by inspecting the point cloud of flat surfaces in the study area, since no independent GNSS dataset is available. Visual inspection (Fig. C8) and reported vertical deviations less than 0.07 m by Zhang et al. (2019), who's set-up was adopted here, confirm that this effect is likely minimal.

## 4.2 Lavaka volumes and area-volume relationships from varying DEM resolutions

The coarser DEM resolution of TanDEM-X and Copernicus results in a systematic underestimation of lavaka volumes for all lavaka in the case of Copernicus and for smaller lavaka in the case of TanDEM-X upon direct comparison with the UAV-SfM volumes (Fig. 3(a)). A first factor that explains this observation is the dependence of the optimal DEM grid resolution on the inherent properties and scale of the geomorphic features under study (Tarolli, 2014; Hengl, 2006; Smith et al., 2019). Theoretically, the minimum size of a landform should be twice the resolution of the DEM, (Theobald, 1989; Frankl et al., 2013), corresponding to 0.16 m$^2$, 576 m$^2$ and 3600 m$^2$ for the UAV-SfM, TanDEM-X and Copernicus DEM, respectively. Comparing these theoretical minima with the identified breakpoints at $800 \pm 250$ m$^2$ for the TanDEM-X DEM and $1.3 \times 10^4 \pm 3500$ m$^2$ for the Copernicus DEM indicates that in practice the aerial DEM resolution has to be rather 2.4 to 3.8 times the landform size in order to accurately capture it.

While for the TanDEM-X DEM the volumes for features larger than the breakpoint closely match those obtained from the UAV-SfM DEM, this is not the case for the Copernicus DEM (Fig. 5(b)). This indicates that for the TanDEM-X DEM the largest volumetric errors are contained within the percentage negative volume, as the breakpoint corresponds to the point where the TanDEM-X volumes no longer deviate from the volumes obtained from the UAV-SfM DEM (Fig. 3(a)). Furthermore, this also resulted in an area-volume relationship for TanDEM-X that is within uncertainty of the UAV-SfM relationship (Fig. 5, Eq. (5) and Eq. (6)). A large deviation between the area-volume relationship obtained for the Copernicus DEM and UAV-SfM DEM remained, even when considering only the lavaka located above the breakpoint (Fig. 5(b)). For the Copernicus DEM the absence of negative volumes in the total volume estimate thus seems to be an insufficient measure to accurately estimate lavaka volumes. This might be related to a second factor that affects estimated volumes, which is the DEM smoothness. The smoothing effect of coarser resolution DEMs on landscape topographical representation is known to result in a reduced ability to capture more complex topography and geomorphic features (Thompson et al., 2001; Wechsler, 2007; Tarolli, 2014; Hengl, 2006). The underestimation of eroded volumes when coarser resolution DEMs are used was also reported by Claessens et al. (2005), who found that the highest landslide erosion and deposition volumes were estimated for the highest resolution DEM and systematically decreased when reducing the DEM resolution. This effect was attributed to the more detailed landscape representation for higher resolution DEMs.

Our results therefore indicate that for the TanDEM-X DEM our $Vpos$-$Vtot$ breakpoint method allows to exclude lavaka that suffer from evident volume reconstruction errors in an objective way. This method can furthermore be applied to regions where no sub-meter resolution DEMs are available for comparison, as the breakpoint can be determined from the difference between $Vtot$ and $Vpos$ using the coarser resolution DEM. However, our results also show that the 30 m resolution Copernicus DEM is too coarse and filters out too many topographic details to accurately calculate volumes of erosional features at the lavaka scale (100 - $10^5$ m$^2$) as large volume underestimations remain, even when considering only the largest features (Fig. 3(c) and Fig. 5(b)).

While coarser resolution DEMs result in lower volume estimates, volumetric growth and lavaka mobilization rates estimated from the TanDEM-X area-volume relationship are 13 to 21% higher than those obtained from the UAV-SfM area-volume

relationship (Table 2). From the area-volume graph (Fig. 5(b)) it can be seen that TanDEM-X slightly underestimates the volumes of lavaka located just above the breakpoint. This 'pulls' the linear regression line down for smaller lavaka, resulting in a slightly higher scaling coefficient for the TanDEM-X area-volume relationship ($1.48 \pm 0.02$) when compared to UAV-SfM ($1.44 \pm 0.04$). This results in lavaka volumes that will be slightly underestimated for the smaller features and overestimated for the larger features. As the largest features account for the majority of the mobilization rates (Fig. C7), the volumetric growth and mobilization rates for the TanDEM-X DEM will be overestimated. Further reducing the maximum RMSE below 1% for the breakpoint determination could resolve this issue but will set the minimum lavaka area higher, further reducing the number of observations.

Gully volumes are typically linked to gully length as most gullies mainly lengthen when they grow (Frankl et al., 2013; Vanmaercke et al., 2021). Lavaka, on the contrary, deepen, widen and lengthen when they grow, which is why we link lavaka volume with area instead of length. While this does not allow direct comparison with other relationships obtained for gullies, previous studies reported that length-volume relationships are region-specific (Frankl et al., 2013). Applying the observed relationship outside of the lake Alaotra region should therefore be done with care and might require validation. While the processes of landslide and lavaka erosion are entirely different, the obtained scaling coefficient $a$ of $1.44 \pm 0.04$ indicates that for a given area, lavaka volumes will be similar to those of deep landslides that typically have an $a$ between 1.3 and 1.6) (Larsen et al., 2010).

In a typical pattern of development lavaka start as raw patches that evolve to step-like headscarps, grow into deep inverse teardrop shaped gullies and finally become longer, broader, gentler and partly filled concavities when stabilizing (Wells et al., 1991). Upon stabilization lavaka will partially fill in, reducing the volume. Not all lavaka will stabilized at the same size, nor grow in the same way. This will likely be one of the main factors explaining the remaining 6 to 9% of variation in lavaka volume that cannot be explained by the area ($R^2$ = 0.94 and 0.91 for TanDEM-X and UAV-SfM, respectively) (Fig. 5).

### 4.3 Lavaka mobilization rates put into perspective

Our calculated lavaka mobilization rates from direct lavaka growth observations over the period 1949-2010s (ca. 70 years) range between $18 \pm 3$ and $311 \pm 82$ t ha[-1] yr [-1] for six ca. 10 km[2] study areas in the Lake Alaotra region with an overall average of $108 \pm 26$ t ha[-1] yr[-1] (Table 2). It should, however, be noted that our highest reported $LMRs$ of $311 \pm 82$ and $148 \pm 34$ t ha[-1] yr [-1] correspond to areas characterized by large lavaka and high lavaka densities (13 and 17 lavaka km[-2], Table B1, Fig. 6(b)). These lavaka densities are higher than the reported average of 6 lavaka km[-2] for the southern part of the Lake Alaotra catchment (Voarintsoa et al., 2012). We therefore argue that these highest values should be perceived as maximum rates, where the rates of 18-53 t ha[-1] yr [-1] obtained for regions with lower lavaka densities (SA 3, 5 and 6, Table B1) will be more representative for the wider Lake Alaotra region (Table 2).

Only limited local data is available that can be used to compare these estimates with. A sedimentation rate of 20 t ha[-1] yr[-1] was obtained by Mietton et al. (2006) for the dammed Bevava lake which is located in the southeastern part of the Lake Alaotra catchment over the period 1987-2005. Lake Bevava has a catchment area of 58 km[2] with a lavaka density of 8 lavaka km[-2] (Mietton et al., 2006). The reported recent lake sedimentation rate of 20 t ha[-1] yr[-1] is less than half of our calculated lavaka

mobilization rate of $53 \pm 19$ t ha$^{-1}$ yr$^{-1}$ for SA3 which has a comparable lavaka density of 9 lavaka km$^{-2}$ (Fig. 6(c), Table B1 and Table 2). While both estimates are the same order of magnitude, this suggests that a considerable proportion (more than 50%) of the mobilized sediment will likely be trapped close to the lavaka and not reach the rivers or lake.

Next to these recent short-term sedimentation rates, long-term catchment wide erosion rates obtained from [10]Be measurements have been reported for the central Malagasy highlands. These [10]Be erosion rates integrate over a timescale of thousands to hundreds of thousands of years and represent long-term averages. Reported long-term [10]Be erosion rates range from 0.16 to 0.54 t ha$^{-1}$ yr$^{-1}$ with the highest rates for the catchments with higher lavaka densities (max. 6 lavaka km$^{-2}$, Cox et al., 2009). Ideally these long-term rates are compared with current sediment yields or sedimentation data (Bartley et al., 2015; Vanacker et al., 2007), as a considerable fraction of the sediment likely never reaches the rivers or lakes. However, the offset of two orders of magnitude between long-term [10]Be erosion rates and current lavaka mobilization rates and lake Bevava sedimentation rates suggests that lavaka erosion has increased over recent time periods in the Lake Alaotra region. This was also concluded by Brosens et al. (2022), where a tenfold increase in floodplain sedimentation rates was observed over the past 1000 years, which was linked to a recent increase in lavaka activity brought about by increasing environmental pressure due to growing human and cattle populations (Joseph et al., 2021).

Globally reported volumetric gully erosion rates range between 0.0002 and 47 430 m$^3$ yr$^{-1}$, with mean and median values of 359 and 2.2 m$^3$ yr$^{-1}$ (Vanmaercke et al., 2016). Our mean and median estimated volumetric growth rates of $1149 \pm 275$ m$^3$ yr$^{-1}$ and $320 \pm 56$ m$^3$ yr$^{-1}$ are at least three times higher than these global averages, indicating that lavaka erosion in the lake Alaotra catchment is occurring at above average gully erosion rates. These reported volumetric gully growth rates correspond to global mean and median aerial gully growth rates of 3.1 and 131 m$^2$ yr$^{-1}$ (Vanmaercke et al., 2016), whereas the mean and median aerial lavaka growth rates for our lavaka dataset are 22 and 11 m$^2$ yr$^{-1}$ (Brosens et al., 2022). This indicates that while volumetric lavaka growth rates are higher than the global averages, their change in aerial extent is below average. This is caused by the specific morphology of lavaka, which are much deeper than average gullies with estimated mean and median depths for our dataset of 23 and 19 m based on the calculated volumes and areas, whereas this is only 2.1 and 1.3 m for the global dataset (Vanmaercke et al., 2016).

## 5    Conclusions

Lavaka volumes were estimated as the difference between the current and interpolated pre-erosion surface for three DEMs with different spatial resolutions: UAV-SfM (0.2 m), TanDEM-X (12 m) and Copernicus (30 m). Volumes estimated from TanDEM-X are similar to those obtained from the UAV-SfM DEM for the larger features. Using the Copernicus DEM results in strong volume underestimations, even for the largest features. This indicates that the Copernicus DEM is not suitable to estimate erosion volumes for geomorphic features at the lavaka-scale (100 - $10^5$ m$^2$), which is caused by a smoothing effect where complex topography and smaller geomorphic features cannot be accurately captured by the coarser resolution. TanDEM-X can be used for volume estimations, but shows a tendency to underestimate volumes for small lavaka. An area-volume relationship, necessary for large scale and past volume assessments, can be established using TanDEM-X or UAV-SfM data.

However, developing a robust relationship based on the TanDEM-X data requires that observations for which the relative reconstruction error is too large are eliminated from the dataset. Here, we proposed and tested a method to identify a cut-off point below which volume estimations are clearly affected by processing errors as evidenced by negative volume estimates. This breakpoint is located at a lavaka volume of ca. $2500\pm1500$ m$^3$, corresponding to an area of ca. $800\pm250$ m$^2$. The proposed objective filtering to eliminate erroneous volumes in the TanDEM-X estimates resulted in deviations in lavaka growth rates and mobilization rates that are ca. 13 to 21% higher compared to the respective UAV-SfM estimates and fall within the uncertainty boundaries of the latter. Our results thus indicate that the TanDEM-X DEM can be used to establish accurate area-volume relationships for erosional features at the lavaka-scale when applying the breakpoint method. As this method does not depend on direct comparison with higher resolution DEMs, it can be applied to regions where only TanDEM-X is available. Over the period 1949-2010s a mean and median lavaka volumetric growth rate of $1149\pm275$ and $320\pm56$ m$^3$ yr$^{-1}$ and lavaka mobilization rates varying between $18\pm3$ and $311\pm82$ t ha$^{-1}$ yr$^{-1}$ were obtained. While our highest lavaka mobilization rates are likely limited to the most lavaka dense regions, our lower estimates are consistent with reservoir siltation rates, placing the current average lavaka mobilization rate for the Lake Alaotra region at 20-50 t ha$^{-1}$ yr$^{-1}$. These rates are furthermore two orders of magnitude higher than earlier reported long-term erosion rates, suggesting a recent increase in lavaka erosion intensity in the Lake Alaotra region.

*Code and data availability.* All 12 m TanDEM-X files (DEM and auxiliary files) were provided by DLR under a scientific use user license. The Copernicus DEM is freely available through ESA's data portal. WorldView-2 imagery was available as a baselayer in ArcMap software from Esri. All data from the lavaka dataset are disposed at https://doi.org/10.6084/m9.figshare.c.5236322.v1. The PyQGIS code used to extract the lavaka volumes, an example dataset excel table with all the calculated volumes and $VGR$ are deposited at https://doi.org/10.5281/zenodo.5768418

## Appendix A: MoransI

We have verified the assumption of perfect autorrelation for the HEM pixels of a lavaka by calculating Moran's I (queen). For the TanDEM-X DEM the HEM-pixels of a lavaka have a mean Moran's I of 0.65 with a median of 0.70. For the Copernicus DEM these values are lower and equal to 0.31 and 0.38 for the mean and median, respectively (Fig. C4(b)). These results indicate that using the same HEM value for a full lavaka will result in a maximum estimate of the uncertainty, as in reality the pixels are not perfectly autocorrelated.

**Appendix B:  Tables**

**Table B1. Study area characteristics and imagery availability.** The availability of the 1949 and 1969 aerial images is indicated by a cross and the satellite acquisition dates are reported. For each study area its surface area, number of lavaka and resulting lavaka density are indicated.

| Study area | Surface [km$^2$] | Aerial picture 1949 | Aerial picture 1969 | Satellite aquisition date | Satellite source | Number of lavaka | Lavaka density [lavaka km$^{-2}$] |
|---|---|---|---|---|---|---|---|
| 1 | 11.47 | X | X | 27/05/2018 | WorldView-2 | 153 | 13 |
| 2 | 10.47 | X | X | 12/09/2011 | WorldView-2 | 128 | 12 |
| 3 | 15.29 | X | X | 10/07/2016 | WorldView-2 | 140 | 9 |
| 4 | 10.48 | X |   | 29/05/2018 | WorldView-2 | 173 | 17 |
| 5 | 11.27 | X | X | 27/05/2018 | WorldView-2 | 55 | 5 |
| 6 | 11.98 |   | X | 27/05/2018 | WorldView-2 | 50 | 4 |

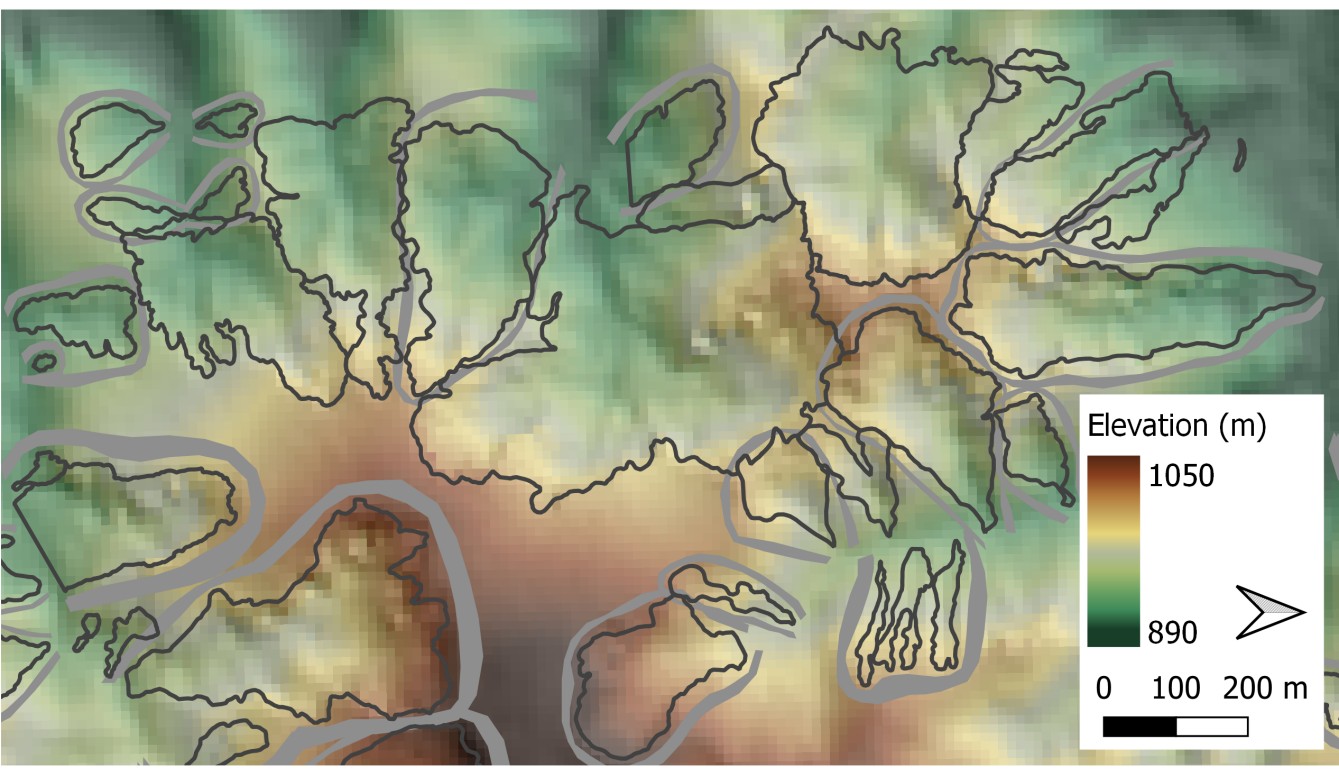

**Figure C1. Example of near absence original surface topography**. Example from study area 1 illustrating the near absence of the original surface topography (especially in the western part of the area) due to the dense presence of lavaka (grey outlines). Grey horseshoshaped polygons indicate the areas unaffected by gully erosion. These could not be derived for all lavaka, and sometimes envelope multiple lavaka that are located next to each other. Displayed elevations are from the TanDEM-X DEM with hillshade (Krieger et al., 2007).

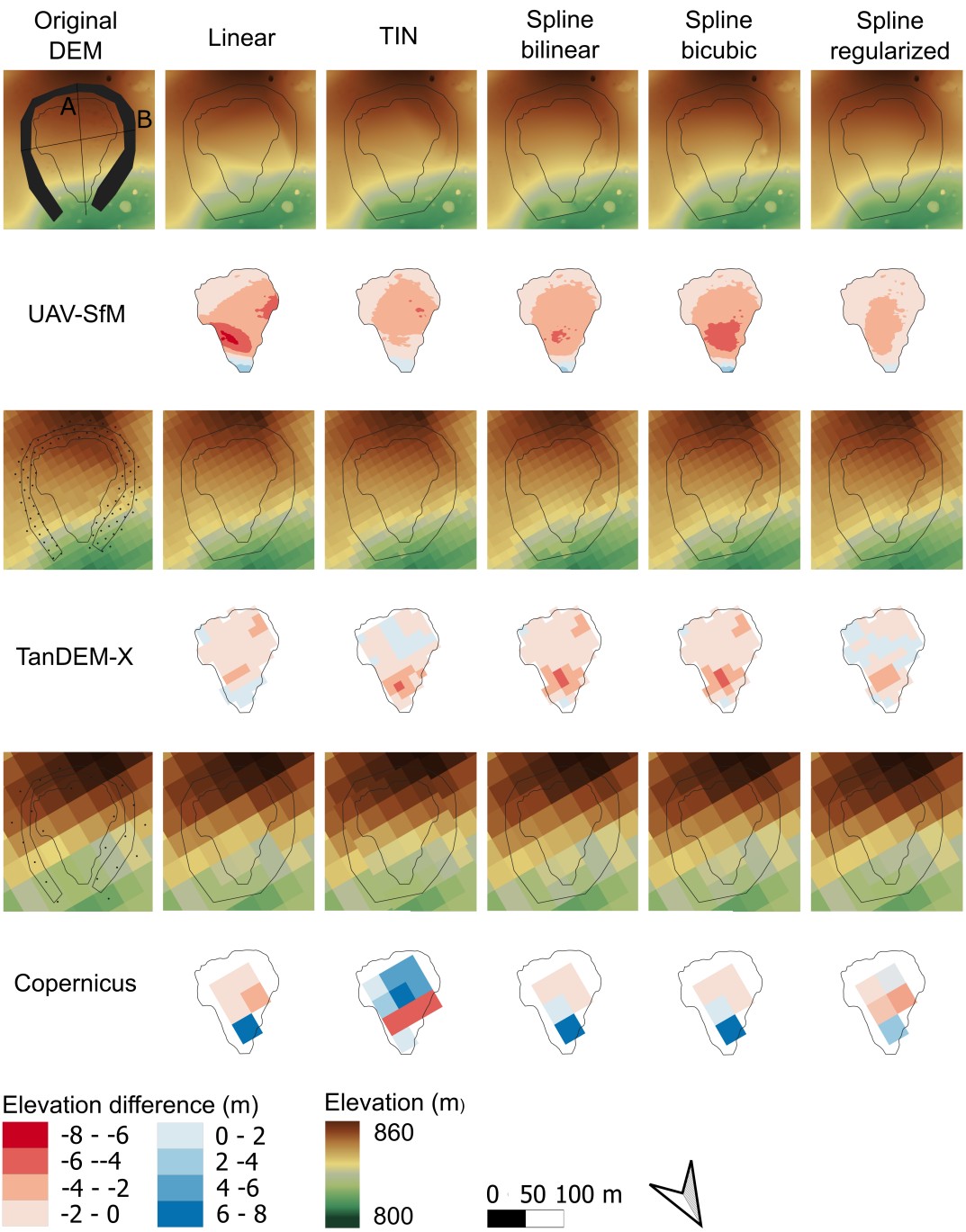

**Figure C2. Interpolation error workflow.** The interpolation error was assessed by placing 50 lavaka polygons and corresponding horseshoe-shaped polygons on intact hillslopes. The difference between the interpolated surface and the DEM gives the interpolation error. This is done for all three DEMs (UAV-SfM (0.2 m), TanDEM-X (12 m) and Copernicus (30 m)) and by using five different interpolation methods (Linear, TIN, Spline bilinear, Spline bicubic and Spline regularized).

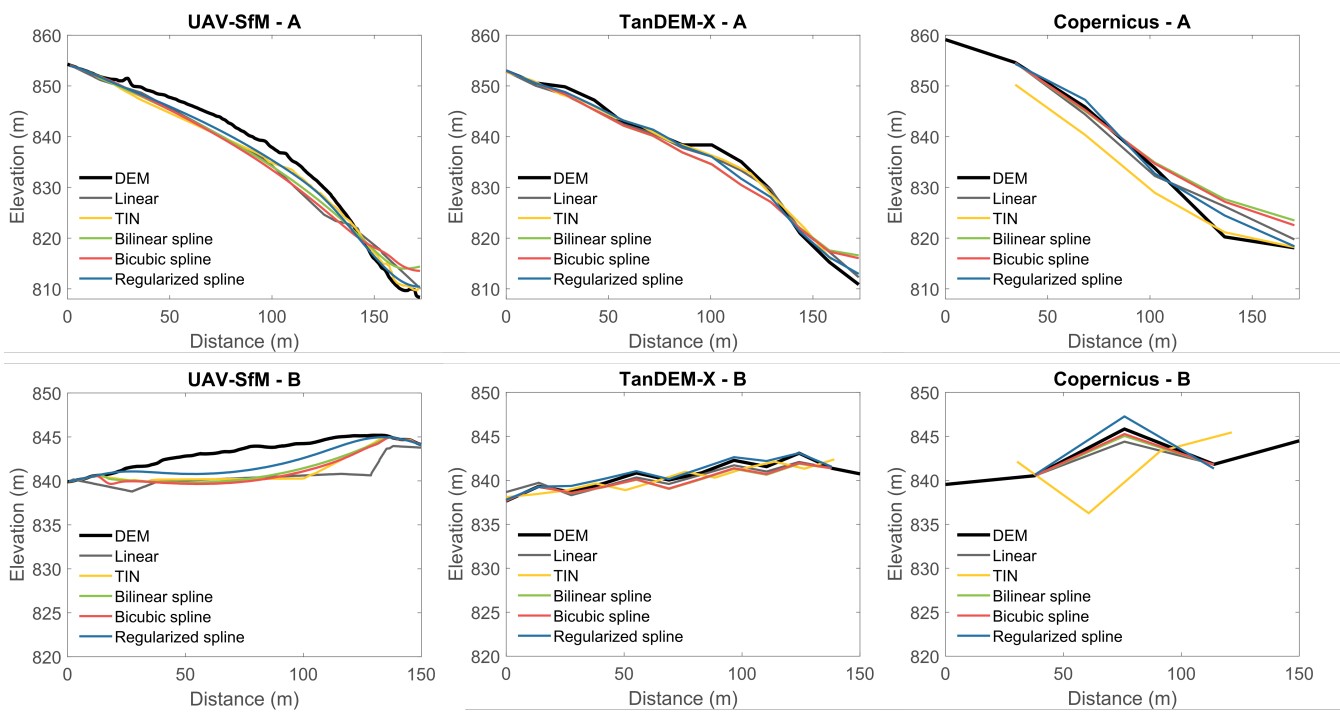

**Figure C3. Interpolation error cross sections.** Cross sections for transect A and B as indicated in Fig. C2 for each of the three DEMs (UAV-SfM (0.2 m), TanDEM-X (12 m) and Copernicus (30 m)) and five interpolation methods (Linear, TIN, Spline bilinear, Spline bicubic and Spline regularized).

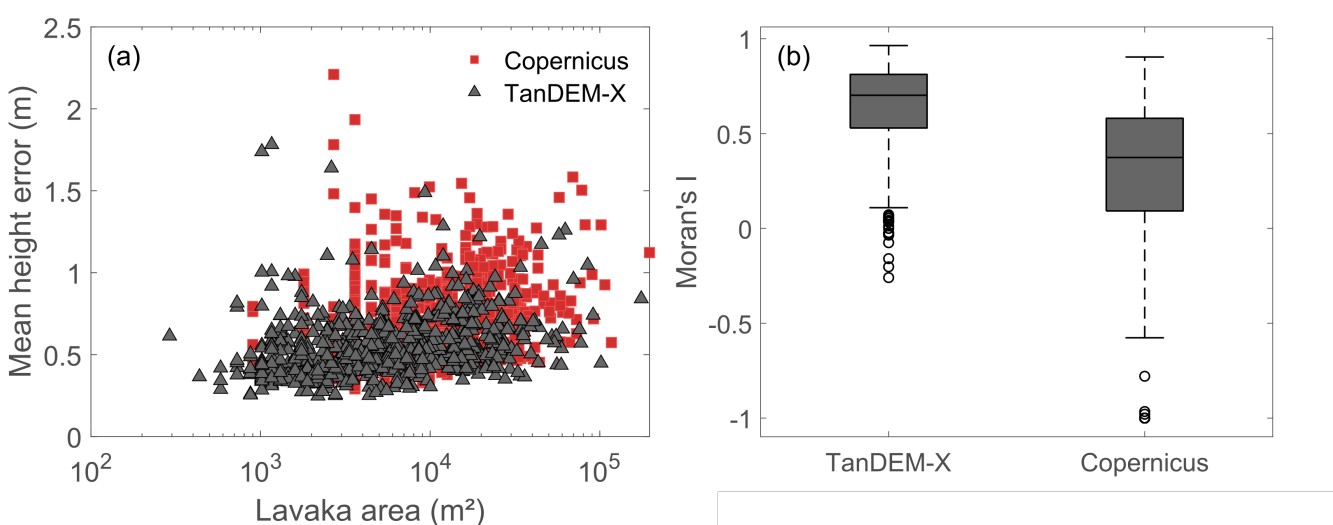

**Figure C4. Relative height error.** (a) The relative height error is estimated based on the Height Error Mask (HEM) of the TanDEM-X and Copernicus DEMs, which represent the random elevation error in the form of the standard deviation. A positive correlation between the mean height error of a lavaka and its surface area is observed. (b) The autocorrelation of the HEM-values is calculated for each lavaka by means of the Moran I (queen) for borth the TanDEM-X and Copernicus DEM. A value of 1 represent a perfect positive autocorrelation, a value of zero a random distribution and a value of -1 indicates negative autocorrelation.

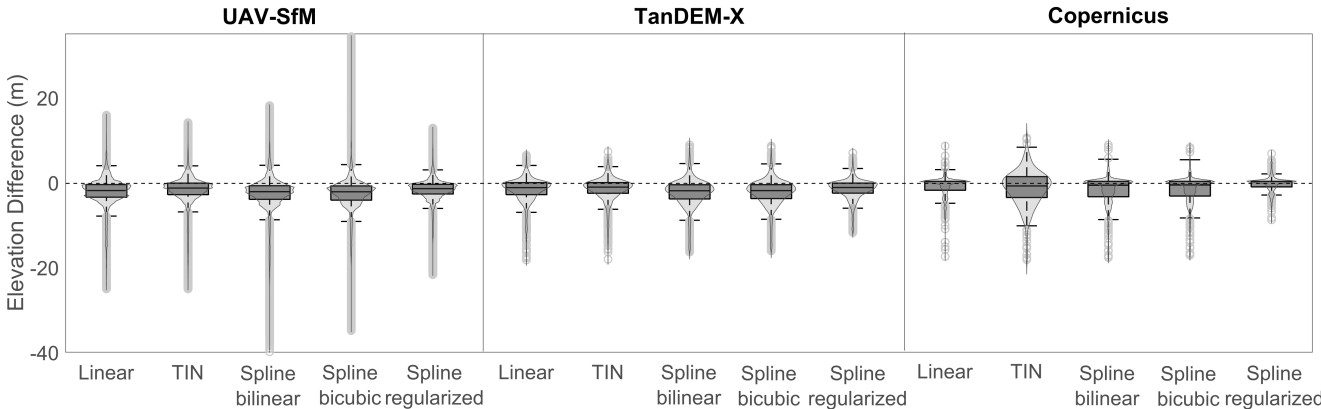

**Figure C5. Interpolation error.** The calculated elevation differences between the interpolated surface and DEM surface respresent the interpolation error and are displayed as violin plots overlain by boxplots. The interpolation error has been determined for all three DEMs (UAV-SfM (0.2 m), TanDEM-X (12 m) and Copernicus (30 m)) and for five interpolation methods (Linear, TIN, Spline bilinear, Spline bicubic and Spline regularized). The distribution of the full dataset containing all the individual pixels is displayed.

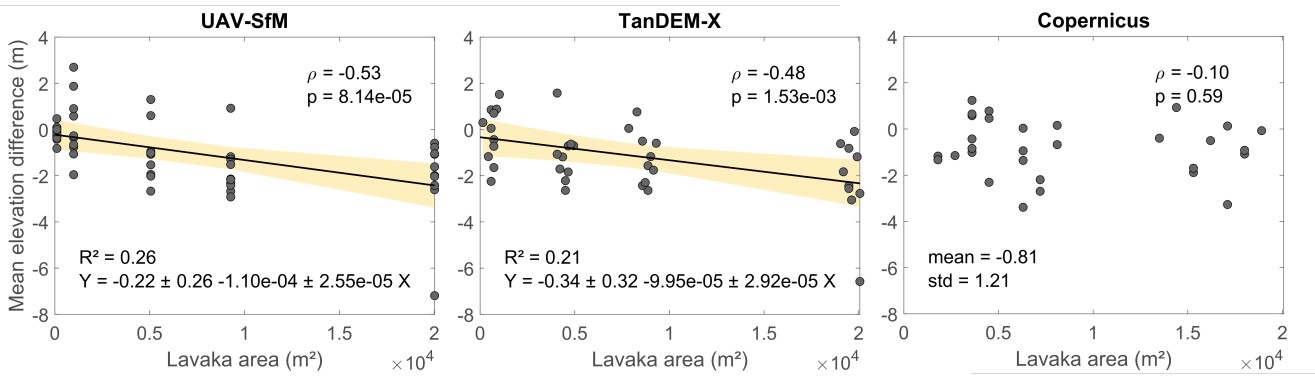

**Figure C6. Mean interpolation error vs. lavaka area.** The correlation of the mean interpolation error, i.e. the difference between the interpolated and DEM surface, per lavaka is verified for all three DEMs: UAV-SfM (0.2 m), TanDEM-X (12 m) and Copernicus (30 m). For the Copernicus DEM a significant correlation between both factors is absent $\rho = -0.10$, p = 0.59. The mean elevation difference of $-0.81 \pm 1.21$ m is used for all lavaka in the case of Copernicus. For the UAV-SfM and TanDEM-X DEM a significant decrease in mean elevation difference is observed with increase lavaka area ($\rho = -0.53$ and -0.48, respectively with p<0.05). The linear relationship between both factors and corresponding uncertainties are used to assess in the interpolation errors in these cases. The shaded area indicates the 95% confidence interval of the fitted relationship, reported uncertainties on the $a$ and $b$ coefficients are the standard deviations.

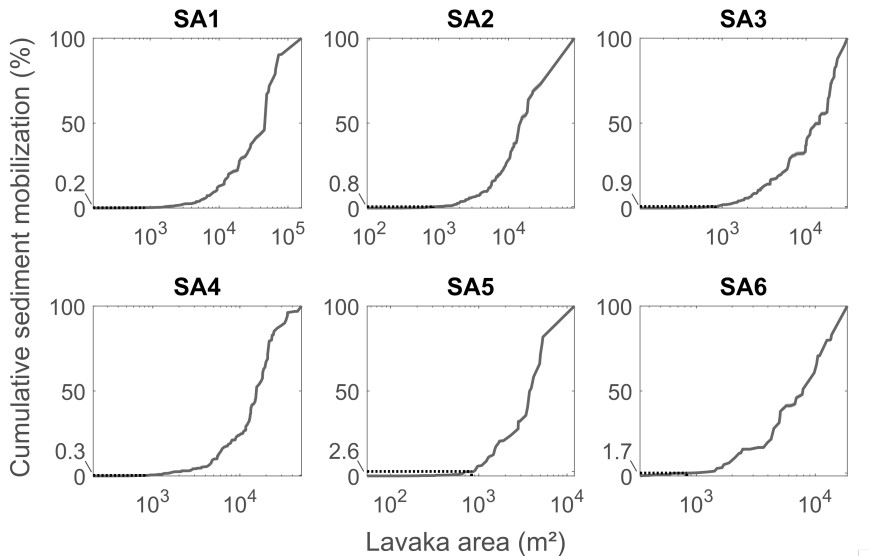

**Figure C7. Cumulative lavaka sediment mobilization per study area** The relative cumulative lavaka sediment mobilization is plotted as a function of lavaka area for all study areas. The fraction of sediment supplied by lavaka smaller than the identified TanDEM-X threshold $(800 \pm 250 \text{ m}^2)$ is indicated by the black dotted lines. This fraction is also added to the y-axis. Note that the lavaka areas are plotted on a log-scale.

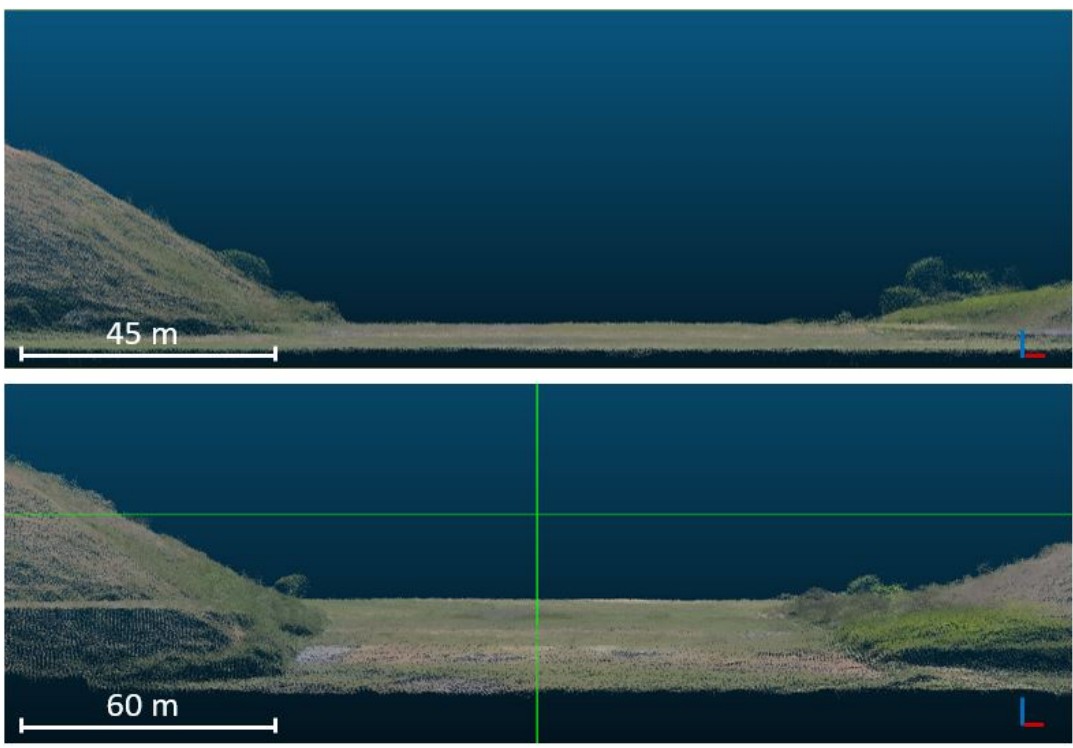

**Figure C8. UAV-SfM point clouds over flat areas.** In order to verify the presence of vertical doming due to the use of a fish-eye lens for the UAV-SfM DEM, the point clouds are visually inspected over flat surfaces. Visual inspection does not indicate the presence of vertical doming.

*Author contributions.* G.G. and S.B. acquired funding for the project and designed the study together with L.J, B.C, T.R. and T.R., where the main supervision was provided by L.J.. Obtaining the necessary drone permits and flights was successful due to the mutual efforts of L.J., B.C., V.F.R., L.B. and E.A.J.. E.A.J. was the remote pilot of the UAV and co-developed the used UAV-PPK-SfM method. L.B. analyzed the data and wrote the manuscript with inputs from all authors.

*Competing interests.* The authors declare that they have no conflict of interest.

*Acknowledgements.* This research would not have been possible without the help of our local partners at Laboratoire des RadioIsotopes, the local authorities of the Ambatondrazaka and Amparafaravola district and the authorization from the Aviation Civil de Madagascar to fly the UAV over their territories. Kristof Van Oost's expertise has been crucial in obtaining high quality UAV DEMs. Ny Riavo Voarintsoa, Rónadh Cox and Steven Bouillon are wholeheartedly thanked for their insights and inspiring discussions on this topic. This research is part of the MaLESA project funded by a KU Leuven Special Research Fund grant (An integrated approach for assessing environmental changes in soil-covered landscapes: the case of Madagascar). L. Brosens and B. Campforts received a doctoral (11B6921N) and postdoctoral (12Z6518N) grant from the Research Foundation Flanders (FWO), respectively. L. Jacobs, B. Campforts, V. F. Razanamahandry and L. Brosens received a YouReCa Mobility Allowance in 2018. L. Brosens and V.F. Razanamahandry received a second YouReCa Mobility Allowance in 2019 and a FWO Travel Grant (V436719N and V436319N). We are very grateful for the constructive feedback of Benjamin Purinton, Armaury Frankl and Rónadh Cox which have drastically improved the methodological rigour and quality of the manuscript.

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
