# Peer review of "Comparative analysis of Copernicus, TanDEM-X and UAV-SfM DEMs to estimate lavaka (gully) volumes and mobilization rates in the Lake Alaotra region (Madagascar)"

_Earth Surface Dynamics, 2021_

## Referee Comment (RC1)

**Review of Earth Surface Dynamics submission: "Comparative analysis of SRTM, TanDEM-X and UAV-SfM DEMs to estimate lavaka (gully) volumes and mobilization rates in the Lake Alaotra region (Madagascar)" by Brosens et al.**

The study of Brosens et al. examines the applicability of three DEMs at three different resolutions (and different sources) to measure the erosion of lavaka (gully) features in Madagascar. The gullies are carefully identified from satellite and aerial imagery at three different dates (1949, 1969, 2011-2018) and their extents are digitized. Following this, pre-erosion surfaces are created for each gully and the volume of excavated sediment is measured. This allows the authors to build area-volume relationships to apply to the three time steps, and measure volumes from area alone. Of great note, they find two order of magnitude higher erosion rates in comparison to cosmogenic radionuclides.

First off, I want to thank the authors for exposing me to these dynamic and fascinating geomorphic features. While I am familiar with significant gully erosion in other places, I had never heard of the lavakas of Madagascar, and they are impressive. I read the study with great interest since bridging gaps in satellite and aerial (i.e., drone) measurements to quantify geomorphic processes presents exciting opportunities, albeit with significant challenges. These challenges are both in terms of spatial and temporal resolution differences (which the authors cover) and dataset accuracy (which the authors mention but do not consider). Overall, I found the paper interesting and think it should be published in ESurf, but there needs to be some major revisions.

I begin by listing my primary concerns, followed by some more specific comments. The references that are not already included in the submitted manuscript are provided at the end of this review.

**Primary Concerns**

The paper could do with a dedicated discussion section. There are many points I make below which would be valid items for a discussion, and other points that may expand the methods and results. Furthermore, the discussion section would allow the authors to more fully place the study and results in the context of other work on gully erosion cited (e.g. Cox et al., 2009, 2010; Perroy et al., 2010; Vanmaercke et al., 2021). Placing the chosen methods, observed volumes, and the causes of (increased?) gully erosion in the context of these other studies will present a fuller, and more citable, study.

I do not agree with using the SRTM DEM as the 30 m resolution dataset for a number of reasons. Firstly, as stated by the authors, SRTM uncertainties are often > 5 m, precluding accurate volume calculations in many cases. Secondly, previous work (Smith and Sandwell, 2003; Farr et al., 2007) has shown that the actual resolution of SRTM is likely on the order of 45-60 m. Finally, better open access DEMs exist. The authors could instead use the ALOS World 3D 30 m (available here: https://www.eorc.jaxa.jp/ALOS/en/aw3d30/index.htm) or, the cutting edge Copernicus DEM (available here: https://portal.opentopography.org/datasetMetadata?otCollectionID=OT.032021.4326.1). The ALOS data has low vertical uncertainties and is based on a resampled 5 m DEM (cf. Purinton and Bookhagen, 2017) and the Copernicus DEM is essentially a 30 m version of the TanDEM-X (references: https://spacedata.copernicus.eu/documents/20126/0/GEO1988-CopernicusDEM-RP-001_ValidationReport_V1.0.pdf and https://spacedata.copernicus.eu/documents/20126/0/GEO1988-

CopernicusDEM-SPE-002_ProductHandbook_I1.00.pdf) I strongly recommend the authors utilize the newer Copernicus DEM in place of SRTM. It may be the case that the newer dataset can be included in the breakpoint analysis, especially considering that the largest lavaka make up the majority of exported sediment (as shown in Figure B2). If a 30 m open-access, near-global DEM (Copernicus or ALOS) provides some decent results, then the impact of this study would be greatly increased.

Vertical uncertainties of the different DEMs are never considered in the analysis. But this is a vital step when using spaceborne DEMs for volume estimations, for instance in the cryospheric (e.g., Brun et al., 2017) and geomorphic (e.g., Purinton and Bookhagen, 2018; Bessette-Kirton et al., 2018) communities. While I think the authors can safely argue for negligible vertical uncertainties (with the proper citations) in the UAV-DEM, these cannot be ignored in the case of TanDEM-X and SRTM (cf. Purinton and Bookhagen, 2017, 2018). In the case of the Copernicus DEM I reference above, TanDEM-X uncertainties can be used given this DEM was generated from the same source. Preliminary reports on Copernicus DEM accuracy are available here: https://spacedata.copernicus.eu/documents/20126/0/GEO1988-CopernicusDEM-RP-001_ValidationReport_V1.0.pdf. Furthermore, depending on the presence of vegetation (i.e., forests) in the DEM pixels, these different datasets (radar X-band for TanDEM; C-band for SRTM; optical images for ALOS) may have additional uncertainties. These uncertainties with regards to the study area characteristics (bare-earth, forests, bushes) should be mentioned.

I think the handling of vertical uncertainties can be done in a discussion section, but it may be better inserted in the methods and results. I suggest: an uncertainty value (e.g., RMSE or NMAD) is selected from the literature regarding each DEM and this uncertainty is propagated to the volume estimates. This would then put error bars on the regressions in e.g., Figures 5 and 6, which could be considered during the power-law fitting. This uncertainty will also propagate in the negative vs. positive volume calculations, presenting a range of percentages rather than an exact value, which implies perfect DEM accuracies.

In Purinton and Bookhagen (2018) we also attempted volume estimation (in this case using the more common DoD approach) between the SRTM and TanDEM-X DEMs. In this case, the uncertainties associated with the SRTM precluded widespread geophysical results, except in the areas of very rapid / large magnitude change. This study highlights the influence of spaceborne DEM accuracy and the care that must be taken when combining older and newer datasets from different sources, with different errors, in a given analysis. While detailed correction steps are not necessary in this case, this work should be noted, particularly since it is in the same journal.

In the interpolation step, I think some more work and justification is needed. I see the temptation to generate curved surfaces using splines, but I think nearest neighbor void filling is somewhat safer (not based on any parametrized curve fitting) and simpler. This may result in "staircase" artifacts, but those may not have a huge impact on the final volume estimates, or the vertical uncertainties of the spaceborne DEMs may have a larger impact. The authors should experiment with the GDAL gridding methods accessible in QGIS (Processing Toolbox > GDAL > Raster Analysis: GRID (nearest neighbor, linear, etc.)), since these are robust and widely used. Previous authors working on gullies have had success generating pre-erosion surfaces with simpler (i.e. parameter-free) interpolation (e.g., Perroy et al., 2010; Eustace et al., 2009, Evans and Lindsay, 2010). Furthermore, I'm not convinced the random-points approach is entirely necessary. Couldn't the authors just use each elevation value that the "horseshoe" overlaps with one time? Otherwise some elevations (single pixels) are used multiple times, which I don't understand the reason for and seems inappropriate.

I suggest the authors at least attempt the processing with the nearest neighbor approach, mentioning that this is parameter free and uses only the original elevation values, and if the results create significantly

more negative volume, then mention this as justification for higher order techniques. This can be an item in the discussion section.

**Specific Comments**

Line 3: "high resolution DEMs", I prefer that the exact resolution in meters is mentioned or the terms high, medium, and low resolution are clearly defined and a range of values for each is given. Twenty years ago 30 m DEMs were very high resolution. For posterity, best to be exact. Please note this change in other places in the manuscript (e.g. Line 31)

Line 10: "SRTM DEM is too coarse", or too inaccurate? This pertains to one of my primary concerns. Also bear in mind the SRTM realistically has a ground resolution closer to 45-60 m (Smith and Sandwell, 2003; Farr et al., 2007)

Line 19-20: What do these different rates correspond to? Are they the range and average of the six different study areas?

Line 26: Rephrase "more and more"

Line 30: "remote sensing product". Well, UAVs are also remote sensing if we define remote sensing as measurements that don't disturb the surface. I would change this to "spaceborne product"

Line 32: They don't necessarily need to all be here, but somewhere (perhaps in a new dedicated section) references to the accuracy of these various datasets should be cited (e.g., Purinton and Bookhagen, 2017; Rizzoli et al., 2017; Wessel et al., 2018)

Line 36: "Gully erosion…", awkward sentence, rephrase.

Line 39: From "where", awkward clause. Consider new sentence and/or rephrasing.

Line 41: What is being referred to by "high resolution surface imagery"? If the authors are referring to GoogleEarth then be explicit and I suggest also referencing Fisher et al. (2012).

Line 63-64: For this step the lavaka area was taken from the most recent (2010s) polygons? Maybe state this.

Line 80: "are available" maybe change to "were generated", since these are data the authors created for this study. And what a nice dataset it is!

Line 81 and Table A1: I'm not familiar with Maxar-Vivid-WVO2, but I suppose this refers to the WorldView-2 satellite? Please call them WorldView-2 if so. Also, are these images proprietary or received through grant? They should be mentioned in the "Code and data availability" section. One more point: the ground resolution of the aerial photos and satellite images should be mentioned here (and/or in the table).

Figure 1: The color-scale could be improved here and elsewhere using e.g. the instructions here: https://gis.stackexchange.com/questions/94978/elevation-color-ramps-for-dems-in-qgis. The authors can decide for themselves, but the current blue to red scale is odd for topography. The red-blue could then be saved and used for topographic difference as is often done (e.g., Wheaton et al., 2010). Furthermore, in (a) the topography around the Alaotra catchment should also be shown, otherwise it looks like this implies ocean around it, which makes the inset map of Madagascar also confusing. Note the Krieger (2007) citation appears twice, drop one of them.

Line 88-89: The method of resampling to UTM coordinates should be mentioned (bilinear?). Also, importantly, the SRTM is likely referenced to the EGM96 geoid, whereas the TanDEM-X is referenced to the WGS84 ellipsoid. Was a vertical datum conversion done to bring the datasets into the same vertical reference? And what is the vertical datum for the UAV DEM?

Line 95-99: A few points here. While the UAV DEM was likely of high quality, the authors should confirm that there is no notable doming effect from using a fish-eye lens and no ground control points. Doming (cf. James and Robson, 2014) is a known issue with SfM-MVS from drones, and fish-eye lenses will exacerbate this. I suggest the authors examine the raw UAV point cloud in either Pix4D or perhaps CloudCompare over flat surfaces in the study area (I realize this may be difficult to find) and take visual note of any large-scale warping. Ideally a reference dataset or independent GNSS points could be used for validation, but that is missing here, correct? One more thing: there are many other citations regarding UAV-DEMs for geomorphic analysis and I recommend including them alongside or in place of Grohmann, 2018 (e.g. Cook, 2017 and others therein and referencing this study). The authors could even give a range of expected vertical accuracy from their UAV-DEM taken from the literature in other cases where GCPs were not used. This would provide justification for passing these off as "negligible".

Line 116: The auxiliary files delivered with TanDEM-X (the COV.tif file) would allow the authors to specifically report the number of coverages used to generate the final DEM in this study area. I suggest reporting this value (mean +/- standard deviation, or range) since it has a large impact on DEM quality (vertical uncertainty).

Line 124-125: As noted, these height errors are really high for the SRTM and this is likely an inappropriate dataset for this application, particularly when I consider the bottom row of Figure 2 where the SRTM differences is maybe less than 5 m? Although again, as I note below, an improved (classified) color-scale would be helpful here.

Line 141: What is meant by "precise identification"? In this case, is the horseshoe drawn to not include the other lavakas? So a sort of broken horseshoe shape? Please provide a visualization of the points selected for interpolation on Figure B1. From this figure I don't see in many cases how enough points could be selected to provide a reasonable interpolation in some of the locations where the lavakas appear to have no, or very little, pre-erosion topography preserved where they are touching.

Line 146: Delete hyphen in "DEM-pixel"

Line 160-162: I'm concerned about this 1-pixel lower limit. A single pixel can easily represent an inaccurate measurement. It would be better to have a multi-pixel lower limit. This could be 5 pixels (i.e., lavakas smaller than 5 pixels in each DEM are not considered), but that would remove a significant number of lavakas from the e.g. 30 m DEM. On the other hand, per the scaling relationships, perhaps these <5 pixel lavakas do not contribute significantly to the sediment budget?

Figure 2: The elevation colorbar is reversed with respect to Figure 1, please check it here and elsewhere (and as suggested consider different color scales). In this figure the elevation difference colorbar should really be classified, not continuous. For instance, if I look at the SRTM it's hard to tell but those green values are maybe < 10 m (maybe even < 5 m)? That's getting awfully close to the vertical uncertainty of SRTM. Classified color scales broken into ~5 m ranges with perceptually distinct colors would help a lot.

Line 216: See primary concerns above. Here the uncertainties on the a and b coefficients may be too small, since the regression does not consider uncertainties on the volume calculation which may be

significant for the TanDEM-X and SRTM (if the authors continue to use the SRTM and not ALOS or Copernicus).

Line 226: "1.3% vs. 0.3%" and others, values should be switched (0.3 vs. 1.3) to match the preceding sentence.

Line 233: "increasing" should be "decreasing"

Line 241: Remove hyphen in "data-areas"

Line 260: Reference Smith et al. (2019) (in this journal) when discussing "optimal grid resolution"

Line 275: "at" should be "of"

Line 295-298: No mention of TanDEM-X vertical uncertainties. Here and elsewhere the uncertainties are only considered with regards to resolution.

Line 308-313: The scaling relationships are considered in the context of landslide studies. Do these results really hold for gully scaling? Can the authors include references or justification for the landslide comparison? I could accept an argument that the scaling relationship is similar, but the processes are different.

Figure 6: Am I correct in my visual interpretation that the break-point is found at the point where the relationship becomes 1:1? In that case, is the broken-stick analysis entirely necessary, or could a simpler approach be to just consider where the RMSE (or some measure of spread) from a 1:1 line passes below some limit (e.g. 5%). Just food for thought, I think the broke-stick is valid, but it may be worth mentioning this 1:1 change-point.

Line 322-323: And this is why uncertainties on the volume estimates are important for the coefficient estimation (small differences in coefficients lead to large differences in volumetric growth and mobilization).

Lines 337-345 and Figure 7: This would be good fodder for a discussion section and would allow expansion and inclusion of other references to gully studies (e.g. Vanmaercke et al., 2021 and references therein). I recommend removing the correlation coefficients from plots b-d in Figure 7 and where they are referenced in the text. These are only six data points and it may be best to just discuss the graphical trends observed, since these are not robust statistics in this case.

Line 345: This Brosens et al. (in review) paper is seemingly important for the discussion of results (reasons for gully changes). Hopefully review progresses there quickly and a final citable result is available for this paper / discussion. This paper leaves me asking "why are the lavaka erosion rates increasing?" In Perroy et al. (2010) (worth citing here) gully erosion increased in response to grazing.

Line 361: "Table A1 and 1", not sure what the 1 is supposed to refer to.

Line 386: "area" should be "are".

I had a look at the three example lavaka files and code on GitHub, thanks for publishing that, it is really useful. However, when I tried to run the LavakaVolumesPyQGIS.py script, I received error messages about importing modules and functions. There are no imports at the top of the script, is this something that is missing? Or could the authors add to the GitHub README the steps to actually run the script? Maybe this is done through a GRASS shell, but I'm not familiar with those steps. I tried adding "import os" and "from qgis.core import *" but then got an error about an undefined "processing" variable. These

do not need to be detailed instructions, but the common way of running a script at the command line (python <script name>) does not work in this case, so maybe just note the steps to open and run the script.

I hope these are helpful and constructive criticisms, and I welcome further discussion in the open online forum.

Sincerely,

Ben Purinton

**References**

Bessette-Kirton, E. K., Coe, J. A., and Zhou, W.: Using Stereo Satellite Imagery to Account for Ablation, Entrainment, and Compaction in Volume Calculations for Rock Avalanches on Glaciers: Application to the 2016 Lamplugh Rock Avalanche in Glacier Bay National Park, Alaska, J. Geophys. Res.-Earth, 123, 622–641, https://doi.org/10.1002/2017JF004512, 2018.

Brun, F., Berthier, E., Wagnon, P. et al. A spatially resolved estimate of High Mountain Asia glacier mass balances from 2000 to 2016. Nature Geosci 10, 668–673 (2017). https://doi.org/10.1038/ngeo2999

Cook, K. L.: An evaluation of the effectiveness of low-cost UAVs and structure from motion for geomorphic change detection, Geomorphology, 278, 195–208, 2017.

Fisher GB, Amos CB, Bookhagen B, Burbank DW, Godard V, Whitmeyer SJ. Channel widths, landslides, faults, and beyond: The new world order of high-spatial resolution Google Earth imagery in the study of earth surface processes. Geological Society of America Special Papers. 2012 Oct 1;492(01):1-22: http://burbank.faculty.geol.ucsb.edu/Site/Publications_files/Fisher_etal_GSASpecPap492_2012.pdf

James MR, Robson S. Mitigating systematic error in topographic models derived from UAV and ground-based image networks. Earth Surface Processes and Landforms. 2014 Aug;39(10):1413-20. https://doi.org/10.1002/esp.3609

Purinton, B., & Bookhagen, B. (2017). Validation of digital elevation models (dems) and comparison of geomorphic metrics on the southern central Andean plateau. Earth Surface Dynamics, 5(2), 211–237. https://doi.org/10.5194/esurf-5-211-2017

Purinton, B., & Bookhagen, B. (2018). Measuring decadal vertical land-level changes from srtm-c (2000) and tandem-x in the south-central Andes. Earth Surface Dynamics, 6(4), 971–987. https://doi.org/10.5194/esurf-6-971-2018

Smith, B., and D. Sandwell (2003), Accuracy and resolution of shuttle radar topography mission data, Geophys. Res. Lett., 30(9), 1467, doi:10.1029/2002GL016643.

Smith, T., Rheinwalt, A., and Bookhagen, B.: Determining the optimal grid resolution for topographic analysis on an airborne lidar dataset, Earth Surf. Dynam., 7, 475–489, https://doi.org/10.5194/esurf-7-475-2019, 2019.

Wessel, B., Huber, M., Wohlfart, C., Marschalk, U., Kosmann, D., and Roth, A.: Accuracy Assessment of the Global TanDEM-X Digital Elevation Model with GPS Data, ISPRS J. Photogram. Remote Sens., 239, 171–182, 2018.

---

## Referee Comment (RC2)

[referee-annotated manuscript omitted]

---

## Community Comment (CC1)

This work by Liesa Brosens and colleagues takes a very interesting approach to lavaka analysis in Madagascar, and I applaud their broad thinking.

However, I share the concerns about model accuracy and reliability raised by Ben Purinton in his review, and wanted to make a few comments on those issues, in addition to raising some additional questions about the geomorphology. These points are intended in the constructive and collegial spirit of the open review process.

The bulk of the image data on which the authors' model is based is not high resolution: only two of their study areas are imaged at 20 cm/pixel, all the others are imaged at 12-30 m/pixel. Many lavakas are only a few pixels across, and elevation changes are in many cases close to the resolution of the imagery, which will not capture internal relief changes within the gullies. It also means that there must be substantial potential error with the area calculation, as a lot of edge detail will not be resolved; and lavaka edges can be quite complex in shape.

The authors apply a simple least-squares fit to area-volume data to provide a relationship that they then use to drive much of their analysis. I did not find a graph of the XY data that they used, or an $R^2$ value or other measure of quality of fit. But I expect that those data are very noisy, and the relationship is not very precise. To test this, I looked at the data that the authors made available on Fileshare, and I plotted area vs relief (as a proxy for the depth data, which was not in the file). I will share two observations. First, the linear fit fails at smaller volumes. This is a problem, because lavaka size follows a power-law distribution, with the majority being at smaller sizes: a robust line of fit should model the bulk of the population, so I think the authors should consider whether it works to have a large proportion of the population not represented by the fit line. Second, the data clouds for the different study areas have different lines of fit, so a one-size-fits-all approach may not be valid.

I also worry about the circularity of using area to derive volume, and then using that derived quantity to an area-volume model, as the authors do in Fig. 5. I would want to see a much more detailed unpacking of the caveats that attend this approach, and in particular I think it would be important to show the original area-volume data used to derive the base relationship. This method may provide interesting ways to look at and think about landscape evolution at a broad scale, but I do think that the noisiness of the base data, and the imperfection of the original line of fit to those data, warrant large error envelopes; and severely limit the precision of downstream models based on that original line of fit. The authors should go into these issues in more detail, because it feels as though they are somewhat brushed under the rug (see comment in previous paragraph about no figure showing the original data relationship with its fit line). It seems to me that there are multiple nested and cumulative uncertainties, which could be more completely and thoroughly addressed/ (I hope I am not missing something obvious here, and will be happy to be corrected if I am).

I am concerned also about the underlying geomorphology. Lavakas evolve through different stages, with very different activity levels over time (per the excellent and detailed work of Neil Wells and Benjamin Andriamihaja). Older lavakas are larger, but also evolve to be more shallow

than (see Wells et al., (1991) ESPL 16: 189-206, Fig. 3). Each lavaka follows its own adventure in terms of growth, deepening, and shallowing (with the shallowing due in large measure to capture within the lavaka of the erosional materials from its walls).  So I have two points here. First is that the known geomorphology of lavakas should probably be considered in any model for their evolution through time; and second, that I would like to see more justification for using landslides as analogues (because although lavakas have headscarps, and evolve via collapse processes, they do not really behave like landslides because of their narrow outfall channels). This may mean that some of the assumptions regarding bias-correction factors may need some adjustment, as these which come out of lanslide modelling (per Lines 195-200), and are baked into the authors' model for area-volume relationships. I'm not saying that landslides are an inappropriate analogue; but I am saying that the authors should provide a firm geomorphologic rationale (which I cannot find section 2.4 or elsewhere in the current manuscript).

In calculating sediment mobilisation rates,  the authors state (Line 210) that they used a bulk density of 1.5 t/m$^3$, based on soil corings 2 m deep.  This value is likely to be too high.  The surface laterite layer in this area is a couple of m thick, so this is what the cores will have sampled.  Below this, and forming the bulk of the material that is evacuated from lavakas, is saprolite, which is highly porous and therefore has lower bulk density.  A better value would be 1.1-1.2 t/m$^3$, in line with previous work (e.g. Heimsath et al. (1997) Nature 388: 358–361, Montgomery (2007) PNAS 104: 13268–13272, and the 2008 UVM thesis of Matt Jungers).

Finally, I query the authors' conclusion "current mobilization rates exceed the long term rates by two orders of magnitude" (Line 370).  The problem is that the comparison being made is between apples and pears: although the authors do provide the proviso that "not all mobilised lavaka sediment will end up in the rivers", they are assuming that these very different datasets are comparable.  In fact, most of the (limited, for sure) evidence suggests that a lot of lavaka sediment is deposited close to (and even within) the lavakas themselves (similar to the landslides on which they model lavakas). Thus, long-term lake infill and river-sediment-derived erosion estimates will miss this material.  If lavaka sediment didn't make it into those archives in the past, then the values from those archives cannot be used as a comparison with mobilisation rates from modern lavakas.

I admire the amount of work and data collection that went into this manuscript, and I hope that the authors will accept this critique in the collegial spirit in which I offer it.

---

## Author Comment (AC1)

Dear Dr. Benjamin Purinton,

First of all we would like to thank you for the very constructive and thorough feedback that you provided on our manuscript. We highly appreciate your suggestions, which will drastically improve the quality and rigor of our analysis and manuscript.

While a more in-depth response and the updated manuscript will be provided upon receiving the decision of the editor, we would already like to address your primary concerns briefly, as we have in the meantime already tested and implemented most of your methodological suggestions. We would be more than happy to further discuss these methods and results and are open to additional suggestions.

1) **Separate discussion section**
   We do agree that the paper would benefit from a separate discussions section in which we can address both technical and contextual aspects in more detail. Frankl (reply 2021) and Cox (reply 2021) have also suggested a more elaborate discussion of possible technical caveats or limitations; this will be foreseen in the adapted manuscript.

2) **Alternative 30 m DEM: Copernicus**
   We were happy to be introduced to this new 30 m Copernicus DEM, which will indeed provide many benefits compared to SRTM. We have therefore replaced the 30 m SRTM DEM by the 30 m Copernicus DEM for all our analysis. Using the Copernicus DEM furthermore facilitates better uncertainty assessment, which is addressed in more detail in point 4.

3) **Vertical uncertainties**
   We agree that vertical uncertainties will be substantial for the TanDEM-X and Copernicus DEMs. After verification of the literature and consideration of our specific application, we have opted to address two specific uncertainties or errors, which we believe to be important in our case: i) the interpolation error, and ii) the relative height error.

   Interpolation error
   After considering your suggestions on the interpolation method (see point 4) we have decided to develop a more rigorous method to determine which interpolation method works best in our landscape setting. To calculate our volumes, we subtract the current DEM elevation from the interpolated pre-erosion surface. We previously assessed the performance of the interpolation methods tested by comparing the percentage of negative volume that resulted from these volume calculations. Therefore, we have now adopted the following method to assess i) which interpolation methods works best ii) what the error/uncertainty of this methods is.

   We have taken five different lavaka polygons with different sizes that span the range of our lavaka polygons (100 m² (n = 10), 1000 m² (n=10), 5000 m² (n=10) and 20 000 m² (n = 10)). These 50 lavaka polygons were then placed on unaffected convex-shaped hillslopes on which lavaka typically occur, together with the corresponding horseshoe-shaped pre-erosion polygons. We then tested different interpolation methods (more details are provided in point 4) and calculated the difference between the interpolated surface and the DEM. This difference then gives the interpolation error, as a perfect interpolation would results in an identical interpolated and original surface.

Based on obtained height differences between interpolated and original surfaces several error metrics were calculated (mean, median, root mean squared error (RMSE), mean absolute error (MAE) and standard deviation (std)), which were then used to i) identify the best interpolation method and ii) estimate the interpolation error. Next, it was verified if the magnitude of the interpolation error depends on the size of the lavaka, in order to correctly account for these errors. Our main preliminary results are presented in Table 1, where our main conclusions are the following:

*Table 1: Mean, median, mean absolute error (MAE), root mean square error (RMSE) and standard deviation (std) of the Interpolation error in meter for the different DEMs and interpolation methods*

| | UAV-SfM (0.20 m) | | | | | TanDEM-X (12 m) | | | | | Copernicus (30 m) | | | | |
|---|---|---|---|---|---|---|---|---|---|---|---|---|---|---|---|
| | Linear | TIN | Spline bilinear | Spline bicubic | Spline reg. | Linear | TIN | Spline bilinear | Spline bicubic | Spline reg. | Linear | TIN | Spline bilinear | Spline bicubic | Spline reg. |
| Mean | -2.29 | -1.83 | -2.63 | -2.66 | -1.75 | -1.93 | -1.62 | -2.63 | -2.58 | -1.76 | -1.44 | -1.58 | -2.68 | -2.54 | -0.91 |
| Median | -1.92 | -1.32 | -2.19 | -2.21 | -1.47 | -1.40 | -1.17 | -2.16 | -2.11 | -1.38 | -1.00 | -1.19 | -2.34 | -2.13 | -0.60 |
| MAE | 2.94 | 2.53 | 3.28 | 3.24 | 2.21 | 2.41 | 2.17 | 3.15 | 3.09 | 2.13 | 2.02 | 3.02 | 3.04 | 2.86 | 1.44 |
| RMSE | 4.21 | 3.90 | 4.66 | 4.41 | 3.05 | 3.48 | 3.18 | 4.22 | 4.14 | 2.97 | 2.86 | 3.95 | 4.11 | 3.91 | 2.05 |
| Std | 3.53 | 3.45 | 3.84 | 3.51 | 2.50 | 2.90 | 2.73 | 3.30 | 3.23 | 2.39 | 2.47 | 3.63 | 3.12 | 2.98 | 1.85 |

i)  The interpolated surface is on average lower than the real surface (mean and median error < 0). Our calculated volumes will therefore, on average, be underestimated.

ii)  Coarse DEMs have a lower error. As the coarser resolution DEM contains less topographic detail, it makes sense that the difference between a generalized interpolated surface and the real surface is smaller as when compared to high resolution DEMs that contain more detailed micro-topography that cannot be easily interpolated

iii)  Overall, the Regularized Spline with tension interpolation results in the lowest errors.

iv)  Based on both pearson and spearman correlation coefficients no significant relationship is observed between lavaka area and the mean or median error for the 30 m Copernicus DEM (Figure 2). However, for the finer resolution TanDEM-X and UAV-SfM DEMs we do find a significant correlation between lavaka area and the mean or median error of a lavaka (Figure 2).

[Figure]

*Figure 2:Ccalculated pearson and spearman correlation coefficients and their p-value for the relationship between lavaka area and the mean interpolation error of a lavaka.*

Based on these results we now take into account the interpolation error for the different DEMs by running a Monte Carlo analysis:

i) Copernicus: draw random values from a normal distribution with the mean = mean lavaka error and std = std of the mean lavaka error (mean = -0.81 m, std = 1.21 m)

ii) TanDEM-X and UAV-SfM: Established a relationship between lavaka area and mean error. Use this relationship to estimate the interpolation error for each lavaka based on its size. Uncertainties are taken into account by considering the uncertainties on the fitted coefficients, which are again drawn from a normal distribution with known mean and std, where we use copula to account for the correlation between both coefficients.

[Figure]

*Figure 3: fitted linear relationships between lavaka area and mean error for the UAV-SfM and TanDEM-X DEM*

Relative height error

Typically, the performance of a DEM is assessed by considering its absolute vertical accuracy. We, however, argue that this metric is not the most suitable in our case, as we are rather interested in relative pixel-to-pixel errors instead of absolute errors: the difference between the interpolated surface and DEM will determine the volume. An absolute error of X m will not result in a different volume estimate if this absolute error is the same for all DEM pixels. However, if the relative height of the pixels is not correct, this will result in different volume estimates. Therefore, we have opted to use the relative height error. We assume that this relative height error will be negligible for our high resolution UAV-SfM DEM and use the height error masks (HEM) that are

provided for the TanDEM-X and Copernicus DEM to assess this error. The height error mask gives the height error for each pixel in the form of the standard deviation. This error is considered to be a random error and does not include any contributions of systematic errors (Wessel, 2016).

For each lavaka we now calculate the mean HEM-value, which we then use in our Monte Carlo simulations: for the relative height error we draw random values from the normal distribution with mean = 0 (height of the pixel) and std = mean HEM of the lavaka.

Both for the interpolation and relative error we use a lavaka as the observational unit, where we work with the mean errors and standard deviations obtained for this observational unit. In this way we make abstraction of the fact that pixels within a lavaka will likely be autocorrelated and correct for the error on the lavaka as a whole.

By combining both errors in one Monte Carlo simulation, we have been able to calculate the mean volume and uncertainty on this volume (std) for each lavaka, which we can further propagate into the establishment of the Area-volume relationship and subsequent calculations.

4) **Interpolation**

Two concerns were raised by the reviewer: i) the use of parameterized spline interpolation and ii) random-points approach.

Interpolation method

First of all we have now improved our method to assess which interpolation method works best (see point 3 interpolation error). Furthermore, we have assessed the performance of different interpolation techniques, all available in Open Source QGIS:

      i)        Linear (GDAL)

      ii)      TIN (Qgis)

      iii)     Spline: Bilinear (GRASS)

      iv)    Spline: Bicubic (GRASS)

      v)     Spline: regularized with tension (GRASS)

We did some manual try-outs with the nearest neighbor (GDAL) algorithm that was proposed by the reviewer. Visual interpretation of these results (Figure 4), together with the fact that this method is typically applied to categorical data made us decide to not consider this method in further analysis.

[Figure]

*Figure 4: Examples of four different interpolation methods for the TanDEM-X DEM: Regularized spline with tension, TIN, Linear and Nearest neighbor interpolation.*

From the different error metrics (Table 1) it was concluded that regularized spline with tension yields the lowest interpolation errors. Therefore, we decided to use this interpolation method for all further analysis. We will, however, add a discussion on the disadvantages and limitation of this method.

Random-points approach

We agree with the reviewer that the creation of the random points in the pre-erosion surface polygon is not necessary and might result in incorrect interpolation results as then indeed the same pixel value might be considered multiple times during interpolation. We have therefore changed this method by following the suggestion of the reviewer where we now create one point per pixel, which are then used for interpolation. The steps followed to derive the lavaka volumes are now the following:

i)      Clip the DEM with the pre-erosion polygon. All bordering pixels are included in order to assure that the width of the pre-erosion raster has at least one pixel for the coarse resolution DEMs
ii)     One point per pixel is created to which the height of the pixel is assigned
iii)    The pre-erosion surface is constructed by interpolating between these points
iv)     The volume is calculated by taking the difference between the interpolated pre-erosion surface and the current surface.

To summarize the changes in our methods:

i)      we now use the 30 m Copernicus DEM instead of the SRTM DEM

ii)     vertical uncertainties are calculated for each lavaka based on both the interpolation error and the relative height error. These errors will be propagated in all subsequent calculations.

iii)    the performance of different interpolation methods is assessed based on the interpolation error. This is done by reconstructing intact hillslopes and by calculating the difference between the interpolated and real surface. Regularized spline with tension has the lowest errors. We therefore applied this interpolation method to our further analysis

iv)     The volume calculation workflow is adapted so that we now use one point per DEM pixel for interpolation instead of the random points.

We would like to highlight that the figures presented here will be further modified/finetuned for the revised version of the manuscript. Minor concerns of the reviewer will be addressed in detail in the formal rebuttal letter and in the revised version of the manuscript.

We want to thank Dr. Benjamin Purinton for his thoughtful feedback that will improve the methodological robustness of this work and are open to any further suggestions to improve the quality of our work.

Sincerely,

Liesa Brosens on behalf of the co-authors

**References**

Cox (reply 2021). CC1: Comment on esurf-2021-64 by Rónadh Cox, 19 Oct 2021. https://doi.org/10.5194/esurf-2021-64-CC1

Frankl (reply 2021). RC2: Comment on esurf-2021-64 by Armaury Frankl Oct 2021. https://doi.org/10.5194/esurf-2021-64-RC2

Wessel B. (2016), "TanDEM-X Ground Segment – DEM Products Specification Document", EOC, DLR, Oberpfaffenhofen, Germany, Public Document TD-GS-PS-0021, Issue 3.1, 2016. [Online]. Available: https://tandemx-science.dlr.de/

---

## Author Comment (AC2)

Dear Prof. Dr. Armaury Frankl,

We are very grateful for the constructive and detailed feedback that you provided on our manuscript. We would therefore like to thank you for investing your time and sharing your insights with us. While a more in-depth response and the updated manuscript will be provided upon receiving the decision of the editor, we would already like to address your primary concerns briefly.

We do agree that a more detailed discussion on possible methodological caveats and uncertainties is needed as this is a technical paper. This concern was also raised by Purinton (reply 2021) and Cox (reply 2021). We will therefore add the necessary information and details on these matters in a dedicated discussion section. We furthermore agree that from a theoretical point of view we could already assess which pixel size and corresponding DEM would be fine enough to study these landforms, we will also add this to the manuscript upfront in the introduction and further discuss this afterwards.

We will further address the more detailed comments in the revision of the manuscript.

Sincerely,

Liesa Brosens on behalf of the co-authors

**References**

Cox (reply 2021). CC1: Comment on esurf-2021-64 by Rónadh Cox, 19 Oct 2021. https://doi.org/10.5194/esurf-2021-64-CC1

Purinton (reply 2021) RC1: Comment on esurf-2021-64 by Benjamin Purinton, 30 Aug 2021. https://doi.org/10.5194/esurf-2021-64-RC1

---

## Author Comment (AC3)

Dear Prof. Dr. Rónadh Cox,

First of all we would like to thank you for taking the effort to provide us with additional feedback on our manuscript with your comment in the open discussion. While a more in-depth response and the updated manuscript will be provided upon receiving the decision of the editor, we would already like to address your primary concerns briefly and point-by-point in this reply.

1. **Area calculation**

   Detailed lavaka areas have been delineated based on high resolution (0.5 m) satellite imagery over the period 2011-2018 and based on 2.4 m resolution historical aerial images (these datasets are described in detail in Brosens *et al.* (2021) and are provided the corresponding FileShare repository). On these previously mapped areas, the area-volume relationships obtained in this manuscript have been applied in order to estimate the lavaka volumes in 1949 and 2010s. We will revise the corresponding sections of the manuscript in order to make sure that this is clearly formulated and that the supplementary raw data are more clearly referred to.

2. **Data of the A-V relationship**

   We regret that it was not indicated clear enough in the manuscript that the supplementary data of this work can be found at the following repository: https://doi.org/10.5281/zenodo.5155317. This is mentioned in the *code and data availability* section, but we will also refer to this in the text to make sure that the reader finds these data. In the repository we provide an excel table containing the original lavaka areas and volumes that were used to establish the area-volume relationship for each of the DEMs. This table also contains the detailed areas of all lavaka (see point 1) and the estimated lavaka volumes based on the established A-V relationships (for TanDEM-X and UAV-SfM) as well as the derived volumetric growth and mobilization rates. In the revised version of the manuscript additional uncertainties that result from both the interpolation and relative elevation error will be added to this table (see also our reply to Purinton (reply 2021) for details on the new uncertainty calculations).

   The area-volume data are shown in Figure 5 in log-transformed form. We will add a figure of the non-log transformed data to the supplementary files. In Figure 5 no $R^2$ or other measure of fit is indicated because we use this figure to illustrate that issues are present for the smallest lavaka for the TanDEM-X dataset. The area-volume data that are used to establish the area-volume relationship are displayed in Figure 6b, where the $R^2$ and the coefficients of the fitted linear relationship are displayed. The uncertainties related to these fitted coefficients are indicated in Equation (3) and (4) and are further discussed and taken into account in our subsequent volumetric growth and mobilization rates calculations (line 215-220).

   The topographic data that are presented in the FigShare repository and that were used to verify the relationships by Cox (reply 2021) are based on the TanDEM-X DEM. While we agree that the depth of the lavaka can probably be approached by the relief, this will result in additional scatter. The observation that the linear fit fails at smaller volumes is in agreement with the data that we show in the manuscript: the TanDEM-X DEM is too coarse to accurately assess the volumes of the smaller lavaka (Figure 5). Our proposed breakpoint analysis allows to identify the point below which these TanDEM-X derived volumes suffer from errors. As we further discuss in lines 346 –

355 the impact of not using the smallest lavaka to establish this area-volume relationship on the final mobilization rates is likely minimal, as these lavaka contribute 1.1 to 21.6% of the total mobilized sediment.

We did not try to fit the A-V relationships for the different study areas separately, as we argue that the most robust fit is obtained using all data.

**3. Uncertainties on the Area-Volume relationship**

We agree that even after the breakpoint-correction relatively large uncertainties apply to the established area-volume relationship. In the current version of the manuscript we considered these uncertainties by running a Monte Carlo analysis where we take into account the uncertainties on the fitted A-V coefficients (line 216-220). Based on the additional uncertainties that we have now calculated for the lavaka volumes, where we take into account both the interpolation and relative elevation error (see reply too Purinton (reply 2021)), these uncertainty envelopes will become bigger. Uncertainties were already calculated for the derived volumetric growth and mobilization rates and are shown in Table 1. We envision to discuss these uncertainties in more detail as this concern was also raised by Purinton (reply 2021) and Frankl (reply 2021).

**4. Underlying geomorphology**

We agree that lavaka development over time follows different phases. However, in this study we do not consider the specific evolution of lavaka over time.

Based on empirical evidence, we establish a general relationship between lavaka areas and volumes. The geomorphological evolution of growing lavaka (they might indeed become shallower at the final stages) is implicitly embedded in this relationship. We calculate current mobilization rates based on measured area changes over the period 1949-2010s, where we are confident that we can obtain total volumes of eroded sediment with their respective uncertainties over this temporal interval (see also reply to Purinton (reply 2021)).

We agree that more justification is needed for the fact that we derive area-volume relationships as is often done for landslides, as opposed to length-volume relationships which is more typical for gullies. This concern was also shared by Frankl (reply 2021). The main reason to use area instead of length is the specific shape and growth of lavaka, which typically both widen and lengthen as they grow. Given the large variety of lavaka shapes we argue that lavaka areas will be more precise in establishing a correct volume relationship. This rationale will be added in further detail to the revised version of the manuscript. We want to point out that the choice of using an area-volume relationship, which is typically used for landslides, does not imply that the applied bias correction is based on landslide modelling assumptions. The bias correction is a statistical concept to correct for changes in coefficients when transforming fitted coefficients from a linear fit through log-transformed data to coefficients of a power function on non-transformed data (this principle is for example well described for the establishment of suspended sediment rating curves (Ferguson, 1986; Crawford, 1991)).

**5. Bulk densities**

We agree that bulk densities will be lower for the deeper layers and will use the proposed bulk density of 1.1-1.2 t/m³ as mentioned in the literature for our revised mobilization rate calculations.

6. **Comparison of current mobilization rates with long-term $^{10}$Be erosion rates**

We agree that we should be cautious in comparing long-term $^{10}$Be erosion rates derived from river sediments with the calculated current lavaka mobilization rates. Indeed, a large part of this mobilized sediment will be trapped close to the lavaka and will not reach the rivers. The lake infill data that we refer to (Mietton *et al.*, 2005) are, however, also recent infill data (1987-2005) and will only entail the sediment that has reached the lake and is therefore not deposited close to the lavaka. The reported lake sedimentation rate of 20 ton ha$^{-1}$ yr$^{-1}$ is less than half of our obtained mobilization rates over the period 1949-2010s, suggesting that indeed a considerable proportion of the sediment that is mobilized by lavaka will not reach the rivers or lakes. These recent lake sedimentation rates are, however, still almost two orders of magnitude higher than the long-term $^{10}$Be erosion rates, which corroborates with a recent increase in erosion rates. We will provide a more thorough discussion on these matters with the necessary precautions and caveats to better frame these results.

We hope that we have clarified most of the concerns raised and are looking forward to further discuss or clarify these matters if needed.

Sincerely,

Liesa Brosens on behalf of the co-authors

**References**:

Brosens, L., Broothaerts, N., Campforts, B., Jacobs, L., Razanamahandry, V. F., Van Moerbeke, Q., Bouillon, S., Razafimbelo, T., Rafolisy, T. and Govers, G. (2021) 'Under pressure: Rapid lavaka erosion and floodplain sedimentation in central Madagascar', *Science of The Total Environment*, 806, p. 150483. doi: 10.1016/j.scitotenv.2021.150483.

Cox (reply 2021). CC1: Comment on esurf-2021-64 by Rónadh Cox, 19 Oct 2021. https://doi.org/10.5194/esurf-2021-64-CC1

Crawford, C. G. (1991) 'Estimation of suspended-sediment rating curves and mean suspended-sediment loads', *Journal of Hydrology*, 129, pp. 331–348. doi: 10.1016/0022-1694(91)90057-O.

Ferguson, R. I. (1986) 'River Loads Underestimated by Rating Curves', *Water Resources Research*, 22(1), pp. 74–76. doi: 10.1029/WR022i001p00074.

Frankl (reply 2021). RC2: Comment on esurf-2021-64 by Armaury Frankl Oct 2021. https://doi.org/10.5194/esurf-2021-64-RC2

Mietton, M., Leprun, J.-C., Andrianaivoarivony, R., Dubar, M., Erismann, J., Beiner, M., Bonnier, F., Grisori, E., Rafanomezana, J.-P. and Grandjeaw, P. (2005) 'Ancienneté et vitesse d'érosion des lavaka à Madagascar', *Actes des Journées scientifiques du réseau Erosion et GCES de l'AUF*, (1954), pp. 87–94.

Purinton (reply 2021) RC1: Comment on esurf-2021-64 by Benjamin Purinton, 30 Aug 2021.
https://doi.org/10.5194/esurf-2021-64-RC1

---

## Author Response (AR1)

**Response to reviewers and community comments:**
**Comparative analysis of Copernicus, TanDEM-X and UAV-SfM DEMs to estimate lavaka (gully) volumes and mobilization rates in the Lake Alaotra region (Madagascar)**

Liesa Brosens

December 14, 2021

We would like to thank the reviewers and community for their detailed and constructive comments, which allowed methodological and structural improvements of our analysis and text. The main changes to the manuscript are the following

- Use of Copernicus DEM instead of SRTM as the 30 m resolution DEM
- Quantification of the uncertainties on the volumes, where the interpolation and relative height error have been taken into account. These volumetric uncertainties are propagated throughout all following analysis.
- Splitting up of the results and discussion in separate sections, where methodological uncertainties and implications of the results are now more elaborate

These changes are described in detail below, where you can find our point-by-point response to the reviewers' comments. For clarity, the comments of the reviewers are in *italic black font*, our response is given in green and references to our revised manuscript are in yellow with indication of the lines in the final manuscript in red and in the track-changed manuscript in blue.
We believe that by implementing the suggestions of the reviewers and community we have considerably improved the methodological rigour and quality of the manuscript.

Yours sincerely,

Liesa Brosens, on behalf of all co-authors.

**RC1: Dr. Benjamin Purinton**

*The study of Brosens et al. examines the applicability of three DEMs at three different resolutions (and different sources) to measure the erosion of lavaka (gully) features in Madagascar. The gullies are carefully identified from satellite and aerial imagery at three different dates (1949, 1969, 2011-2018) and their extents are digitized. Following this, pre-erosion surfaces are created for each gully and the volume of excavated sediment is measured. This allows the authors to build area-volume relationships to apply to the three time steps, and measure volumes from area alone. Of great note, they find two order of magnitude higher erosion rates in comparison to cosmogenic radionuclides.*

*First off, I want to thank the authors for exposing me to these dynamic and fascinating geomorphic features. While I am familiar with significant gully erosion in other places, I had never heard of the lavakas of Madagascar, and they are impressive. I read the study with great interest since bridging gaps in satellite and aerial (i.e., drone) measurements to quantify geomorphic processes presents exciting opportunities, albeit with significant challenges. These challenges are both in terms of spatial and temporal resolution differences (which the authors cover) and dataset accuracy (which the authors mention but do not consider). Overall, I found the paper interesting and think it should be published in ESurf, but there needs to be some major revisions.*

*I begin by listing my primary concerns, followed by some more specific comments. The references that are not already included in the submitted manuscript are provided at the end of this review.*

> We thank the reviewer for the in-depth review of our work and for sharing his technical know-how. We are grateful for the many practical tips provided and for the suggested changes, which have made the manuscript more technically rigorous with a deeper discussion.

**Primary Concerns**

**RC1.1 - Separate discussion section**

*The paper could do with a dedicated discussion section. There are many points I make below which would be valid items for a discussion, and other points that may expand the methods and results. Furthermore, the discussion section would allow the authors to more fully place the study and results in the context of other work on gully erosion cited (e.g. Cox et al., 2009, 2010; Perroy et al., 2010; Vanmaercke et al., 2021). Placing the chosen methods, observed volumes, and the causes of (increased?) gully erosion in the context of these other studies will present a fuller, and more citable, study.*

> We do agree that the paper would benefit from a separate discussions section in which we can address both technical and contextual aspects in more detail, which was also a comment raised by the other reviewers. We have now added a separate discussions section, in which we discuss the following aspects in more detail:
>
> L401-429 L530-559: *4.1. Interpolation methods and DEM uncertainties*
>
> L430-484 L560-614: *4.2. Lavaka volumes and area-volume relationships from varying DEM resolutions*
>
> L485-521 L615-656 : *4.3. Lavaka mobilization rates put into perspective*

**RC1.2 - 30 m resolution DEM**

*I do not agree with using the SRTM DEM as the 30 m resolution dataset for a number of reasons. Firstly, as stated by the authors, SRTM uncertainties are often > 5 m, precluding accurate volume calculations in many cases. Secondly, previous work (Smith and Sandwell, 2003; Farr et al., 2007) has shown that the actual resolution of SRTM is likely on the order of 45-60 m. Finally, better open access DEMs exist. The authors could instead use the ALOS World 3D 30 m (available here: https://www.eorc.jaxa.jp/ALOS/en/aw3d30/index.htm) or, the cutting edge Copernicus DEM (available here: https://portal.opentopography.org/datasetMetadata?otCollectionID=OT.032021.4326.1). The ALOS data has low vertical uncertainties and is based on a resampled 5 m DEM (cf. Purinton and Bookhagen, 2017) and the Copernicus DEM is essentially a 30 m version of the TanDEM-X (references: https://spacedata.copernicus.eu/documents/20126/0/GEO1988-CopernicusDEM-RP-001_ValidationReport_V1.0.pdf and https://spacedata.copernicus.eu/documents/20126/0/GEO1988-*

*CopernicusDEM-SPE-002_ProductHandbook_I1.00.pdf) I strongly recommend the authors utilize the newer Copernicus DEM in place of SRTM. It may be the case that the newer dataset can be included in the breakpoint analysis, especially considering that the largest lavaka make up the majority of exported sediment (as shown in Figure B2). If a 30 m open-access, near-global DEM (Copernicus or ALOS) provides some decent results, then the impact of this study would be greatly increased.*

We were happy to be introduced to this new 30 m Copernicus DEM, which will indeed provide many benefits compared to the 30 m SRTM DEM. We have therefore replaced the 30 m SRTM DEM by the 30 m Copernicus DEM for all our analysis and have adapted the manuscript accordingly where the Copernicus DEM and its uncertainties are now described in section 2.2.3:

*L135-146 L151-164: The 1 arcsecond (ca. 30 m) resolution global Copernicus DEM (GLO-30) was released in 2021 by the European Space Agency (ESA) and AIRBUS. The DEM is based on the WorldDEM$^{TM}$ which, on its turn, is based on edited and smoothed radar satellite data acquired during the TanDEM-X Mission (AIRBUS, 2020a). The reported global absolute vertical accuracy is 2.17 m with a RMSE of 1.68 m. The relative vertical accuracy is smaller than 2 m for <20° slopes and less than 4 m on >20° slopes (AIRBUS, 2020b). Given its recent release, only limited additional validation has been carried out (Guth and Geoffroy, 2021). A lower absolute vertical accuracy of GLO-30 has been reported for mountainous areas in Europe with RMSE values between 7 and 14 m (Marešová et al., 2021). These estimates should, however, be viewed as maximum estimates as these high relief terrains are one of the most challenging terrains for DEM acquisitions. Upon comparison of different global 1 arcsecond DEMs, Purinton and Bookhagen (2021) concluded that the Copernicus DEM provides the highest quality landscape representation and should be the preferred DEM for topographic analysis in areas that lack higher resolution DEMs. They furthermore report a high inter-pixel consistency for both the TanDEM-X and Copernicus DEM, indicating low relative vertical errors for these DEMs.*

Whereas in the previous version of the manuscript we did not consider the 30 m resolution DEM after the initial volume comparison, we have now continued the rest of the analysis not only on the UAV-SfM and TanDEM-X DEMs, but also on the 30 m resolution DEM. The breakpoint method also allowed the identification of the point below which the reconstructed volumes clearly suffer from errors as evidence by the difference between the positive and total volume for the

Copernicus DEM. This, however, proved to be insufficient for the accurate establishment of an accurate area-volume relationship for the Copernicus DEM, which still strongly deviates from the UAV-SfM and TanDEM-X relationships. This observation is described in the results and discussion:

*L362-368 L469-483: While fitting the relationship only through the data above the breakpoint results in an area-volume relationship within uncertainty of the ground truth UAV-SfM relationship for the TanDEM-X DEM this is not the case for the Copernicus DEM. Even when keeping only the data above the identified breakpoint, the fitted area-volume relationship still largely deviates from the UAV-SfM relationship, with a lower scaling coefficient and higher intercept (Fig. 5, Eq. (5) and Eq. (7)). Given the large discrepancy between the UAV-SfM and Copernicus relationship the latter will not be used for further calculations of the volumetric growth and mobilization rates.*

[Figure]

***Figure 5. Breakpoint analysis and final A-V relationships.*** *(a) The breakpoint is identified as the point where the RMSE from the 1:1 line of the log-transformed positive (V pos) and total (V tot) volumes is smaller than 1%. The identified breakpoint is located at log(V pos) = 3.41±0.24 $m^3$ for TanDEM-X and at 5.07±0.16 $m^3$ for Copernicus. (b) Linear area-volume relationships fitted through the log-transformed lavaka area and volume data for the full UAV-SfM dataset and for the TanDEM-X and Copernicus volumes that exceed the identified breakpoints (log(Vpos)>3.41±0.24 $m^3$ for TanDEM-X and > 5.07±0.16 $m^3$ for Copernicus). Shaded areas indicate the 95% confidence intervals of the fitted relationships and the standard deviation of the breakpoints. Grey error bars are the standard deviations of the mean calculated volumes representing the total uncertainty (interpolation and relative DEM uncertainty).*

*L440–454 L571-585: While for the TanDEM-X DEM the volumes for features larger than the breakpoint closely match those obtained from the UAV-SfM DEM, this is not the case for the Copernicus DEM (Fig. 5(b)). This indicates that for the TanDEM-X DEM the largest volumetric errors are contained within the percentage negative volume, as the breakpoint corresponds to the point where the TanDEM-X volumes no longer deviate from the volumes obtained from the UAV-SfM DEM (Fig. 3(a)). Furthermore, this also resulted in an area-volume relationship for TanDEM-X that is within uncertainty of the UAV-SfM relationship (Fig. 5, Eq. (5) and Eq. (6)). A large deviation between the area-volume relationship obtained for the Copernicus DEM and UAV-SfM DEM*

*remained, even when considering only the lavaka located above the breakpoint (Fig. 5(b)). For the Copernicus DEM the absence of negative volumes in the total volume estimate thus seems to be an insufficient measure to accurately estimate lavaka volumes. This might be related to a second factor that affects estimated volumes, which is the DEM smoothness. The smoothing effect of coarser resolution DEMs on landscape topographical representation is known to result in a reduced ability to capture more complex topography and geomorphic features (Thompson et al., 2001; Wechsler, 2007; Tarolli, 2014; Hengl,2006). The underestimation of eroded volumes when coarser resolution DEMs are used was also reported by Claessens et al. (2005), who found that the highest landslide erosion and deposition volumes were estimated for the highest resolution DEM and systematically decreased when reducing the DEM resolution. This effect was attributed to the more detailed landscape representation for higher resolution DEMs.*

**RC1.3 - Vertical uncertainties**

*Vertical uncertainties of the different DEMs are never considered in the analysis. But this is a vital step when using spaceborne DEMs for volume estimations, for instance in the cryospheric (e.g., Brun et al., 2017) and geomorphic (e.g., Purinton and Bookhagen, 2018; Bessette-Kirton et al., 2018) communities. While I think the authors can safely argue for negligible vertical uncertainties (with the proper citations) in the UAV-DEM, these cannot be ignored in the case of TanDEM-X and SRTM (cf. Purinton and Bookhagen, 2017, 2018). In the case of the Copernicus DEM I reference above, TanDEM-X uncertainties can be used given this DEM was generated from the same source. Preliminary reports on Copernicus DEM accuracy are available here: https://spacedata.copernicus.eu/documents/20126/0/GEO1988-CopernicusDEM-RP-001_ValidationReport_V1.0.pdf. Furthermore, depending on the presence of vegetation (i.e., forests) in the DEM pixels, these different datasets (radar X-band for TanDEM; C-band for SRTM; optical images for ALOS) may have additional uncertainties. These uncertainties with regards to the study area characteristics (bare-earth, forests, bushes) should be mentioned.*

*I think the handling of vertical uncertainties can be done in a discussion section, but it may be better inserted in the methods and results. I suggest: an uncertainty value (e.g., RMSE or NMAD) is selected from the literature regarding each DEM and this uncertainty is propagated to the volume estimates. This would then put error bars on the regressions in e.g., Figures 5 and 6, which could be considered during the power-law fitting. This uncertainty will also propagate in the negative vs. positive volume calculations, presenting a range of percentages rather than an exact value, which implies perfect DEM accuracies.*

*In Purinton and Bookhagen (2018) we also attempted volume estimation (in this case using the more common DoD approach) between the SRTM and TanDEM-X DEMs. In this case, the uncertainties associated with the SRTM precluded widespread geophysical results, except in the areas of very rapid / large magnitude change. This study highlights the influence of spaceborne DEM accuracy and the care that must be taken when combining older and newer datasets from different sources, with different errors, in a given analysis. While detailed correction steps are not necessary in this case, this work should be noted, particularly since it is in the same journal.*

We have now implemented vertical uncertainties for the different DEMs in three ways:

1) For each DEM a description of the vertical accuracy and precision has been added to the material and method sections where the DEMs are described (section 2.2):

*L116-122 L125-131: This method was reported to result in a robust and accurate alternative for georeferencing based on ground control points (GCP) with a MAE of 0.02 m and RMSE of 0.03 m for the vertical accuracy and a precision of 0.04 m (Zhang et al., 2019). Comparable studies over relatively flat areas with an UAV-RTK setup report similar vertical accuracies with RMSE values between 0.03 and 0.07 m (Taddia et al., 2020; Stott et al., 2020). UAV-SfM surveys with GCP's over more complex terrain report higher RMSE values between 0.10 and 0.45 m (Clapuyt et al., 2016; Cook, 2017). Given the reported high accuracies of optical acquisitions that are georeferenced with RTK-GPS data, this DEM surface can be considered as the reference of the 'true' elevation (Grohmann, 2018). In this study we therefore consider the UAV-SfM DEM as the ground-truth reference.*

*L131-134 141-144: A good performance of the TanDEM-X DEM has been reported, with a final global absolute vertical accuracy of 3.49 m and relative vertical accuracy of 0.99 m and 1.37 m on flat (<20°) and steep (>20°) terrain, respectively (Rizzoli et al., 2017). These results are in line with Wessel et al. (2018) and Purinton and Bookhagen (2017) who reported absolute vertical accuracies of 0.20±1.5 m and 1.41±1.97 m, respectively.*

*L138-143 L155-161: The reported global absolute vertical accuracy is 2.17 m with a RMSE of 1.68 m. The relative vertical accuracy is smaller than 2 m for <20° slopes and less than 4 m on >20° slopes (AIRBUS, 2020b). Given its recent release, only limited additional validation has been carried out (Guth and Geoffroy, 2021). A lower absolute vertical accuracy of GLO-30 has been reported for mountainous areas in Europe with RMSE values between 7 and 14 m (Marešová et al., 2021). These estimates should, however, be viewed as maximum estimates as these high relief terrains are one of the most challenging settings for DEM acquisitions.*

2) We have now quantified the vertical uncertainties for all DEMs by addressing the interpolation error and the relative height error. Both methods are described in detail in the corresponding section that has been added to the material and methods (section 2.4).

[revised manuscript text omitted]

**RC1.4 - Interpolation**

*In the interpolation step, I think some more work and justification is needed. I see the temptation to generate curved surfaces using splines, but I think nearest neighbor void filling is somewhat safer (not based on any parametrized curve fitting) and simpler. This may result in "staircase" artifacts, but those may not have a huge impact on the final volume estimates, or the vertical uncertainties of the spaceborne DEMs may have a larger impact. The authors should experiment with the GDAL gridding methods accessible in QGIS (Processing Toolbox > GDAL > Raster Analysis: GRID (nearest neighbor, linear, etc.)), since these are robust and widely used. Previous authors working on gullies have had success generating pre-erosion surfaces with simpler (i.e. parameter-free) interpolation (e.g., Perroy et al., 2010; Eustace et al., 2009, Evans and Lindsay, 2010). Furthermore, I'm not convinced the random-points approach is entirely necessary. Couldn't the authors just use each elevation value that the "horseshoe" overlaps with one time? Otherwise some elevations (single pixels) are used multiple times, which I don't understand the reason for and seems inappropriate.*

*I suggest the authors at least attempt the processing with the nearest neighbor approach, mentioning that this is parameter free and uses only the original elevation values, and if the results create significantly more negative volume, then mention this as justification for higher order techniques. This can be an item in the discussion section.*

Two concerns were raised by the reviewer: i) the use of parameterized spline interpolation and ii) random-points approach.

Interpolation method

We have now improved our method to assess which interpolation method works best by assessing the interpolation error on 50 intact hillslopes where we have verified the performance of five different interpolation methods:

i)      Linear (GDAL)

ii)     TIN (Qgis)

iii)    Spline: Bilinear (GRASS)

iv)    Spline: Bicubic (GRASS)

v)     Spline: regularized with tension (GRASS)

We did some manual try-outs with the nearest neighbor (GDAL) algorithm that was proposed by the reviewer. Visual interpretation of these results (Figure R1), together with the fact that this method is typically applied to categorical data made us decide to not consider this method in further analysis. (see reply to RC1.3 and text L214 L258).

[Figure]

Figure R1: Examples of four different interpolation methods for the TanDEM-X DEM: Regularized spline with tension, TIN, Linear and Nearest neighbor interpolation.

From the different error metrics (Table 1) it was concluded that regularized spline with tension yields the lowest interpolation errors. Therefore, we decided to use this interpolation method for all further analysis. The caveats that come with this interpolation method are added to the discussion (see reply to RC1.3 and text L402-409 L531-539).

Random-points approach

We agree with the reviewer that the creation of the random points in the pre-erosion surface polygon is not necessary and might result in incorrect interpolation results as then indeed the same pixel value might be considered multiple times during interpolation. We have therefore changed this method by following the suggestion of the reviewer where we now create one point per pixel, which are then used for interpolation. The steps followed to derive the lavaka volumes are now the following:

L158-176 L185-195:

*STEP 1: Interpolate the pre-erosion surface*

*First, the DEM raster layer is clipped with the horseshoe-shaped polygon in order to extract the pixels not affected by gully erosion. All pixels that fall within this polygon are extracted in order to have a minimum width of one pixel. Next, one point per clipped DEM pixel is generated and used as input for the interpolation. Finally, these points are used to interpolate the pre-erosion surface. Five interpolation methods were tested, of which the method with the lowest error was applied to the lavaka dataset (see section 2.4.1 and section 3.1). Examples of the interpolated pre-erosion surface are shown in Fig. 2 for TIN (b) and regularized spline (c) interpolation.*

*STEP 2: Calculate elevation difference*

*The current DEM is subtracted from the interpolated pre-erosion surface. The result is a difference raster with positive values indicating a current surface that is lower than the reconstructed pre-erosion surface. Negative values indicate that the current topography is higher than the reconstructed topography.*

*STEP 3: Elevation difference clipped to lavaka extent*

*The lavaka extent, which is given by the digitized lavaka polygon, is clipped from the elevation difference raster. In this way a raster with the elevation difference over the lavaka area is obtained (Fig. 2(b)-(c)). If the lavaka is smaller than one pixel (0.04 m², 144 m² and 900 m² for the UAV-SfM, TanDEM-X and Copernicus DEM, respectively) the resulting raster is empty and no volume can be calculated.*

*STEP 4: Export results*

*The unique values report of the lavaka elevation difference raster is exported. It contains the unique elevation values, their count and dimensions of the raster pixels. These results are used to calculate the volumes of each lavaka.*

**Specific Comments**

**RC1.5** *Line 3: "high resolution DEMs", I prefer that the exact resolution in meters is mentioned or the terms high, medium, and low resolution are clearly defined and a range of values for each is given. Twenty years ago 30 m DEMs were very high resolution. For posterity, best to be exact. Please note this change in other places in the manuscript (e.g. Line 31)*

> We agree that the terms "high", "medium" and "low" resolution change throughout time. Therefore, we now only use the exact resolution of the DEMs (0.20, 12 or 30 m resolution):
>
> *L30-33 L31-35: Obtaining sub-meter resolution DEMs from UAV-SfM still requires substantial fieldwork and is spatially limited due to the nature of the technology (Bangen et al., 2014). On the other hand, TanDEM-X is a spaceborne product with global coverage at 12 m resolution and, while being less detailed and accurate than these sub-meter resolution DEMs, is a major step forward in comparison to the 30 m DEMs with a global coverage (Mudd, 2020).*
>
> *L39 L41-42: More recently, however, (sub-)meter resolution DEMs have enabled the development of (semi-)automated gully...*
>
> *L44-45 L47-49: This latter question is relevant since sub-meter resolution surface imagery from a multitude of sources and moments in time is now globally and freely available.*
>
> *L55-58 L58-60: Here, we evaluate the performance of TanDEM-X to estimate gully volumes and to establish area-volume relationships by comparing estimates obtained from a 0.2 m resolution UAV-SfM, the 12 m resolution TanDEM-X DEM and the 30 m resolution Copernicus DEM.*
>
> *L76-77 L80-83: This procedure was followed for a 0.2 m resolution UAV-SfM DEM, the 12 m resolution TanDEM-X DEM and the 30 m resolution Copernicus DEM.*
>
> *L95-98 L100-103: Lavaka volumes were determined from three digital elevation models with a range of horizontal resolutions. For two study areas a 0.2 m resolution UAV-SfM DEM was obtained from a field campaign in 2018. For all study areas the 12 m and 30 m resolution TanDEM-X and Copernicus DEMs are available.*

**RC1.6** *Line 10: "SRTM DEM is too coarse", or too inaccurate? This pertains to one of my primary concerns. Also bear in mind the SRTM realistically has a ground resolution closer to 45-60 m (Smith and Sandwell, 2003; Farr et al., 2007)*

> Given that we have now replaced the SRM DEM with the 30 m Copernicus DEM which is derived from the 12 m resolution TanDEM-X DEM, we are now more confident that it is the loss of information due to the coarser resolution rather than the inaccuracy of the 30 m DEM that causes the incorrect volume estimates. We have therefore retained this wording.

**RC 1.7** *Line 19-20: What do these different rates correspond to? Are they the range and average of the six different study areas?*

> These rates indeed indicate indeed the range for the different study areas and the average for the full dataset. This is added to the text:

*L18-20 L19-21: Our calibrated area-volume relationship enabled us to obtain large-scale lavaka mobilization rates ranging between 18±3 and 311±82 t ha$^{-1}$ yr$^{-1}$ for the six different study areas, with an average of 108±26 t ha$^{-1}$ yr$^{-1}$ for the full dataset.*

**RC1.8** *Line 26: Rephrase "more and more"*

We restructured the sentence as follows, where "more and more" was replaced by "increasingly":

*L27 L28-29: Over the past decades advanced technology has become increasingly available for the assessment of surface topography*

**RC1.9** *Line 30: "remote sensing product". Well, UAVs are also remote sensing if we define remote sensing as measurements that don't disturb the surface. I would change this to "spaceborne product"*

Changes as suggested:

*L31 L32-33: On the other hand, TanDEM-X is a spaceborne product*

**RC1.10** *Line 32: They don't necessarily need to all be here, but somewhere (perhaps in a new dedicated section) references to the accuracy of these various datasets should be cited (e.g., Purinton and Bookhagen, 2017; Rizzoli et al., 2017; Wessel et al., 2018)*

The information on the accuracy of the different DEMs and corresponding references is added to the material and methods section where the absolute and relative vertical accuracy of the TanDEM-X DEM is now added to the text (see reply RC1.3, text L116-122 L125-131, L131-134 L141-144, L138-143 L155-161).

**RC1.11** *Line 36: "Gully erosion…", awkward sentence, rephrase.*

Rephrased as follows:

*L37-38 L40 -41: The mapping and monitoring of gully erosion was conventionally based on time consuming and spatially limited field surveys.*

**RC1.12** *Line 39: From "where", awkward clause. Consider new sentence and/or rephrasing.*

Split up the sentence in two sentences and rephrased:

*L39-42 L41-45: More recently, however, high resolution DEMs have enabled the development of (semi-)automated gully-delineation and volume determination methods (Niculita et al., 2020; Evans and Lindsay, 2010, Perroy et al., 2010; Eustace et al, 2009, Liu et al., 2016). TanDEM-X has, for example, already been successfully used for automatic gully detection (Orti et al., 2019).*

**RC1.13** *Line 41: What is being referred to by "high resolution surface imagery"? If the authors are referring to GoogleEarth then be explicit, and I suggest also referencing Fisher et al. (2012).*

We have explicitly stated Google Earth imagery and referred to the proposed work:

*L44-46 L47-49: This latter question is relevant since sub-meter resolution surface imagery from a multitude of sources and moments in time is now globally and freely available through, for example Google Earth (Fisher et al., 2012).*

**RC1.14** *Line 63-64: For this step the lavaka area was taken from the most recent (2010s) polygons? Maybe state this.*

It has been added to the text that the lavaka volumes were calculated for the 2010s lavaka polygons:

*L70-72 L75-77: In a first step, lavaka volumes were calculated for the 2010s lavaka polygons from the DEM as the difference between a reconstructed pre-erosion surface and the current topography. Next, a lavaka area-volume relationship was established between the current lavaka areas (2010s) and calculated volumes.*

**RC1.15** *Line 80: "are available" maybe change to "were generated", since these are data the authors created for this study. And what a nice dataset it is!*

Replaced as suggested:

*L86-89 L92-95: For the six selected study areas digitized lavaka polygons were generated from orthorectified and georeferenced historical aerial images from 1949 and 1969 (2.4 m resolution) and from recent (2011-2018 referred to as 2010s) satellite imagery (WorldView-2, 0.5 m resolution) (Fig. 1(a), Table B1).*

**RC1.16** *Line 81 and Table A1: I'm not familiar with Maxar-Vivid-WVO2, but I suppose this refers to the WorldView-2 satellite? Please call them WorldView-2 if so. Also, are these images proprietary or received through grant? They should be mentioned in the "Code and data availability" section. One more point: the ground resolution of the aerial photos and satellite images should be mentioned here (and/or in the table).*

Maxar-Vivid-WVO2 indeed refers to the WordView-2 satellite. We changed this in the text and corresponding Table (Table B1) and have added the spatial resolution of both the satellite and historical pictures imagery both to the text (L86-89 L92-95, see reply RC1.16).

Table B1. Study area characteristics and imagery availability. The availability of the 1949 and 1969 aerial images is indicated by a cross and the satellite acquisition dates are reported. For each study area its surface area, number of lavaka and resulting lavaka density are indicated.

| Study area | Surface [km²] | Aerial picture 1949 | Aerial picture 1969 | Satellite aquisition date | Satellite source | Number of lavaka | Lavaka density [lavaka km⁻²] |
|---|---|---|---|---|---|---|---|
| 1 | 11.47 | X | X | 27/05/2018 | WorldView-2 | 153 | 13 |
| 2 | 10.47 | X | X | 12/09/2011 | WorldView-2 | 128 | 12 |
| 3 | 15.29 | X | X | 10/07/2016 | WorldView-2 | 140 | 9 |
| 4 | 10.48 | X | | 29/05/2018 | WorldView-2 | 173 | 17 |
| 5 | 11.27 | X | X | 27/05/2018 | WorldView-2 | 55 | 5 |
| 6 | 11.98 | | X | 27/05/2018 | WorldView-2 | 50 | 4 |

The WorldView-2 images were available as a baselayer in Arcmap software of Esri. This is added to the "Code and data availability section":

*L556-557 L684-685: WorldView-2 imagery was available as a baselayer in ArcMap software from Esri.*

**RC1.17** *Figure 1: The color-scale could be improved here and elsewhere using e.g. the instructions here: https://gis.stackexchange.com/questions/94978/elevation-color-ramps-for-dems-in-qgis. The authors can decide for themselves, but the current blue to red scale is odd for topography. The red-blue could then be saved and used for topographic difference as is often done (e.g., Wheaton et al., 2010). Furthermore, in (a) the topography around the Alaotra catchment should also be shown, otherwise it looks like this implies ocean around it, which makes the inset map of Madagascar also confusing. Note the Krieger (2007) citation appears twice, drop one of them.*

We thank the reviewer for his feedback on the maps and figures. We have incorporated his suggestions, where the topography is now displayed using one of the CPT-city QGIS Color Ramps for topography, which was slightly modified. The topography surrounding the Alaotra catchment is now also displayed to avoid confusion with the island of Madagascar in the inset. We have removed the first reference to Krieger (2007).

[Figure]

***Figure 1. Study areas, examples of each digital elevation model (DEM) and lavaka examples.** (a) Six study areas of ca. 10 km² in the Lake Alaotra catchment shown on the TanDEM-X DEM with hillshade. The UAV-SfM DEM is available for study area 5 and 6 (red) (data collected June 2018). (b)-(d) Examples of the Copernicus (AIRBUS, 2020a), TanDEM-X (Krieger et al., 2007) and UAV-SfM DEM (data collected June 2018) in SA6 located at 48°15'18.6"E 17°58'51.7"S with hillshade. Grey outlines indicate the digitized lavaka. (e) UAV fish-eye picture from 200 m height of the eastern ridge shown in (b)-(d). (f) UAV fish-eye picture (200 m height) from two typical amphitheater-shaped lavaka (pictures taken June 2018).*

The elevation differences are now displayed using a blue to red gradient with discretized colors and legend:

[Figure]

**Figure 2.** *Lavaka volume determination workflow. Lavaka volumes were calculated for each individual lavaka following an automated workflow for the three studied DEMs: UAV-SfM (0.20 m, top, data collected June 2018), TanDEM-X (12 m, middle, Krieger et al. (2007)) and Copernicus (30 m, bottom, AIRBUS (2020a)). (a) The digitized lavaka outline (grey), manually determined horseshoe-shaped polygon on the unaffected hillslope surrounding the lavaka and current DEM are the three required inputs for the automated volume-procedure. The DEM pixels that are not affected by erosion are clipped from the DEM with the horseshoe-shaped polygon and one point per pixel is generated. The pre-erosion surface is then reconstructed by interpolating between these points (STEP 1). Two interpolation methods are shown as an example here: TIN (b) and regularized spline (c) interpolation. The grey polygon indicates the outer edge of the interpolated area. The elevation difference between the interpolated pre-erosion surface and current DEM surface is then calculated, which is clipped to the lavaka extent (STEP 2-3). (d) Cross sections of transect A for the DEM, TIN and regularized spline interpolation.*

[Figure]

**Figure C2. Interpolation error workflow.** *The interpolation error was assessed by placing 50 lavaka polygons and corresponding pre-erosion polygons on intact hillslopes. The difference between the interpolated surface and the DEM gives the interpolation error. This is done for all three DEMs (UAV-SfM (0.2 m), TanDEM-X (12 m) and Copernicus (30 m)) and by using five different interpolation methods (Linear, TIN, Spline bilinear, Spline bicubic and Spline regularized).*

**RC1.18** *Line 88-89: The method of resampling to UTM coordinates should be mentioned (bilinear?). Also, importantly, the SRTM is likely referenced to the EGM96 geoid, whereas the TanDEM-X is referenced to the WGS84 ellipsoid. Was a vertical datum conversion done to bring the datasets into the same vertical reference? And what is the vertical datum for the UAV DEM?*

> We transformed the DEMs from WGS84 to UTM coordinates by using a nearest neighbor resampling method. This has been added to the text:

> L95-96 L102-103: All DEMS were transformed to WGS84-UTM39S (EPSG: 32739) coordinates using a nearest neighbor resampling method.

> The TanDEM-X and the UAV-SfM DEM are referenced to the WGS84 ellipsoid, the Copernicus DEM is referenced to the EGM2008 geoid. We did not carry out any vertical datum conversions as we believe this is not needed for our analysis: Elevation differences are always measured as the difference between an interpolated surface and the DEM surface. While a datum conversion would be needed when we would compare the absolute elevations differences between the DEMs, we don't think this is need for relative elevation differences within the same DEM. In order to avoid confusion, we have not mentioned these vertical reference systems in the manuscript.

**RC1.19** *Line 95-99: A few points here. While the UAV DEM was likely of high quality, the authors should confirm that there is no notable doming effect from using a fish-eye lens and no ground control points. Doming (cf. James and Robson, 2014) is a known issue with SfM-MVS from drones, and fish-eye lenses will exacerbate this. I suggest the authors examine the raw UAV point cloud in either Pix4D or perhaps CloudCompare over flat surfaces in the study area (I realize this may be difficult to find) and take visual note of any large-scale warping. Ideally a reference dataset or independent GNSS points could be used for validation, but that is missing here, correct? One more thing: there are many other citations regarding UAV-DEMs for geomorphic analysis and I recommend including them alongside or in place of Grohmann, 2018 (e.g. Cook, 2017 and others therein and referencing this study). The authors could even give a range of expected vertical accuracy from their UAV-DEM taken from the literature in other cases where GCPs were not used. This would provide justification for passing these off as "negligible".*

> Since we indeed don't have a reference dataset or GNSS points we could not evaluate the accuracy or precision of our UAV-SfM DEM. We have verified the possible doming effect by visual inspection of the point cloud over some flat rice paddy areas that were present in the DEMs. We could visually not directly see clear large-scale warping or doming in the vertical plane over these flat areas. The vertical doming effect of the UAV-RTK set-up of Zhang et al (2019) that we applied was verified by them, where they report vertical differences between check points smaller than 0.07 m and therefore conclude that vertical doming can be greatly eliminated or mitigated due to the dense and precise control of camera positions. Flights were furthermore carried out with a slightly tilted camera and in a course-aligned way, resulting in oblique images and the imaging of overlapping areas under a different angle, which is reported to reduce error propagation and doming (James and Robson, 2014). The discussion with regard to this possible doming effect is added to the discussion:

*L424-429 L554-559: A possible caveat when using a fish-eye camera for UAV image acquisition is vertical 'doming'. However, our flights were carried out with a slightly tilted camera and in a course-aligned way, resulting in oblique images with overlapping areas under a different angle, which is reported to reduce error propagation and doming (James and Robson, 2014). Possible vertical doming could only be verified visually in our case by inspecting the point cloud of flat surfaces in the study area, since no independent GNSS dataset is available. Visual inspection (Fig. C8) and reported vertical deviations less than 0.07 m by Zhang et al. (2019) who's set-up was adopted here, confirm that this effect is likely minimal.*

We furthermore added two examples of the point clouds over relatively flat rice paddies to the appendices:

[Figure]

**Figure C8. UAV-SfM point clouds over flat areas.** *In order to verify the presence of vertical doming due to the use of a fish-eye lens for the UAV-SfM DEM, the point clouds are visually inspected over flat surfaces. Visual inspection does not indicate the presence of vertical doming.*

We have also added more information and previously reported values of the accuracy and precision that can be expected from similar UAV-SfM set-ups, where the needed extra references are added to the text:

*L116-119 L125-129: This method was reported to result in a robust and accurate alternative for georeferencing based on ground control points (GCP) with a MAE of 0.02 m and RMSE of 0.03 m for the vertical accuracy and a precision of 0.04 m (Zhang et al., 2019). Comparable studies over relatively flat areas with an UAV-RTK setup report similar vertical accuracies with RMSE values*

*between 0.03 and 0.07 m (Taddia et al., 2020; Stott et al., 2020). UAV-SfM surveys with GCP's over more complex terrain report higher RMSE values between 0.10 and 0.45 m (Clapuyt et al., 2016; Cook, 2017).*

**RC1.20** *Line 116: The auxiliary files delivered with TanDEM-X (the COV.tif file) would allow the authors to specifically report the number of coverages used to generate the final DEM in this study area. I suggest reporting this value (mean +/- standard deviation, or range) since it has a large impact on DEM quality (vertical uncertainty).*

Based on the COV.tif file we have determined the range and mean +- std coverages over our study areas, which are now added to the text:

*L130-131 L140-141: Our study areas were imaged 5 to 9 times with an average of 7±1, indicating a good coverage.*

**RC1.21** *Line 124-125: As noted, these height errors are really high for the SRTM and this is likely an inappropriate dataset for this application, particularly when I consider the bottom row of Figure 2 where the SRTM differences is maybe less than 5 m? Although again, as I note below, an improved (classified) color-scale would be helpful here.*

We have replaced the 30 m SRTM DEM with the 30 m Copernicus DEM and have now discretized and adapted the colorscales (see also response to RC1.2 and RC1.17).

**RC1.22** *Line 141: What is meant by "precise identification"? In this case, is the horseshoe drawn to not include the other lavakas? So a sort of broken horseshoe shape? Please provide a visualization of the points selected for interpolation on Figure B1. From this figure I don't see in many cases how enough points could be selected to provide a reasonable interpolation in some of the locations where the lavakas appear to have no, or very little, pre-erosion topography preserved where they are touching.*

We have now added the pre-erosion polygons to the figure of one of the most lavaka-dense parts of the study area to better show how the horseshoe-shaped polygons were constructed. In some cases, no polygon could be delineated given the bare absence of pixels unaffected by gully erosion. In other cases, we grouped lavaka into one bigger enveloping polygon to be able to interpolate the original surface. We have added this additional information to the text:

*L180-183 L212-215: Second, the very dense presence of lavaka often results in a highly dissected topography and a near absence of topography not affected by lavaka erosion, requiring a precise identification of the areas not affected by erosion. For some lavaka no horseshoe-shaped polygons could be delineated. In other cases, lavaka were grouped in one enveloping polygon when they were located next to each other (Fig. C1).*

[Figure]

*Figure C1. **Example of near absence original surface topography.** Example from study area 1 illustrating the near absence of the original surface topography (especially in the western part of the area) due to the dense presence of lavaka (grey outlines). Grey bands indicate the pre-erosion surface polygons that could not be derived for all lavaka, and sometimes envelope multiple lavaka that are located next to each other. Elevations from the TanDEM-X DEM with hillshade (Krieger et al., 2007).*

**RC1.23** *Line 146: Delete hyphen in "DEM-pixel"*

Changed as suggested:

*L161 L187: Next, one point per clipped DEM pixel is generated*

**RC1.24** *Line 160-162: I'm concerned about this 1-pixel lower limit. A single pixel can easily represent an inaccurate measurement. It would be better to have a multi-pixel lower limit. This could be 5 pixels (i.e., lavakas smaller than 5 pixels in each DEM are not considered), but that would remove a significant number of lavakas from the e.g. 30 m DEM. On the other hand, per the scaling relationships, perhaps these <5 pixel lavakas do not contribute significantly to the sediment budget?*

We have now translated the determined breakpoint not only in an area-limit but also in a pixel-limit. This confirms that no accurate volumes can be obtained when considering only 1 pixel and that for the TanDEM-X DEM a minimum of 6±2 pixels is required, which increases to 14±5 pixels for the Copernicus DEM:

*L350-353 L440-443: This breakpoint is for the TanDEM-X DEM located at a positive volume of ca. 2500 ± 1500 m³ and corresponding surface area of ca. 800 ± 250 m² or 6 ± 2 pixels. For the Copernicus DEM this point is located at a positive volume 355 of ca. 120 000 ± 45 000 m³, corresponding to a lavaka surface area of 13 000 ± 3500 m² or 14 ± 5 pixels (Fig. 5(a)).*

**RC1.25** *Figure 2: The elevation colorbar is reversed with respect to Figure 1, please check it here and elsewhere (and as suggested consider different color scales). In this figure the elevation difference colorbar should really be classified, not continuous. For instance, if I look at the SRTM it's hard to tell but those green values are maybe < 10 m (maybe even < 5 m)? That's getting awfully close to the vertical uncertainty of SRTM. Classified color scales broken into ~5 m ranges with perceptually distinct colors would help a lot.*

We have changed the colorscale of both the elevation and elevation differences, where we have now also plot the elevation differences in discretized intervals of 5 m (Figure 2, see response RC1.17).

**RC1.26** *Line 216: See primary concerns above. Here the uncertainties on the a and b coefficients may be too small, since the regression does not consider uncertainties on the volume calculation which may be significant for the TanDEM-X and SRTM (if the authors continue to use the SRTM and not ALOS or Copernicus).*

We have now propagated the uncertainties on the calculated volumes into the area-volume fitting by running a Monte Carlo analysis where we fit the linear relationship for each of the simulated volume ensembles. This results in $10^5$ linear fits, from which the mean and std of the fitted *a* and *b* coefficients is calculated, representing the uncertainty on the fitted coefficients. This did, however, not result in a higher uncertainty on the fitted coefficients. This might be explained by the fact that the uncertainty on the fitted coefficients and corresponding uncertainties and plotted 95% confidence intervals represent the expected variation in the *mean* estimated volumes given a specific area. If we would on the other hand look at the prediction intervals (not plotted), these have become wider by implementing the uncertainties on the individual volumes, as the prediction interval gives the 95% range in which the next *individual* volume estimate would fall given a specific area. As we are here interested in the general relationship between area and volume, and the corresponding uncertainties in the *mean* estimated values, these show to be barely impacted by the volumetric uncertainties. We have further clarified the meaning of these uncertainties on the fitted coefficients in the text:

*L268-273 L313-317: The volumetric uncertainties are propagated into the area-volume relationship by fitting the linear relationship for all the $10^4$ volume estimates from the Monte Carlo simulation and calculating the mean and std of the fitted a and b coefficients. These uncertainties are plotted as the 95% confidence intervals and represent the expected variation in the mean estimated volume given a specific area. They do not represent the range in which the next individual volume estimate would fall given a specific interval (i.e. the prediction interval).*

**RC1.27** *Line 226: "1.3% vs. 0.3%" and others, values should be switched (0.3 vs. 1.3) to match the preceding sentence.*

This part of the text has been removed since we have changed our methodology concerning the selection of the best interpolation method.

**RC1.28** *Line 233: "increasing" should be "decreasing"*

This part of the text has been removed since we have changed our methodology concerning the selection of the best interpolation method.

**RC1.29** *Line 241: Remove hyphen in "data-areas"*

Changed as suggested:

*L403-405 L532-534: Bergonse and Reis (2015) concluded that spline interpolation methods result in smaller errors compared to linear methods as they are better adjusted to a gully geomorphic context by allowing curved surfaces in no data areas, which is not the case for linear interpolation methods.*

**RC1.30** *Line 260: Reference Smith et al. (2019) (in this journal) when discussing "optimal grid resolution"*

We thank the reviewer for pointing us to this interesting work and have added the reference to the section where we discuss the optimal grid resolution:

*L433-434 L563-564: A first factor that explains this observation is the dependence of the optimal DEM grid resolution on the inherent properties and scale of the geomorphic features under study (Tarolli, 2014; Hengl, 2006; Smith et al., 2019).*

**RC1.31** *Line 275: "at" should be "of"*

Corrected:

*L403-405 L532-534: Bergonse and Reis (2015) concluded that spline interpolation methods result in smaller errors compared to linear methods as they are better adjusted to a gully geomorphic context by allowing curved surfaces in no data areas, which is not the case for linear interpolation methods.*

**RC1.32** *Line 295-298: No mention of TanDEM-X vertical uncertainties. Here and elsewhere the uncertainties are only considered with regards to resolution.*

See response RC1.3.

**RC1.33** *Line 308-313: The scaling relationships are considered in the context of landslide studies. Do these results really hold for gully scaling? Can the authors include references or justification for the landslide comparison? I could accept an argument that the scaling relationship is similar, but the processes are different.*

We have now added a paragraph to the discussion where we first repeat the rationale on why we have opted for an area-volume instead of length-volume relationship, which is typically done for gullies. Next, we compare the obtained scaling coefficient with those expected for landslides, where we now clearly state that the processes are entirely different and that we can merely expect the same volume for a given area for deep landslides and lavaka:

*L472-479 L603-609: Gully volumes are typically linked to gully length as most gullies mainly lengthen when they grow (Frankl et al., 2013; Vanmaercke et al., 2021). Lavaka, on the contrary, deepen, widen and lengthen when they grow, which is why we link lavaka volume with area instead of length. While this does not allow direct comparison with other relationships obtained for gullies, previous studies reported that length-volume relationships are region-specific (Frankl et al., 2013). Applying the observed relationship outside of the lake Alaotra region should therefore*

*be done with care and might require validation. While the processes of landslide and lavaka erosion are entirely different, the obtained scaling coefficient a of 1.44±0.04 indicates that for a given area, lavaka volumes will be similar to those of deep landslides that typically have an a between 1.3 and 1.6) (Larsen et al., 2010).*

**RC1.34** *Figure 6: Am I correct in my visual interpretation that the break-point is found at the point where the relationship becomes 1:1? In that case, is the broken-stick analysis entirely necessary, or could a simpler approach be to just consider where the RMSE (or some measure of spread) from a 1:1 line passes below some limit (e.g. 5%). Just food for thought, I think the broke-stick is valid, but it may be worth mentioning this 1:1 change-point.*

We are indeed interested in the point where the observations no longer strongly deviate from the 1:1 line of Vpos-Vtot. As the proposed methodology where you consider the point where the RMSE from the 1:1 line passes below a given limit is more intuitive and does not require a more complex broken-stick regression algorithm, we have implied this method to determine the breakpoint. Based on a comparison with the broken-stick regression and visual analysis we have set the limit at 1% RMSE.

*L348-351 L435-440: Therefore, we tried to identify the point below which the analysis based on TanDEM-X and Copernicus suffers from errors in volume reconstruction as evidenced by negative volume pixels. This breakpoint was identified as the point where the RMSE from the 1:1 V pos-V tot line becomes smaller than 1%.*

**RC1.35** *Line 322-323: And this is why uncertainties on the volume estimates are important for the coefficient estimation (small differences in coefficients lead to large differences in volumetric growth and mobilization).*

Indeed, but as argued above propagating the uncertainties of the volumes does not lead to large differences in the estimated uncertainties on the coefficients (see reply to RC1.26).

**RC1.36** *Lines 337-345 and Figure 7: This would be good fodder for a discussion section and would allow expansion and inclusion of other references to gully studies (e.g. Vanmaercke et al., 2021 and references therein). I recommend removing the correlation coefficients from plots b-d in Figure 7 and where they are referenced in the text. These are only six data points and it may be best to just discuss the graphical trends observed, since these are not robust statistics in this case.*

We have removed the correlation coefficients from the plots and the text and have added a separate discussion section in which these results are further elaborated upon:

*L384-387 L510-514: This can be explained by the positive correlation between lavaka area and volumetric growth rate (r = 0.27, p = 1e-10, Fig. 6(a)): larger lavaka mobilize more material. LMR also increase with increasing lavaka density (Fig. 6(c)), which is logical but is also partially explained by the positive correlation between lavaka density and mean lavaka surface area (Fig.6(d)). The main variations in LMR between our six study areas thus seem to depend mainly on the lavaka density and area distribution.*

[Figure]

**Figure 6. Variations in volumetric growth rates and lavaka mobilization rates.** *a) Lavaka volumetric growth rates (VGR) are positively related with lavaka area (spearman correlation coefficient r = 0.27, p = 1e-10). b) Lavaka mobilization rates (LMR) are higher for study areas with larger lavaka. Mean lavaka areas are indicated by the cross in the boxplot. Higher lavaka mobilization rates are linked to higher lavaka densities (c), which are also positively correlated with lavaka area (d). n indicates the number of observations and the error bars indicate the standard deviation of the mean LMR as obtained from the Monte Carlo simulations taking into account the uncertainties on the fitted a and b coefficients.*

*L513-521 L647-656: Globally reported volumetric gully erosion rates range between 0.0002 and 47 430 m³ yr⁻¹, with mean and median values of 359 and 2.2 m³ yr⁻¹ (Vanmaercke et al., 2016). Our mean and median estimated volumetric growth rates of 1149 ± 275 m² yr⁻¹ and 320 ± 56 m³ yr⁻¹ are at least three times higher than these global averages, indicating that lavaka erosion in the lake Alaotra catchment is occurring at above average gully erosion rates. These reported volumetric gully growth rates correspond to global mean and median aerial gully growth rates of 3.1 and 131 m² yr⁻¹ (Vanmaercke et al., 2016), whereas the mean and median aerial lavaka growth rates for our lavaka dataset are 22 and 11 m² yr⁻¹ (Brosens et al., 2022). This indicates that while volumetric lavaka growth rates are higher than the global averages, their change in aerial extent is below average. This is caused by the specific morphology of lavaka, which are much deeper than average gullies with estimated mean and median depths for our dataset of 23 and 19 m based on the calculated volumes and areas, whereas this is only 2.1 and 1.3 m for the global dataset (Vanmaercke et al., 2016).*

**RC1.37** *Line 345: This Brosens et al. (in review) paper is seemingly important for the discussion of results (reasons for gully changes). Hopefully review progresses there quickly and a final citable result is available for this paper / discussion. This paper leaves me asking "why are the lavaka erosion rates increasing?" In Perroy et al. (2010) (worth citing here) gully erosion increased in response to grazing.*

The Brosens et al. (2022) paper has by now been published in Science of the Total Environment (doi: 10.1016/j.scitotenv.2021.150483). This allows a better discussion of the possible causes of the increased erosion levels, which we link indeed to increasing human and cattle populations. This has been added to the discussion:

*L508-511 L644-646: This was also concluded by Brosens et al. (2022), where a tenfold increase in floodplain sedimentation rates was observed over the past 1000 years, which was linked to a recent increase in lavaka activity brought about by increasing environmental pressure due to growing human and cattle populations (Joseph et al., 2021).*

**RC1.38** *Line 361: "Table A1 and 1", not sure what the 1 is supposed to refer to.*

Here we refer to Table A1 and Table 1. This is now added to the text:

*L497-499 L630-632: The reported recent lake sedimentation rate of 20 t ha$^{-1}$ yr$^{-1}$ is less than half of our calculated lavaka mobilization rate of 53±19 t ha$^{-1}$ yr$^{-1}$ for SA3 which has a comparable lavaka density of 9 lavaka km$^{-2}$ (Fig. 6(c), Table B1 and Table 2).*

**RC1.39** *Line 386: "area" should be "are".*

Corrected

**RC1.40** *I had a look at the three example lavaka files and code on GitHub, thanks for publishing that, it is really useful. However, when I tried to run the LavakaVolumesPyQGIS.py script, I received error messages about importing modules and functions. There are no imports at the top of the script, is this something that is missing? Or could the authors add to the GitHub README the steps to actually run the script? Maybe this is done through a GRASS shell, but I'm not familiar with those steps. I tried adding "import os" and "from qgis.core import *" but then got an error about an undefined "processing" variable. These do not need to be detailed instructions, but the common way of running a script at the command line (python <script name>) does not work in this case, so maybe just note the steps to open and run the script.*

We regret to notice that we forgot to add the first two lines of the code in the online repository where the necessary packages are imported. We have now added the updates scripts and example dataset containing the updated workflow. We also added a description of how the scripts can be run from withing QGIS, as this clearly required more detailed instructions. The updates files can be found here: https://doi.org/10.5281/zenodo.5768418. This link has been updated in the manuscript (L150 L168, L296 L342 and L550 L687).

*I hope these are helpful and constructive criticisms, and I welcome further discussion in the open online forum.*

*Sincerely,*

*Ben Purinton*

**RC2: Prof. Dr. Armaury Frankl**

**Primary Concerns**

**RC2.1** *Well done. It is a well presented case study on gully quantification using RS products. Given that this is a technical paper, I would give more context on specific, but well-known, issues in volumetric quantifications using RS, such as the 2.5D problem caused by gully bank undercutting, or DEM vs DSM when vegetation was not filtered out. Also, from a theoretical point of view, one can already assess whether the pixel size is fine enough to study a landform of a particular size; and this should already be mentioned from the start.*

*Please see detailed comments and suggestions in the pdf. After considering suggestions, I consider that with minor revisions this paper can be accepted for publication.*

We would like to thank the reviewer to go through our manuscript in detail and for providing very useful suggestions on how to strengthen the discussion, pointing us to possible technical caveats or clarifying the text.

We have now added a discussion on the several possible issues in volumetric quantification using remote sensing, such as the 2.5D problem and the possible impact of vegetation:

[revised manuscript text omitted]

**RC2.3** *Line 27: Use gender-neutral writing, uncrewed instead of unmanned.*

Changed as suggested:

*L3-4 L3-4: Moreover, ongoing developments in uncrewed aerial vehicle (UAV) technology …*

*L27-28 L28-29: Over the past decades advanced technology has become increasingly available for the assessment of surface topography: SfM (structure-from-motion) algorithms applied to UAV (uncrewed aerial vehicle) imagery now allow centimeter-scale resolution, …*

**RC2.4** *Line 39: Be aware that gully volume estimation from aerial imagery yield errors due to undercut areas and complex morphologies and not being visible and thus, providing 2.5D and not real 3D models when using, for example, UAV. This should be discussed as it is generally overlooked. See also discussion here: Frankl A., Stal C., Abraha A., Nyssen J., Rieke-Zapp D., De Wulf A., Poesen J.2015. Detailed recording of gully morphology in 3D through image-based modelling. Catena 127: 92-101. So if you only compare RS products, you may not never really assess the accuracy of the different approaches with the real volume.*

We have now added a discussion section (section 4.1) in which we cover the concept of 2.5 vs 3D models and discuss the possible impacts on our volume estimates. (see reply RC2.1).

**RC2.5** *Line 44: More often, Length - Volume relationships are used, since gullies are linear features and area is not always easy to map. This could be different for the gullies in Madagascar but could be mentioned. Be aware that G-A relationships may not be easy to apply to other gullies wordwide.*

We now mention these length-volume relationships upfront in the introduction:

*L46-48 L49-51: This imagery can be used to identify geomorphic features and estimate their surface area with great detail. Area-volume or length-volume relationships then enable to obtain estimates of volume-changes over time when historical imagery from which areas or lengths can be derived is available.*

In the material and methods we have added the rationale on why in the case of lavaka area-volume seems more appropriate:

*L266-267 L310-312: As lavaka typically have a specific inverse-teardrop shape and both lengthen and widen when they grow (Wells et al., 1991) we use lavaka area instead of length as a size measure.*

We now also come back to this in the discussion:

*L472-476 L603-607: Gully volumes are typically linked to gully length as most gullies mainly lengthen when they grow (Frankl et al., 2013; Vanmaercke et al., 2021). Lavaka, on the contrary, deepen, widen and lengthen when they grow, which is why we link lavaka volume with area instead of length. While this does not allow direct comparison with other relationships obtained for gullies, previous studies reported that length-volume relationships are region-specific (Frankl et al., 2013). Applying the observed relationship outside of the lake Alaotra region should therefore be done with care and might require validation.*

**RC2.6** *Line 50: I appreciate a case-study based approach, but there is also a theoretical approach to is. I have dealt with this myself some years ago and wrote "Landforms should have dimensions of at least twice the DEM resolution to be defined in a grid-based DEM (Warren et al., 2004)." See Frankl, A., Poesen, J., Scholiers, N., Jacob, M., Haile, M., Deckers, J., Nyssen, J., 2013. Factors controlling the morphology and volume (V)-length (L) relations of permanent gullies in the northern Ethiopian Highlands. Earth Surf. Process. Landforms 38, 1672–1684. Before jumping into a case study approach, the theoretical limiations should be discussed too.*

We agree that it is more interesting to estimated upfront which lavaka size would have been necessary for a given DEM to extract lavaka. We have now added this concept upfront in the introduction, where we couple back to this in the discussion where we compare these theoretical lower limits with the identified breakpoints. (see response RC2.1)

**RC2.7** *Line 54: So indeed, V-A because of the specific shape.*

See reply RC2.5

**RC2.8** *Line 58: I don't think there is something as a 'conventional' gully. Surface feeder channels are also often not present elsewhere.*

We agree that 'conventional' was not the best wording and have replaced this with 'other':

*L63-64 L67-69: Unlike other gullies they typically lack surface feeder channels and tend to form on mid-slopes, broadening uphill trough headward erosion (Wells et al., 1991; Wells and Andriamihaja, 1993).*

**RC2.9** *Line 62: The first estimate is almost certainly an underestimation (undercutting problem). This could be mentioned in the discussion. If less relevant for your study area, at least this would be very important for gullies elsewhere. Bank undercutting is a major process in gully development when erosion rates are very high.*

We have now added a section where we discuss the 2.5D vs. 3D and the corresponding volume underestimation to the discussion (L411-414 L540-543, see reply RC2.1).

**RC2.10** *Figure 1: Shadow in the gullies may have caused image matching problems. And trees (or other vegetation) in the model result in a DSM and not a DEM, which could also result into underestimations. Given it is a technical paper, these issues should be discussed.*

We have added a section to the discussion where we now address in detail the possible remaining errors in the UAV-SfM DEM that are linked to shadowing and the remaining vegetation (L414-416 L540-544 L543-545, see reply RC2.1).

**RC2.11** *Line 108: So basically you are saying that the vegetation is part of the true surface. For a technical paper, this is a major limitation that I find difficult to justify.*

See response RC2.10 and RC2.1.

**RC2.12** *Line 109: Resolution is one thing, but what about the error assessment.*

We have now assessed interpolation and relative accuracy errors and uncertainties on the calculated volumes (see RC1.3).

**RC2.13** *Title 2.3 quantification?*

Changed as suggested:

*L147 L165: 2.3 Lavaka volume quantification*

**RC2.14** *Line 139: The last lines are beyond the scope of step 0, input data.*

We agree that this information is too detailed to add to the different steps. We have moved (and slightly extended) this information on the horseshoe shaped polygons that are placed on the hillslope parts surrounding the lavaka that are unaffected by gully erosion to the text after the list of different steps.

*L177-183 L209-215: A manual delineation of the pre-erosion surface polygons was preferred over an automated pre-erosion interpolation (e.g. Evans and Lindsay, 2010) or interpolation based on the lavaka outlines for two reasons. First, digitized lavaka outlines from aerial imagery are often*

*located on DEM pixels that already have lower values. This is especially the case for the coarser resolution DEMs due to surface smoothing (e.g. Fig. 1(c)-(d)). Second, the very dense presence of lavaka often results in a highly dissected topography and a near absence of the original surface, requiring a precise identification of the pre-erosion surface locations. For some lavaka no pre-erosion surface could be delineated. In other cases, lavaka were grouped in one enveloping pre-erosion polygon when they were located next to each other (Fig. C1).*

**RC2.15** *Line 142: "Creating point" isn't really a step. I think it would make more sense to group these steps into one, it all considers the reconstruction of the original topography before gullying.*

We have now grouped the previous step 1-3 into one new step (STEP 1) in which the interpolation of the pre-erosion surface is described:

*L158-164 L185-195: STEP 1: Interpolate the pre-erosion surface*

*First, the DEM raster layer is clipped with the horseshoe-shaped polygon in order to extract the pixels not affected by gully erosion. All pixels that fall within this polygon are extracted in order to have a minimum width of one pixel. Next, one point per clipped DEM pixel is generated and used as input for the interpolation. Finally, these points are used to interpolate the pre-erosion surface. Five interpolation methods were tested, of which the method with the lowest error was applied to the lavaka dataset (see section 2.4.1 and section 3.1). Examples of the interpolated pre-erosion surface are shown in Fig. 2 for TIN (b) and regularized spline (c) interpolation.*

**RC2.16** *Line 171: A tree would be a perfect suspect for negative values. Not sure if this is less likely..*

The possible impact of the presence of vegetation is now added to the discussion (L416-423 L546-553, see reply RC2.1).

**RC2.17** *Line 183: Ok, but some of the typical errors, as I indicated about could at least be mentioned, instead of keeping is a bit vague.*

The uncertainties of the different interpolation methods and DEMs are now quantified (section 4.2) and further discussed in section 4.1 (see also response to RC3.1).

**RC2.18** *Figure 2: Ok, but a cross section would probably be more informative to see the actual result on how the slope topography was reconstructed. Clearly, the gullies are in a valley.*

We have now added cross-sections to Figure 2 and to the interpolated surfaces displayed in Figure C2 in Figure C3:

[Figure]

**Figure 2.** *Lavaka volume determination workflow.* *Lavaka volumes were calculated for each individual lavaka following an automated workflow for the three studied DEMs: UAV-SfM (0.20 m, top, data collected June 2018), TanDEM-X (12 m, middle, Krieger et al. (2007)) and Copernicus (30 m, bottom, AIRBUS (2020a)). (a) The digitized lavaka outline (grey), manually determined horseshoe-shaped polygon on the unaffected hillslope surrounding the lavaka and current DEM are the three required inputs for the automated volume-procedure. The DEM pixels that are not affected by erosion are clipped from the DEM with the horseshoe-shaped polygon and one point per pixel is generated. The pre-erosion surface is then reconstructed by interpolating between these points (STEP 1). Two interpolation methods are shown as an example here: TIN (b) and regularized spline (c) interpolation. The grey polygon indicates the outer edge of the interpolated area. The elevation difference between the interpolated pre-erosion surface and current DEM surface is then calculated, which is clipped to the lavaka extent (STEP 2-3). (d) Cross sections of transect A for the DEM, TIN and regularized spline interpolation.*

[Figure]

**Figure C2. Interpolation error workflow.** *The interpolation error was assessed by placing 50 lavaka polygons and corresponding pre-erosion polygons on intact hillslopes. The difference between the interpolated surface and the DEM gives the interpolation error. This is done for all three DEMs (UAV-SfM (0.2 m), TanDEM-X (12 m) and Copernicus (30 m)) and by using five different interpolation methods (Linear, TIN, Spline bilinear, Spline bicubic and Spline regularized).*

[Figure]

*Figure C3. Interpolation error cross sections. Cross sections for transect A and B as indicated in Fig. C2 for each of the three DEMs (UAV-SfM (0.2 m), TanDEM-X (12 m) and Copernicus (30 m)) and five interpolation methods (Linear, TIN, Spline bilinear, Spline bicubic and Spline regularized).*

**RC2.19** *Line 259: This comes a bit late, as you would have known this before the study and thus finetune what you really want to understand from your study.*

See RC2.6 and RC2.1

**RC2.20** *Line 298: Would be good to translate this breakpoint into the idea of number of pixels you need versus size of landform before you can really start estimating the volume. Could be a more universally useful concept.*

We have now translated the breakpoint lower limit into to the number of pixels and now also compare these findings with the theoretically expected minimum (see RC2.1):

*L351-353 L440-443: This breakpoint is for the TanDEM-X DEM located at a positive volume of ca. 2500 ± 1500 m³ and corresponding surface area of ca. 800 ± 250 m² or 6 ± 2 pixels. For the Copernicus DEM this point is located at a positive volume 355 of ca. 120 000 ± 45 000 m³, corresponding to a lavaka surface area of 13 000 ± 3500 m² or 14 ± 5 pixels (Fig. 5(a)).*

**RC2.21** *Line 339: So the larger they are the faster they grow. Also a bit obvious no? What about the growth rate since time of first incision of the slope. I saw vegetation growth on the lower section of gullies. Could be see a slowing down of erosion rates in the largest gullies and/or sediment deposition? See also Frankl, A., Nyssen, J., Vanmaercke, M., Poesen, J., 2021. 23 and ion and control: Techniques, failures and effectiveness. Earth Surf. Process. Landforms 46, 220–238.*

Lavaka will grow in different stages, where indeed in the last stage more vegetation will become present in the lavaka when they are stabilizing. We could not see a slowing down in the growth

rates for the largest features (the aerial growth rates of the lavaka dataset are discussed in Brosens et al. (2022)). We have now added a section to the discussion where we shortly explain the different stages of lavaka growth, and how this might impact our area-volume relationship, as stabilized lavaka will likely have a lower volume compared to active lavaka of the same size:

*L480-484 L610-614: In a typical pattern of development lavaka start as raw patches that evolve to step-like headscarps, grow into deep inverse teardrop shaped gullies and finally become longer, broader, gentler and partly filled concavities when stabilizing (Wells et al., 1991). Upon stabilization lavaka will partially fill in, reducing the volume. Not all lavaka will stabilized at the same size, nor grow in the same way. This will likely be one of the main factors explaining the remaining 6 to 9% of variation in lavaka volume that cannot be explained by the area (R2 = 0.94 and 0.91 for TanDEM-X and UAV-SfM, respectively) (Fig. 5).*

**CC1: Prof. Dr. Rónadh Cox**

*This work by Liesa Brosens and colleagues takes a very interesting approach to lavaka analysis in Madagascar, and I applaud their broad thinking.*

*However, I share the concerns about model accuracy and reliability raised by Ben Purinton in his review, and wanted to make a few comments on those issues, in addition to raising some additional questions about the geomorphology. These points are intended in the constructive and collegial spirit of the open review process.*

> We would like to thank Prof. Dr. Cox for taking the time and effort to go through our manuscript and providing us with additional comments. We have tried to address the issues raised in the replies below and are open for further discussion on these topics.

**CC1.1** *The bulk of the image data on which the authors' model is based is not high resolution: only two of their study areas are imaged at 20 cm/pixel, all the others are imaged at 12-30 m/pixel. Many lavakas are only a few pixels across, and elevation changes are in many cases close to the resolution of the imagery, which will not capture internal relief changes within the gullies. It also means that there must be substantial potential error with the area calculation, as a lot of edge detail will not be resolved; and lavaka edges can be quite complex in shape.*

> Detailed lavaka areas have been delineated based on high resolution (0.5 m) satellite imagery over the period 2011-2018 and based on 2.4 m resolution historical aerial images. These datasets are described in detail in Brosens *et al.* (2022) and are provided the corresponding FigShare repository (https://doi.org/10.6084/m9.figshare.c.5236322.v1.). On these previously mapped areas, the area-volume relationships obtained in this manuscript have been applied to estimate the lavaka volumes in 1949 and 2010s.

> To better clarify this to the reader we have added the resolution of the imagery based on which the lavaka areas have been mapped and added the reference to the repository in the text (this repository is also referred to in the data and code availability section).

> *L86-92 L92-97: For the six selected study areas digitized lavaka polygons were generated from orthorectified and georeferenced historical aerial images from 1949 and 1969 (2.4 m resolution) and from recent (2011-2018 referred to as 2010s) satellite imagery (WorldView-2, 0.5 m resolution) (Fig. 1a, Table B1). All shapefiles have WGS84-UTM 39S (EPSG: 32739) coordinates. The dataset (available at https://doi.org/10.6084/m9.figshare.c.5236322.v1) contains the changes in surface area of 699 lavaka over the period 1949-2010s for SA1-5 and 1969-2010s for SA6.*

**CC1.2** *The authors apply a simple least-squares fit to area-volume data to provide a relationship that they then use to drive much of their analysis. I did not find a graph of the XY data that they used, or an $R^2$ value or other measure of quality of fit. But I expect that those data are very noisy, and the relationship is not very precise. To test this, I looked at the data that the authors made available on FigShare, and I plotted area vs relief (as a proxy for the depth data, which was not in the file). I will share two observations. First, the linear fit fails at smaller volumes. This is a problem, because lavaka size follows a power-law distribution, with the majority being at smaller sizes: a robust line of fit should model the bulk of the*

*population, so I think the authors should consider whether it works to have a large proportion of the population not represented by the fit line. Second, the data clouds for the different study areas have different lines of fit, so a one-size-fits-all approach may not be valid.*

The data deposited in the FigShare repository of the Brosens et al., (2020) paper do not allow to repeat the analysis of this paper. They contain the data on the digitized lavaka areas in 1949, 1969 and 2010s but do not contain the derived volumes. The data of this paper can be found in the following repository: https://doi.org/10.5281/zenodo.5768418, which we referred to in the code and data availability section. This repository contains an excel file the original lavaka areas and volumes that were used to establish the area-volume relationship for each of the DEMs. This table also contains the detailed areas of all lavaka and the estimated lavaka volumes based on the established area-volume relationships, as well as the derived volumetric growth and mobilization rates. We have now also added the standard deviations of the calculated volumes as inferred from the uncertainty analysis to this table.

We now also added references to this repository in the text so that the reader can more easily find this data:

*L149-150 L167-168: This was done by developing an automated workflow in PyQGIS written in QGIS version 3.16.10 (code and example dataset available at https://doi.org/10.5281/zenodo.5768418).*

*L295-296 L342: All the calculated volumes, volumetric growth rates and mobilization rates are available at: https://doi.org/10.5281/zenodo.5768418.*

We have now added the equations and R² values of all the fitted aera-volume relationships to the respective figures (Figure 4 and Figure 5b).

[Figure]

***Figure 4. Area-Volume relationships.*** *Fitted linear area-volume relationships between the log-transformed lavaka areas and volumes for (a) the pairwise dataset and (b) the full datasets containing all lavaka volumes for all three DEMs. Grey error bars are the standard deviations of the mean calculated volumes representing the total uncertainty (interpolation and relative DEM*

*uncertainty). Shaded bands indicate the 95% confidence intervals of the fitted relationships where the volumetric uncertainties are propagated through Monte Carlo simulations.*

[Figure]

***Figure 5. Breakpoint analysis and final area-volume relationships.** (a) The breakpoint is identified as the point where the RMSE from the 1:1 line of the log-transformed positive (V pos) and total (V tot) volumes is smaller than 1%. The identified breakpoint is located at log(V pos) = 3.41±0.24 m³ for TanDEM-X and at 5.07±0.16 m³ for Copernicus. (b) Linear area-volume relationships fitted through the log-transformed lavaka area and volume data for the full UAV-SfM dataset and for the TanDEM-X and Copernicus volumes that exceed the identified breakpoints (log(Vpos)>3.41±0.24 m³ for TanDEM-X and > 5.07±0.16 m³ for Copernicus). Shaded areas indicate the 95% confidence intervals of the fitted relationships and the standard deviation of the breakpoints. Grey error bars are the standard deviations of the mean calculated volumes representing the total uncertainty (interpolation and relative DEM uncertainty).*

As we show in Figure 4 there are indeed issues with this area-volume relationship for the smallest lavaka, with strongly deviating calculated volumes for the TanDEM-X, and especially, the Copernicus DEM. Therefore, we apply our breakpoint analysis in order to eliminate lavaka that suffer from erroneous volume estimates as evidenced by the presence of negative volumes. Taking into account only the lavaka that are larger than the identified breakpoint results in a good fit with the area-volume relationship for the UAV-SfM DEM for the TanDEM-X DEM, with R² values of 0.91 and 0.94, respectively. The uncertainties on these fitted coefficients now entail the uncertainties on the calculated volumes, which are indicated in Equations (5)-(7) and are propagated into the subsequent volumetric growth and mobilization rate calculations.

*L268-271 L313-314: The volumetric uncertainties are propagated into the area-volume relationship by fitting the linear relationship for all the $10^4$ volume estimates from the Monte Carlo simulation and calculating the mean and std of the fitted a and b coefficients.*

The lavaka size distribution indeed follows a power-law distribution, where the majority of the lavaka indeed has a smaller size. This is also the main reason why we log-transformed that data, which allows us to fit the relationship for the bulk of the data:

*L264-266 L308-310: A linear relationship is typically fitted on the log-transformed data in order to obtain equally distributed residual errors, resulting in a more robust fit: log(V ) = a+b log(A) (e.g. Guzzetti et al., 2009; Crawford,1991).*

We have quantified the impact of extrapolating the fitted TanDEM-X relationship to smaller lavaka areas that are not used to obtain the final fit (lavaka with areas smaller than the identified breakpoint are not used to establish the final area-volume relationship for the TanDEM-X DEM). We demonstrate that these smallest lavaka where the area-volume fit deviates most do not largely impact the final mobilization rates:

*L392-399 L519-526: In order to further evaluate the possible impact of fitting the TanDEM-X relationship on the larger features only (> 800 ± 250 m²), we quantified the share of the total mobilized sediment that is provided by lavaka smaller than 800 ± 250 m². From the relative cumulative sediment mobilization curves it is apparent that larger lavaka contribute most of the mobilized sediment (Fig. C7). Lavaka that are smaller than the identified threshold contribute 0.2% of the total mobilized sediment in the study areas with the largest lavaka and up to 2.6% in the regions with smaller lavaka (Fig. C7). This indicates that the share of smaller lavaka to the total amount of sediment that is mobilized is generally low in our study areas, therefore reducing the risk of erroneous estimates in the case where these smaller lavaka could not be used to establish the TanDEM-X based A-V relationships.*

We did not fit the A-V relationships for the different study areas separately, as we argue that the most robust fit is obtained using all data:

*L192-194 L235-237: The lavaka volumes as obtained from the different DEMs were compared in two ways: i) pairwise comparison using only the lavaka for which the volume was determined for each DEM enabling a one-to-one comparison, and ii) using all the data in order to establish the most robust relationships calibrated on a higher number of observations.*

**CC1.3** *I also worry about the circularity of using area to derive volume, and then using that derived quantity to an area-volume model, as the authors do in Fig. 5. I would want to see a much more detailed unpacking of the caveats that attend this approach, and in particular I think it would be important to show the original area-volume data used to derive the base relationship. This method may provide interesting ways to look at and think about landscape evolution at a broad scale, but I do think that the noisiness of the base data, and the imperfection of the original line of fit to those data, warrant large error envelopes; and severely limit the precision of downstream models based on that original line of fit. The authors should go into these issues in more detail, because it feels as though they are somewhat brushed under the rug (see comment in previous paragraph about no figure showing the original data relationship with its fit line). It seems to me that there are multiple nested and cumulative uncertainties, which could be more completely and thoroughly addressed/ (I hope I am not missing something obvious here, and will be happy to be corrected if I am).*

The lavaka volumes are derived as the difference between the interpolated pre-erosion surface and the current DEM, which are clipped to the current lavaka extent which has been delineated on the 0.5 m resolution WorldView-2 satellite imagery. This is now stated more explicitly:

*L169-173 L200-205: STEP 3: The lavaka extent, which is given by the digitized lavaka polygons from the 0.5 m resolutionWorldView-2 imagery from 2011-2018, is clipped from the elevation difference raster. In this way a raster with the elevation difference over the lavaka area is obtained (Fig. 2(b)-(c)). If the lavaka is smaller than one pixel (0.04 m², 144 m² and 900 m² for the UAV-SfM, TanDEM-X and Copernicus DEM, respectively) the resulting raster is empty and no volume can be calculated.*

The lavaka area is, in a first step, thus solely used to clip the calculated elevation differences from which the lavaka volumes are derived to the lavaka extent. In a next step, we then establish a relationship between lavaka area and volume. The original data from which the base relationship is derived are shown in Figure 4b where we plot the calculated lavaka volumes versus their area for all the data. The finally applied relationships are shown in Figure 5, after applying the breakpoint method where the smallest lavaka are excluded for the TanDEM-X and Copernicus DEMs.

In the revised version of the manuscript we have now addressed the uncertainties on the calculated volumes by assessing both the interpolation and relative height error. These uncertainties on the obtained lavaka volumes are then propagated by ways of different Monte Carlo simulations (see response to RC1.3).

**CC1.4** *I am concerned also about the underlying geomorphology. Lavakas evolve through different stages, with very different activity levels over time (per the excellent and detailed work of Neil Wells and Benjamin Andriamihaja). Older lavakas are larger, but also evolve to be more shallow than (see Wells et al., (1991) ESPL 16: 189-206, Fig. 3). Each lavaka follows its own adventure in terms of growth, deepening, and shallowing (with the shallowing due in large measure to capture within the lavaka of the erosional materials from its walls). So I have two points here. First is that the known geomorphology of lavakas should probably be considered in any model for their evolution through time; and second, that I would like to see more justification for using landslides as analogues (because although lavakas have headscarps, and evolve via collapse processes, they do not really behave like landslides because of their narrow outfall channels). This may mean that some of the assumptions regarding bias-correction factors may need some adjustment, as these which come out of landslide modelling (per Lines 195-200), and are baked into the authors' model for area-volume relationships. I'm not saying that landslides are an inappropriate analogue; but I am saying that the authors should provide a firm geomorphologic rationale (which I cannot find section 2.4 or elsewhere in the current manuscript).*

Concerning the first point about lavaka geomorphology, we agree that lavaka development over time follows different phases. Based on empirical evidence, we establish a general relationship between lavaka areas and volumes. We argue that the geomorphological evolution of growing lavaka (they will indeed become shallower at the final stages) is implicitly embedded in this relationship, but will likely be one of the main factors resulting in the remaining scatter in the data. This has been added to the discussion:

*L480-484 L610-614: In a typical pattern of development lavaka start as raw patches that evolve to step-like headscarps, grow into deep inverse teardrop shaped gullies and finally become longer, broader, gentler and partly filled concavities when stabilizing (Wells et al., 1991). Upon stabilization lavaka will partially fill in, reducing the volume. Not all lavaka will stabilized at the*

*same size, nor grow in the same way. This will likely be one of the main factors explaining the remaining 6 to 9% of variations in lavaka area that cannot be explained by the volume ($R^2$ = 0.94 and 0.91 for TanDEM-X and UAV-SfM, respectively) (Fig. 5).*

Concerning the second point about using landslides as analogues, we have added the necessary clarifications to the text on why we have chosen to work with area-volume (typically landslides) instead of length-volume relationships (typically gullies):

*L262-268 L306-312: Area-volume (typically landslides) and length-volume (typically gullies) relationships obey a power-law relationship $V = aA^b$, where the predicted volume V for a given area A depends on the scaling exponent a and intercept b (Larsen et al., 2010; Frankl et al., 2013). A linear relationship is typically fitted on the log-transformed data in order to obtain equally distributed residual errors, resulting in a more robust fit: $\log(V) = a+b \log(A)$ (e.g. Guzzetti et al., 2009; Crawford, 1991). As lavaka typically have a specific inverse-teardrop shape and both lengthen and widen when they grow (Wells et al., 1991) we use lavaka area instead of length as a size measure. We have therefore established the relationship between lavaka area and volume by fitting a linear least-squares regression through the log-transformed data (base 10 log).*

*L472-479 L603-609: Gully volumes are typically linked to gully length as most gullies mainly lengthen when they grow without becoming much wider (Frankl et al., 2013; Vanmaercke et al., 2021). Lavaka, on the contrary, deepen, widen and lengthen when they grow, which is why we link lavaka volume with area instead of length. While this does not allow direct comparison with other relationships obtained for gullies, previous studies on length-volume relationships reported that these are region-specific (Frankl et al., 2013). Applying the observed relationship outside of the lake Alaotra region should thus be done with care and might require validation. While the processes of landslide and lavaka erosion are entirely different, the obtained scaling coefficient a of 1.44±0.04 indicates that for a given area, lavaka volumes will be similar to those of deep landslides that typically have an a between 1.3 and 1.6) (Larsen et al., 2010).*

We want to point out that the choice of using an area-volume relationship, which is typically used for landslides, does not imply that the applied bias correction is based on landslide modelling assumptions. The bias correction is a statistical concept to correct for changes in coefficients when transforming fitted coefficients from a linear fit through log-transformed data to coefficients of a power function on non-transformed data (this principle is for example well described for the establishment of suspended sediment rating curves (Ferguson, 1986; Crawford, 1991).:

*L273-276 L317-320: When back-transforming the coefficients of the fitted linear relationship to a power-function a systematic statistical bias enters. This is accounted for by adding a bias-correction factor which depends on the variance $\sigma^2$ (Ferguson, 1986; Crawford, 1991): $V = \exp(a+2.65\ \sigma^2)A^b$. This correction assumes that the residual errors of the fitted linear relationship are normally distributed with a mean of zero and variance $\sigma^2$.*

**CC1.5** *In calculating sediment mobilisation rates, the authors state (Line 210) that they used a bulk density of 1.5 t/m3, based on soil corings 2 m deep. This value is likely to be too high. The surface laterite layer in this area is a couple of m thick, so this is what the cores will have sampled. Below this, and forming the*

*bulk of the material that is evacuated from lavakas, is saprolite, which is highly porous and therefore has lower bulk density. A better value would be 1.1-1.2 t/m₃, in line with previous work (e.g. Heimsath et al. (1997) Nature 388: 358–361, Montgomery (2007) PNAS 104: 13268–13272, and the 2008 UVM thesis of Matt Jungers).*

We have now used the bulk density of 1.2 t m$^{-3}$ as proposed by Montgomery (2007):

*L286-287 L330-331: To obtain LMR, lavaka volumes were converted to mass using a dry bulk density (ρ) of 1.2 t m$^{-3}$ (Montgomery, 2007).*

**CC1.6** *Finally, I query the authors' conclusion "current mobilization rates exceed the long term rates by two orders of magnitude" (Line 370). The problem is that the comparison being made is between apples and pears: although the authors do provide the proviso that "not all mobilised lavaka sediment will end up in the rivers", they are assuming that these very different datasets are comparable. In fact, most of the (limited, for sure) evidence suggests that a lot of lavaka sediment is deposited close to (and even within) the lavakas themselves (similar to the landslides on which they model lavakas). Thus, long-term lake infill and river-sediment-derived erosion estimates will miss this material. If lavaka sediment didn't make it into those archives in the past, then the values from those archives cannot be used as a comparison with mobilisation rates from modern lavakas.*

We agree that we should be cautious in comparing long-term $^{10}$Be erosion rates derived from river sediments with the calculated current lavaka mobilization rates. We have therefore added a more thorough discussion on these matters to the manuscript:

*L494-511 L627-646: Only limited local data is available that can be used to compare these estimates with. A sedimentation rate of 20 t ha$^{-1}$ yr$^{-1}$was obtained by Mietton et al. (2006) for the dammed Bevava lake which is located in the southeastern part of the Lake Alaotra catchment over the period 1987-2005. Lake Bevava has a catchment area of 58 km² with a lavaka density of 8 lavaka km$^{-2}$ (Mietton et al., 2006). The reported recent lake sedimentation rate of 20 t ha$^{-1}$ yr$^{-1}$ is less than half of our calculated lavaka mobilization rate of 53±19 t ha$^{-1}$ yr$^{-1}$ for SA3 which has a comparable lavaka density of 9 lavaka km$^{-2}$ (Fig. 6(c), Table B1 and Table 2). While both estimates are the same order of magnitude, this suggests that a considerable proportion (more than 50%) of the mobilized sediment will likely be trapped close to the lavaka and not reach the rivers or lake.*

*Next to these recent short-term sedimentation rates, long-term catchment wide erosion rates obtained from $^{10}$Be measurements have been reported for the central Malagasy highlands. These $^{10}$Be erosion rates integrate over a timescale of thousands to hundreds of thousands of years and represent long-term averages. Reported long-term $^{10}$Be erosion rates range from 0.16 to 0.54 t ha$^{-1}$ yr$^{-1}$ with the highest rates for the catchments with higher lavaka densities (max. 6 lavaka km$^{-2}$, Cox et al., 2009). Ideally these long-term rates are compared with current sediment yields or sedimentation data (Bartley et al., 2015; Vanacker et al., 2007), as a considerable fraction of the sediment likely never reaches the rivers or lakes. However, the offset of two orders of magnitude between long-term 10Be erosion rates and current lavaka mobilization rates and lake Bevava sedimentation rates suggests that lavaka erosion has increased over recent time periods in the Lake Alaotra region. This was also concluded by Brosens et al. (2022), where a tenfold increase in floodplain sedimentation rates was observed over the past 1000 years, which was linked to a*

*recent increase in lavaka activity brought about by increasing environmental pressure due to growing human and cattle populations (Joseph et al., 2021).*

*I admire the amount of work and data collection that went into this manuscript, and I hope that the authors will accept this critique in the collegial spirit in which I offer it.*

We are very grateful for the comments and suggestions offered and hope that with this reply we have addressed or clarified these concerns.